# Achieving Tractable Minimax Optimal Regret in Average Reward MDPs

**Victor Boone**
victor.boone@univ-grenoble-alpes.fr
Univ. Grenoble Alpes, Inria, CNRS, Grenoble INP, LIG, 38000 Grenoble, France

**Zihan Zhang**
zz5478@princeton.edu
Princeton University

## Abstract

In recent years, significant attention has been directed towards learning average-reward Markov Decision Processes (MDPs). However, existing algorithms either suffer from sub-optimal regret guarantees or computational inefficiencies. In this paper, we present the first *tractable* algorithm with minimax optimal regret of $\widetilde{O}\left(\sqrt{\text{sp}\,(h^*) S A T}\right)$,[1] where $\text{sp}\,(h^*)$ is the span of the optimal bias function $h^*$, $S \times A$ is the size of the state-action space and $T$ the number of learning steps. Remarkably, our algorithm does not require prior information on $\text{sp}\,(h^*)$.

Our algorithm relies on a novel subroutine, **P**rojected **M**itigated **E**xtended **V**alue **I**teration (`PMEVI`), to compute bias-constrained optimal policies efficiently. This subroutine can be applied to various previous algorithms to improve regret bounds.

## 1  Introduction

Reinforcement learning (RL) Burnetas and Katehakis [1997], Sutton and Barto [2018] has become a popular approach for solving complex sequential decision-making tasks and has recently achieved notable advancements in diverse fields of application. RL problems are generally formulated with Markov Decision Processes (MDPs) Puterman [1994], where a learning agent seeks to maximize the rewards that are gathered by interacting with an unknown environment.

This paper focuses on average reward MDPs where the learning agent must maximize the sum of rewards in the long run without any reset mechanism. In this setting, the proper balancing between exploration (i.e., playing sub-optimally to learn the unknown environment) and exploitation (i.e., planning optimally according to the current knowledge), usually known as the *exploration-exploitation trade-off*, is key to learn efficiently. The measure of learning performance that we adopt throughout is the *regret*, that compares the aggregate rewards collected by the learning agent during the learning process to the expected performance of an omniscient agent that knows everything in advance. The seminal work of Auer et al. [2009] provides a *minimax* regret lower bound of $\Omega\left(\sqrt{DSAT}\right)$, where $D$ is the diameter (the maximal distance between two different states), $S$ the number of states, $A$ the number of actions and $T$ the learning horizon. They also provide an algorithm achieving regret $\widetilde{O}\left(\sqrt{D^2 S^2 AT}\right)$, where $\widetilde{O}(-)$. Ever since Auer et al. [2009], many works have been devoted to close the gap between the regret lower and upper bounds in the average reward setting Auer et al. [2009], Bartlett and Tewari [2009], Filippi et al. [2010], Talebi and Maillard [2018], Fruit et al. [2018, 2020], Bourel et al. [2020], Zhang and Ji [2019], Ouyang et al. [2017], Agrawal and Jia [2023],

---

[1] $\widetilde{O}(\cdot)$ hides logarithmic factors of $(S, A, T)$.

38th Conference on Neural Information Processing Systems (NeurIPS 2024).

Abbasi-Yadkori et al. [2019], Wei et al. [2020] and more. Subsequent works Fruit et al. [2018], Zhang and Ji [2019] refined the minimax regret lower bound to $\Omega\left(\sqrt{\text{sp}(h^*)SAT}\right)$ where $\text{sp}(h^*)$ is the span of the bias function, which is the maximal gap of the long-term accumulative rewards starting from two different states. The difference is significant, since $\text{sp}(h^*) \leq D$ and the gap between the two can be arbitrarily large. However, no existing work achieves the following three requirements simultaneously:

(1) The method achieves minimax optimal regret guarantees $\widetilde{O}\left(\sqrt{\text{sp}(h^*)SAT}\right)$;
(2) The proposed method is tractable;
(3) No prior knowledge on the model is required.

Most algorithms simply fail to achieve minimax optimal regret, and the only method achieving it Zhang and Ji [2019] is intractable because it relies an oracle to solve difficult optimization problems along the learning process. Naturally, we raise the question of whether these three requirements can be met all at once:

> **Is there a tractable algorithm with $\widetilde{O}\left(\sqrt{\text{sp}(h^*)SAT}\right)$ minimax regret without prior knowledge?**

**Contributions.** In this paper, we answer the above question affirmatively, by proposing a polynomial time algorithm with regret guarantees $\widetilde{O}\left(\sqrt{\text{sp}(h^*)SAT}\right)$ for average-reward MDPs. Our method can further incorporate almost arbitrary prior bias information $\mathcal{H}_* \subseteq \mathbf{R}^S$ to further improve its regret.

**Theorem 1** (Informal). *Provided that the confidence region used by* PMEVI-DT *satisfy mild regularity conditions (see Assumption 1-3), then for every weakly communicating model M with* $\text{sp}(h^*) \leq T^{1/5}$ *and* $sp(h^*) \in \mathcal{H}_*$, PMEVI-DT$(\mathcal{H}_*, \delta, T)$ *achieves regret:*

$$O\left(\sqrt{\text{sp}(h^*)SAT \log\left(\frac{SAT}{\delta}\right)}\right) + O\left(\text{sp}(h^*)S^{\frac{5}{2}}A^{\frac{3}{2}}T^{\frac{9}{20}} \log^2\left(\frac{SAT}{\delta}\right)\right)$$

*with probability* $1 - 26\delta$. *Moreover, if* PMEVI-DT *runs with the same confidence regions that* UCRL2 *Auer et al. [2009] on a communicating environment, it has a time complexity* $O(DS^3AT)$.

Taking $\delta = \sqrt{1/T}$, we also obtain a $\widetilde{O}\left(\sqrt{\text{sp}(h^*)SAT}\right)$ regret bound in expectation. The geometry of the prior bias region $\mathcal{H}_*$ that PMEVI-DT can support is discussed later (see Assumption 4). It can be taken trivial with $\mathcal{H}_* = \mathbf{R}^S$ to obtain a completely prior-less algorithm.

To the best of our knowledge, PMEVI-DT is the first tractable algorithm with minimax optimal regret bounds (up to logarithmic factors). The algorithm does not necessitate any prior knowledge of $\text{sp}(h^*)$, thus circumventing the potentially high cost associated with learning $\text{sp}(h^*)$. On the technical side, a key novelty of our method is the subroutine named PMEVI (see Algorithm 2) that improves and can replace EVI Auer et al. [2009] in any algorithm that relies on it Auer et al. [2009], Fruit et al. [2018], Filippi et al. [2010], Fruit et al. [2020], Bourel et al. [2020] to boost its performance and achieve minimax optimal regret.

**Related works on average reward MDPs.** For communicating MDPs, the notable work of Auer et al. [2009] proposes the famous UCRL2 algorithm, a mature version of their prior UCRL Auer and Ortner [2006], achieving a regret bound of $\widetilde{O}(DS\sqrt{AT})$. This paper pioneered the use *optimistic* methods to learn MDPs efficiently. A line of papers Filippi et al. [2010], Fruit et al. [2020], Bourel et al. [2020] developed this direction by tightening the confidence region that UCRL2 relies on, and sharpened the analysis through the use of local properties of MDPs, such as local diameters and local bias variances. However, none of these works went beyond regret guarantees of order $S\sqrt{DAT}$ and suffer from an extra $\sqrt{S}$. A parallel direction was initiated by Bartlett and Tewari [2009] with REGAL, obtaining regret bounds scaling with $\text{sp}(h^*)$ instead of $D$, and extending the regret bounds to weakly-communicating MDPs in the mean time. The computational intractability of REGAL is addressed by Fruit et al. [2018] with SCAL, and regret guarantees are further improved by Zhang and Ji [2019] with EBF, eventually reaching optimal minimax regret but loosing tractability.

Another successful design approach is Bayesian-flavored sampling, derived from Thompson Sampling Thompson [1933], that usually replaces optimism. The regret guarantees of these algorithms usually stick to the Bayesian setting however Ouyang et al. [2017], Theocharous et al. [2017], although

Table 1: Comparison of related works on RL algorithms for average-reward MDP, where $S \times A$ is the size of state-action space, $T$ is the total number of steps, $D$ ($D_s$) is the (local) diameter, $\mathrm{sp}(h^*) \leq D$ is the span of the bias vector, $t_{\mathrm{mix}}$ is the worst-case mixing time, $t_{\mathrm{hit}}$ is the hitting time (i.e., the expected time cost to visit some certain state under any policy).

| Algorithm | Regret in $\widetilde{O}(-)$ | Tractable | Comment/Requirements |
|---|---|---|---|
| REGAL Bartlett and Tewari [2009] | $\mathrm{sp}(h^*)S\sqrt{AT}$ | $\times$ | knowledge of $\mathrm{sp}(h^*)$ |
| UCRL2 Auer et al. [2009] | $DS\sqrt{AT}$ | $\checkmark$ | - |
| PSRL Agrawal and Jia [2023] | $DS\sqrt{AT}$ | $\checkmark$ | Bayesian regret |
| SCAL Fruit et al. [2018] | $\mathrm{sp}(h^*)S\sqrt{AT}$ | $\checkmark$ | knowledge of $\mathrm{sp}(h^*)$ |
| UCRL2B Fruit et al. [2020] | $S\sqrt{DAT}$ | $\checkmark$ | extra $\sqrt{\log(T)}$ in upper-bound |
| UCRL3 Bourel et al. [2020] | $D + \sqrt{T \sum_{s,a} D_s^2 L_{s,a}}$ | $\checkmark$ | $L_{s,a} := \sum_{s'} \sqrt{p(s'|s,a)(1 - p(s'|s,a))}$ |
| KL-UCRL Filippi et al. [2010], Talebi and Maillard [2018] | $S\sqrt{DAT}$ | $\checkmark$ | - |
| EBF Zhang and Ji [2019] | $\sqrt{\mathrm{sp}(h)^* SAT}$ | $\times$ | optimal, knowledge of $\mathrm{sp}(h^*)$ |
| Optimistic-Q Wei et al. [2020] | $\mathrm{sp}(h^*)(SA)^{\frac{1}{3}}T^{\frac{2}{3}}$ | $\checkmark$ | model-free |
| UCB-AVG Zhang and Xie [2023] | $S^5 A^2 \mathrm{sp}(h^*)\sqrt{T}$ | $\checkmark$ | model-free, knowledge of $\mathrm{sp}(h^*)$ |
| MDP-OOMD Wei et al. [2020] | $\sqrt{(t_{\mathrm{mix}})^2 t_{\mathrm{hit}}AT}$ | $\checkmark$ | ergodic |
| Politex Abbasi-Yadkori et al. [2019] | $(t_{\mathrm{mix}})^3 t_{\mathrm{hit}}\sqrt{SA}T^{\frac{3}{4}}$ | $\checkmark$ | model-free, ergodic |
| PMEVI-DT (**this work**) | $\sqrt{\mathrm{sp}(h^*)SAT}$ | $\checkmark$ | - |
| **Lower bound** | $\Omega\left(\sqrt{\mathrm{sp}(h^*)SAT}\right)$ | - | - |

Agrawal and Jia [2023] also enjoys $\widetilde{O}(S\sqrt{DAT})$ high probability regret by coupling posterior sampling with optimism. Another line of research focuses on the study of ergodic MDPs, where the environment is such that all states are visited infinitely often under every policy. To name a few, the model-free algorithm Politex Abbasi-Yadkori et al. [2019] attains a regret of $\widetilde{O}((t_{\mathrm{mix}})^3 t_{\mathrm{hit}}\sqrt{SA}T^{\frac{3}{4}})$ where $t_{\mathrm{mix}}$ and $t_{\mathrm{hit}}$ are respectively the mixing and the hitting times of the ergodic environment. By leveraging an optimistic mirror descent algorithm, Wei et al. [2020] achieve an enhanced regret of $\widetilde{O}(\sqrt{(t_{\mathrm{mix}})^2 t_{\mathrm{hit}}AT})$.

We refer the readers to Table 1 for a (non-exhaustive) list of existing algorithms.

## 2 Preliminaries

We fix a finite state-action space structure $\mathcal{X} := \bigcup_{s \in \mathcal{S}} \{s\} \times \mathcal{A}(s)$, and denote $\mathcal{M}$ the collection of all MDPs with state-action space $\mathcal{X}$ and rewards supported in $[0, 1]$.

**Infinite-horizon MDP.** An element $M \in \mathcal{M}$ is a tuple $(\mathcal{S}, \mathcal{A}, p, r)$ where $p$ is the transition kernel and $r$ the reward function. The random state-action pair played by the agent at time $t$ is denoted $X_t \equiv (S_t, A_t)$, and the achieved reward is $R_t$. A policy is a *deterministic* rule $\pi : \mathcal{S} \to \mathcal{A}$ and we write $\Pi$ the space of policies. When coupled with a MDP $M \in \mathcal{M}$, a policy properly defines the distribution of $(X_t, R_t)$ whose associated probability probability and expectation operators are denoted $\mathbf{P}_s^\pi, \mathbf{E}_s^\pi$, where $s \in \mathcal{S}$ is the initial state. Under $M$, a fixed policy has a reward function $r^\pi(s) := r(s, \pi(s))$, a transition matrix $P^\pi$, a gain $g^\pi(s) := \lim \frac{1}{T}\mathbf{E}_s^\pi[R_0 + \ldots + R_{T-1}]$ and a bias $h^\pi(s) :=$ Cesàro-$\lim \mathbf{E}_s^\pi[\sum_{t=0}^{T-1}(R_t - g(S_t))]$, that all together satisfy the Poisson equation $h^\pi + g^\pi = r^\pi + P^\pi h^\pi$, see Puterman [1994]. The *Bellman operator* of the MDP is:

$$Lu(s) := \max_{a \in \mathcal{A}(s)} \{r(s,a) + p(s,a)u\} \tag{1}$$

The *optimal gain* is $g^*(s) := \max_\pi g^\pi(s)$ and the *optimal bias* is $h^*(s) := \max\{h^\pi(s) : \pi \text{ s.t. } g^\pi = g^*\}$.

**Weakly-communicating MDPs.** $M$ is weakly-communicating Puterman [1994], Bartlett and Tewari [2009] if the state space can be divided into two sets: (1) the transient set, consisting in states that are transient under all policies; (2) the non-transient set, where every state is reachable starting from any other non-transient state. In this case, $h^*$ is a span-fixpoint of $L$ (see Puterman [1994]), i.e., $Lh^* - h^* \in \mathbf{R}e$ where $e$ is the vector full of ones. We write $h^* \in \mathrm{Fix}(L)$. Then $g^* = Lh^* - h^*$ and every policy $\pi$ satisfies $r^\pi + P^\pi h^* \leq g^* + h^*$. We accordingly define the *Bellman gaps*:

$$\Delta^*(s,a) := h^*(s) + g^*(s) - r(s,a) - p(s,a)h^* \geq 0. \tag{2}$$

Another important concept is the *diameter*, that describes the maximal distance from one state to another state. It is given by $D := \sup_{s \neq s'} \inf_\pi \mathbf{E}_s^\pi[\inf\{t \geq 1 : S_t = s'\}]$. An MDP is said *communicating*

if its diameter $D$ is finite, in which case $\text{sp}(h^*) \leq \text{sp}(r)D$, see Bartlett and Tewari [2009], Fruit [2019], where $\text{sp}(-)$ is the *span* function given by $\text{sp}(u) := \max(u) - \min(u)$.

**Reinforcement learning.** The learner is only aware that $M \in \mathcal{M}$ but doesn't have a clue about what $M$ further looks like. From the past observations and the current state $S_t$, the agent picks an available action $\mathcal{A}(S_t)$, receives a reward $R_t$ and observe the new state $S_{t+1}$. The *regret* of the agent is:

$$\text{Reg}(T) := Tg^* - \sum_{t=0}^{T-1} R_t. \tag{3}$$

Its expected value satisfies $\mathbf{E}[\text{Reg}(T)] = \mathbf{E}[\sum_{t=0}^{T-1} \Delta^*(X_t)] + \mathbf{E}[h^*(S_0) - h^*(S_T)]$ and the quantity $\sum_{t=0}^{T-1} \Delta^*(X_t)$ will be referred to as the *pseudo-regret*. This paper focuses on *minimax regret guarantees*. Specifically, for $c \geq 1$, denote $\mathcal{M}_c := \{M \in \mathcal{M} : \exists h^* \in \text{Fix}(L(M)), \text{sp}(h^*) \leq c\}$ the set of weakly-communicating MDPs that admit a bias function with span at most $c$. Following Auer et al. [2009], every algorithm **A**, for all $c > 0$, we have

$$\max_{M \in \mathcal{M}_c} \mathbf{E}^{M,\mathbf{A}}[\text{Reg}(T)] = \Omega\left(\sqrt{cSAT}\right). \tag{4}$$

The goal of this work is to reach this lower bound with a tractable algorithm.

## 3 Algorithm `PMEVI-DT`

The algorithm `PMEVI-DT` that we present in this work is actually a general method can be applied to improve various existing algorithms Auer et al. [2009], Filippi et al. [2010], Fruit et al. [2018], Bourel et al. [2020], Tewari and Bartlett [2007]. All these algorithms work episodically, by maintaining a policy $\pi_k$ that drives play during a time-window $\{t_k, \ldots, t_{k+1} - 1\}$ called an episode. An episode rule determines when $\pi_k$ should be considered obsolete and defines the time $t_{k+1}$ at which the policy is renewed. To compute $\pi_k$, these algorithms follow the *optimism-in-face-of-certainty* (OFU) design principle, by choosing $\pi_k$ that achieves the largest possible gain that is plausible under their current information. This is done by building a confidence region $\mathcal{M}_t \subseteq \mathcal{M}$ for the hidden model $M$, then searching for a policy $\pi$ solving the optimization problem:

$$g^*(\mathcal{M}_t) := \sup\{g^\pi(\mathcal{M}_t) : \pi \in \Pi, \text{sp}(g^\pi(\mathcal{M}_t)) = 0\} \text{ with } g^\pi(\mathcal{M}_t) := \sup\{g^\pi(\widetilde{M}) : \widetilde{M} \in \mathcal{M}_t\}. \tag{5}$$

The design of the confidence region $\mathcal{M}_t$ varies from a work to another. Given a confidence region $(\mathcal{M}_t)_{t \geq 0}$, OFU-algorithms work as follows: At the start of episode $k$, the optimization problem (5) is solved, and its solution $\pi_k$ is played until the end of episode. The duration of episodes can be managed in various ways, although the most popular is arguably the *doubling trick* (DT), that essentially waits until a state-action pair is about to double the visit count it had at the beginning of the current episode (see Algorithm 1).

**Notations.** In the rest of this section, we use $\hat{p}_t(s, a)$ (and $\hat{r}_t(s, a)$) to denote the empirical transition (and reward) of the latest doubling update before the $t$-th step, and further denote $\hat{M}_t := (\hat{r}_t, \hat{p}_t)$.

**Extended Bellman operators and EVI.** To solve (5) efficiently, the celebrated work Auer et al. [2009] introduce the *extended value iteration* algorithm (`EVI`), that can be run whenever $\mathcal{M}_t$ is a $(s, a)$-rectangular confidence region, meaning that $\mathcal{M}_t \equiv \prod_{s,a}(\mathcal{R}_t(s, a) \times \mathcal{P}_t(s, a))$ where $\mathcal{R}_t(s, a)$ and $\mathcal{P}_t(s, a)$ are respectively the confidence region for $r(s, a)$ and $p(s, a)$ after $t$ learning steps. `EVI` is the algorithm computing the sequence defined by:

$$v_{i+1}(s) \equiv \mathcal{L}_t v_i(s) := \max_{a \in \mathcal{A}(s)} \max_{\tilde{r}(s,a) \in \mathcal{R}_t(s,a)} \max_{\tilde{p}(s,a) \in \mathcal{P}_t(s,a)} (\tilde{r}(s, a) + \tilde{p}(s, a) \cdot v_i) \tag{6}$$

until $\text{sp}(v_{i+1} - v_i) < \epsilon$ where $\epsilon > 0$ is the numerical precision. When the process stops, it is known that any policy $\pi$ such that $\pi(s)$ achieves $\mathcal{L}_t v_i$ in (6) satisfies $g^\pi(\mathcal{M}_t) \geq g^*(M) - \epsilon$, hence is nearly optimistically optimal. This process gets its name from the observation that $\mathcal{L}_t$ is the Bellman operator of $\mathcal{M}_t$ seen as a MDP, hence `EVI` is just the Value Iteration algorithm Puterman [1994] ran in $\mathcal{M}_t$. A choice of action from $s \in \mathcal{S}$ in $\mathcal{M}_t$ consists in (1) a choice of action $a \in \mathcal{A}(s)$, (2) a choice of reward $\tilde{r}(s, a) \in \mathcal{R}_t(s, a)$ and (3) a choice of transition $\tilde{p}(s, a) \in \mathcal{P}_t(s, a)$; It is an *extended* version of $\mathcal{A}(s)$.

**Towards Projected Mitigated EVI.** Obviously, the regret of an OFU-algorithm is directly related to the quality of the confidence region $\mathcal{M}_t$. That is why most previous works tried to approach the regret lower bound $\sqrt{DSAT}$ of Auer et al. [2009] by refining $\mathcal{M}_t$. The older works of Auer et al. [2009], Bartlett and Tewari [2009], Filippi et al. [2010] have been improved with a variance aware analysis Talebi and Maillard [2018], Fruit et al. [2018, 2020], Bourel et al. [2020] that essentially make use of tightened kernel confidence regions $\mathcal{P}_t$. While all these algorithms successively reduce the gap between the regret upper and lower bounds, they fail to achieve optimal regret $\sqrt{DSAT}$. Meanwhile, the EBF algorithm of Zhang and Ji [2019] is minimax optimal but (1) the algorithm is intractable because it relies on an oracle to retrieve optimistically optimal policies and (2) needs prior information on the bias function. Nonetheless, the method of Zhang and Ji [2019] strongly suggests that inferring bias information from the available data is key to achieve minimax optimal regret.

Rather surprisingly and in opposition to this previous line of work, our work suggests that the choice of the confidence region $\mathcal{M}_t$ has little importance. Instead, our algorithm takes an arbitrary (well-behaved) confidence region in, infer bias information similarly to Zhang and Ji [2019] and makes use of it to refine the extended Bellman operator (6) associated to the input confidence region. Our algorithm can further take arbitrary prior information (possibly none) on the bias vector to tighten its bias confidence region. The pseudo-code given in Algorithm 1 is the high level structure our algorithm PMEVI-DT. In the next Section 3.1, we explain how (6) is refined using bias information.

---

**Algorithm 1:** PMEVI-DT$(\mathcal{H}_*, T, t \mapsto \mathcal{M}_t)$

**Parameters:** Bias prior $\mathcal{H}_*$, horizon $T$, a system of confidence regions $t \mapsto \mathcal{M}_t$

1: **for** $k = 1, 2, \ldots$ **do**
2:   Set $t_k \leftarrow t$, update confidence region $\mathcal{M}_t$;
3:   $\mathcal{H}'_t \leftarrow \texttt{BiasEstimation}(\mathcal{F}_t, \mathcal{M}_t, \delta)$:
4:   $\mathcal{H}_t \leftarrow \mathcal{H}_* \cap \{u : \mathrm{sp}(u) \leq T^{1/5}\} \cap \mathcal{H}'_t$;
5:   $\Gamma_t \leftarrow \texttt{BiasProjection}(\mathcal{H}_t, -)$;
6:   $\beta_t \leftarrow \texttt{VarianceApprox}(\mathcal{H}'_t, \mathcal{F}_t)$;
7:   $\mathfrak{h}_k \leftarrow \texttt{PMEVI}(\mathcal{M}_t, \beta_t, \Gamma_t, \sqrt{\log(t)/t})$ ;
8:   $\mathfrak{g}_k \leftarrow \mathfrak{L}_t \mathfrak{h}_k - \mathfrak{h}_k$ ;
9:   Update policy $\pi_k \leftarrow \texttt{Greedy}(\mathcal{M}_t, \mathfrak{h}_k, \beta_t)$;
10:   **repeat**
11:     Play $A_t \leftarrow \pi_k(S_t)$, observe $R_t, S_{t+1}$;
12:     Increment $t \leftarrow t + 1$;
13:   **until** (DT) $N_t(S_t, \pi_k(S_t)) \geq 1 \vee 2N_{t_k}(X_t)$.
14: **end for**

---

**Algorithm 2:** PMEVI$(\mathcal{M}, \beta, \Gamma, \epsilon)$

**Parameters:** region $\mathcal{M}$, mitigation $\beta$, projection $\Gamma$, precision $\epsilon$, initial vector $v_0$ (optional)

1: **if** $v_0$ not initialized **then** set $v_0 \leftarrow 0$;
2: $n \leftarrow 0$
3: $\mathcal{L} \leftarrow$ extended operator associated to $\mathcal{M}$;
4: **repeat**
5:   $v_{n+\frac{1}{2}} \leftarrow \mathcal{L}^\beta v_n$;
6:   $v_{n+1} \leftarrow \Gamma v_{n+\frac{1}{2}}$;
7:   $n \leftarrow n + 1$;
8: **until** $\mathrm{sp}(v_n - v_{n-1}) < \epsilon$
9: **return** $v_n$.

---

### 3.1 Projected mitigated extended value iteration (PMEVI)

Assume that an external mechanism provides a confidence region $\mathcal{H}_t$ for the bias function $h^*$. Provided that $\mathcal{M}_t$ is correct ($M \in \mathcal{M}_t$) and that $\mathcal{H}_t$ is correct ($h^* \in \mathcal{H}_t$), we want to find a policy-model pair $(\pi, \widetilde{M})$ that maximizes the gain among pairs with $h^\pi(\widetilde{M}) \in \mathcal{H}_t$. This is done with an improved version of (6) combining two ideas, that are both necessary to achieve minimax optimal regret in the analysis.

1. **Projection (Section 3.2).** Whenever it is correct, the bias confidence region $\mathcal{H}_t$ informs the learner that the search of an optimistic model can be constrained to those with bias within $\mathcal{H}_t$. This is done by projecting $\mathcal{L}_t^\beta$ (see *mitigation*) using an operator $\Gamma_t : \mathbf{R}^\mathcal{S} \to \mathcal{H}_t$, that has to satisfy a few non-trivial regularity conditions that are specified in Proposition 2.

2. **Mitigation (Section 3.3).** When one is aware that $h^* \in \mathcal{H}_t$, the *dynamical bias update* $\tilde{p}(s, a)v_i$ in (6) can be controlled better, by trying to restrict (6) to the $\tilde{p}(s, a)$ such that $\tilde{p}(s, a)v_i \leq \hat{p}_t(s, a)v_i + (p(s, a) - \hat{p}_t(s, a))v_i$ with the knowledge that $v_i \in \mathcal{H}_t$. However, controlling the error $(p(s, a) - \hat{p}_t(s, a))v_i$ by doing a union-bound on all possible values of $v_i$ is equivalent to building a confidence region for $p(s, a)$, which produces an extra $S^{1/2}$ in the error term that cannot be afforded by a minimax optimal algorithm.

   We take a different approach instead. For a fixed $u \in \mathbf{R}^\mathcal{S}$, the empirical Bernstein inequality (Lemma 38) provides a variance bound of the form $(\hat{p}_t(s, a) - p(s, a))u \leq \beta_t(s, a, u)$. By

estimating $\beta_t(s, a) := \max_{u \in \mathcal{H}_t} \beta_t(s, a, u)$, the search makes sure that $(\hat{p}_t(s, a) - p(s, a))u \leq \beta_t(s, a)$ holds with high probability for $u = h^*$, even though $h^*$ is unknown. For $\beta \in \mathbf{R}_+^X$, we introduce the $\beta$-*mitigated* extended Bellman operator:

$$\mathcal{L}_t^\beta u(s) := \max_{a \in \mathcal{A}(s)} \sup_{\tilde{r}(s,a) \in \mathcal{R}_t(s,a)} \sup_{\tilde{p}(s,a) \in \mathcal{P}_t(s,a)} \left\{ \tilde{r}(s, a) + \min\left\{ \tilde{p}(s, a)u_i, \hat{p}_t(s, a)u_i + \beta(s, a) \right\} \right\} \quad (7)$$

The mitigation $\beta(s, a)$ is independent of $u$, which is crucial for $\mathcal{L}_t^\beta$ to be well-behaved.

The proposition below shows how well-behaved the composition $\mathfrak{L}_t := \Gamma_t \circ \mathcal{L}_t^\beta$ is. Its proof requires to build a complete analysis of projected mitigated Bellman operators. This is deferred to the appendix.

**Proposition 2.** *Fix $\beta \in \mathbf{R}_+^X$ and assume that there exists a projection operator $\Gamma_t : \mathbf{R}^X \to \mathcal{H}_t$ which is (O1) monotone: $u \leq v \Rightarrow \Gamma u \leq \Gamma v$; (O2) non span-expansive: $\mathrm{sp}\,(\Gamma u - \Gamma v) \leq \mathrm{sp}\,(u - v)$; (O3) linear: $\Gamma(u + \lambda e) = \Gamma u + \lambda e$ and (O4) $\Gamma u \leq u$. Then, the projected mitigated extended Bellman operator $\mathfrak{L}_t := \Gamma_t \circ \mathcal{L}_t^\beta$ has the following properties:*

*(1) There exists a unique $\mathfrak{g}_t \in \mathbf{R}e$ such that $\exists \mathfrak{h}_t \in \mathcal{H}_t, \mathfrak{L}_t \mathfrak{h}_t = \mathfrak{h}_t + \mathfrak{g}_t$;*

*(2) If $M \in \mathcal{M}_t$, $h^* \in \mathcal{H}_t$ and $(\hat{p}_t(s, a) - p(s, a))h^* \leq \beta(s, a)$, then $\mathfrak{g}_t \geq g^*(M)$;*

*(3) If $\mathcal{M}_t$ is convex, then for all $u \in \mathbf{R}^\mathcal{S}$, the policy $\pi =: \mathtt{Greedy}(\mathcal{M}_t, u, \beta)$ picking the actions achieving $\mathcal{L}_t^\beta u$ satisfies $\mathfrak{L}_t u = \tilde{r}^\pi + \tilde{P}^\pi u$ for $\tilde{r}^\pi(s) \leq \sup \mathcal{R}_t(s, \pi(s))$ and $\tilde{P}^\pi(s) \in \mathcal{P}_t(s, \pi(s))$;*

*(4) For all $u \in \mathbf{R}^\mathcal{S}$ and $n \geq 0$, $\mathrm{sp}\left(\mathfrak{L}_t^{n+1} u - \mathfrak{L}_t^n u\right) \leq \mathrm{sp}\left((\mathcal{L}_t)^{n+1} u - (\mathcal{L}_t)^n u\right)$.*

The property (1) guarantees that $\mathfrak{L}_t$ has a fix-point while (2) states that this fix-point corresponds to an optimistic gain $\mathfrak{g}_t$ if the model and the bias confidence region are correct and the mitigation isn't too aggressive. Combined with (3), the Poisson equation of a policy corresponds to this fix-point, i.e., $\tilde{r}^\pi + \tilde{P}^\pi \mathfrak{h}_t = \mathfrak{h}_t + \mathfrak{g}_t$, so that $\mathfrak{g}_t$ is the gain and $\mathfrak{h}_t \in \mathcal{H}_t$ is a legal bias for $\pi$ under the model $(\tilde{r}^\pi, \tilde{P}^\pi)$. Lastly, the property (4) guarantees that the iterates $\mathfrak{L}_t^n u$ converge to a fix-point of $\mathfrak{L}$ at least as quickly as $\mathcal{L}_t^n u$ goes to a fix-point of $\mathcal{L}_t$; The convergence of $(\mathcal{L}_t)^n u$ is already guaranteed by existing studies and is discussed in the appendix.

Provided that the bias confidence region is constructed, Proposition 2 foreshadows how powerful the construction is: The algorithm PMEVI, obtained by iterating $\mathfrak{L}_t$ instead of $\mathcal{L}_t$ in EVI, can replace the well-known EVI within any algorithm of the literature that relies on it (UCRL2 Auer et al. [2009], UCRL2B Fruit et al. [2020] or KL-UCRL Filippi et al. [2010]) for an immediate improvement of its theoretical guarantees.

### 3.2 Building the bias confidence region and its projection operator

The bias confidence region used by PMEVI-DT is obtained as a collection of constraints of the form:

$$\forall s \neq s', \quad \mathfrak{h}(s) - \mathfrak{h}(s') - c(s, s') \leq d(s, s'). \quad (8)$$

Such constraints include (1) prior bias constraints (if any) of the form of $\mathfrak{h}(s) - \mathfrak{h}(s') \leq c_*(s, s')$; (2) span constraints of the form $\mathfrak{h}(s) - \mathfrak{h}(s') \leq c_0 := T^{1/5}$ spawning the span semi-ball $\{u : \mathrm{sp}\,(u) \leq T^{1/5}\}$; and (3) pair-wise constraints obtained by estimating bias differences in the style of Zhang and Ji [2019], Zhang and Xie [2023] that we further improve. We start by defining a bias difference estimator.

**Definition 1** (Bias difference estimator). *Given a pair of states $s \neq s'$, their sequence of commute times $(\tau_i^{s \leftrightarrow s'})_{i \geq 0}$ is defined by $\tau_{2i}^{s \leftrightarrow s'} := \inf\{t > \tau_{2i-1}^{s \leftrightarrow s'} : S_t = s\}$ and $\tau_{2i+1}^{s \leftrightarrow s'} := \inf\{t > \tau_{2i}^{s \leftrightarrow s'} : S_t = s'\}$ with the convention that $\tau_{-1}^{s \leftrightarrow s'} = -\infty$. The number of commutations up to time $t$ is $N_t(s \leftrightarrow s') := \inf\{i : \tau_i^{s \leftrightarrow s'} \leq t\}$, and $\hat{g}(t) := \frac{1}{t} \sum_{i=0}^{t-1} R_i$ is the empirical gain. The bias difference estimator at time $T$ is any quantity $c_T(s, s') \in \mathbf{R}$ such that:*

$$N_t(s \leftrightarrow s')c_T(s, s') = \sum_{t=0}^{N_T(s \leftrightarrow s')-1} (-1)^i \sum_{t=\tau_i^{s \leftrightarrow s'}}^{\tau_{i+1}^{s \leftrightarrow s'}-1} (\hat{g}(T) - R_t). \quad (9)$$

**Lemma 3.** *With probability $1 - 2\delta$, for all $T' \leq T$, we have*

$$N_{T'}(s \leftrightarrow s') \left| h^*(s) - h^*(s') - c_{T'}(s, s') \right| \leq 3\mathrm{sp}\,(h^*) + (1 + \mathrm{sp}\,(h^*)) \sqrt{8T \log(\tfrac{2}{\delta})} + 2 \sum_{t=0}^{T'-1} (g^* - R_t). \quad (10)$$

Lemma 3 says that the quality of the estimator $c_T(s, s')$ is directly linked to the number of observed commutes between $s$ and $s'$ as well as the regret. The idea is that if the algorithm makes many commutes between $s$ and $s'$ and if its regret is small, then the algorithm mostly takes optimal paths from $s$ to $s'$. The bound provided by Lemma 3 is not accessible to the learner however, because sp $(h^*)$ is unknown in general. To overcome this issue, sp $(h^*)$ is upper-bounded by $c_0 := T^{1/5}$. Overall, this leads to the design of the algorithm estimating the bias confidence region as specified in Algorithm 3.

| **Algorithm 3:** BiasEstimation$(\mathcal{F}_t, \mathcal{M}_t, \delta)$ | **Algorithm 4:** BiasProjection$(\mathcal{H}_t, u)$ |
|---|---|
| **Parameters:** History $\mathcal{F}_t$, model region $\mathcal{M}_t$, confidence $\delta > 0$ | **Parameters:** $\mathcal{H}_t$ a collection of linear constraints (8), $u \in \mathbf{R}^\mathcal{S}$ to project |
| 1: Estimate bias differences $c_t$ via (9); | 1: $v \leftarrow 0^\mathcal{S}$; |
| 2: Estimate optimistic gain $\tilde{g} \leftarrow \min_{k < K(t)} \mathfrak{g}_k$; | 2: **for** $s \in \mathcal{S}$ **do** |
| 3: Inner regret estimation $B_0 \leftarrow t\tilde{g} - \sum_{i=0}^{t-1} R_i$; | 3:     Using linear programming, compute: |
| 4: $\ell \leftarrow \sqrt{8T \log\left(\frac{2}{\delta}\right)}, c_0 \leftarrow T^{\frac{1}{5}}$; | 4:     $v(s) \leftarrow \sup \{w(s) : w \le u \text{ and } w \in \mathcal{H}_t\}$; |
| 5: Estimate the bias difference errors as: | 5: **end for** |
| $$d_t(s, s') \equiv \text{error}(c_t, s, s') := \frac{3c_0 + (1 + c_0)(1 + \ell) + 2B_0}{N_t(s \leftrightarrow s')}$$ | 6: **return** $v$. |
| 6: **return** $(c_t, \text{error}(c_t, -, -))$, (8) defines $\mathcal{H}'_t$. | |

Coupled with prior information and span constraints, the bias confidence region $\mathcal{H}_t$ is a polyhedron of the same kind as the one encountered in Zhang and Xie [2023]. When generated by constraints of the form (8), following [Zhang and Xie, 2023, Proposition 3], one can project onto $\mathcal{H}_t$ in polynomial time with Algorithm 4. Moreover, the resulting projection operator satisfies the prerequisites (**O1-4**) of Proposition 2, making PMEVI (Algorithm 2) well-behaved. See Appendix B.2 for proofs.

**Lemma 4.** *Assume that $\mathcal{H}$ is a set of $\mathfrak{h} \in \mathbf{R}^\mathcal{S}$ satisfying a system of equations of the form of (8). If $\mathcal{H}$ is non empty, then the operator $\Gamma u := \text{BiasProjection}(\mathcal{H}, u)$ (see Algorithm 4) is a projection on $\mathcal{H}$ and satisfies the properties (**O1-4**) defined in Proposition 2.*

### 3.3 Mitigation using finer bias dynamical error

The fact that $h^* \in \mathcal{H}_t$ with high probability is used in PMEVI-DT to restrict the search of EVI by reducing the dynamical bias error. This reduction is based on a empirical Bernstein inequality (see Lemma 38) applied to $(\hat{p}(s, a) - p(s, a))u$. Here, it gives that with probability $1 - \delta$, we have:

$$(\hat{p}_t(s, a) - p(s, a)) u \le \sqrt{\frac{2\mathbf{V}(\hat{p}_t(s, a), u) \log\left(\frac{3T}{\delta}\right)}{\max\{1, N_t(s, a)\}}} + \frac{3\text{sp}(u) \log\left(\frac{3T}{\delta}\right)}{\max\{1, N_t(s, a)\}} =: \beta_t(s, a, u) \qquad (11)$$

where $\mathbf{V}(\hat{p}_t(s, a), u)$ is the variance of $u$ under the probability vector $\hat{p}_t(s, a)$. More specifically, if $q$ is a probability on $\mathcal{S}$ and $q \in \mathbf{R}^\mathcal{S}$, we set $\mathbf{V}(q, u) := \sum_s q(s)(u(s) - q \cdot u)^2$. In (11), $u \in \mathbf{R}^\mathcal{S}$, $(s, a) \in \mathcal{X}$ and $T \ge 1$ are fixed. Once is tempted to use (11) directly to mitigate the extended Bellman operator, but the resulting operator is ill-behaved because it loses monotony. This issue is avoided by changing $\beta_t(s, a, u)$ to $\max_{u \in \mathcal{H}_t} \beta_t(s, a, u)$ in (11). The resulting inequality is *not* guaranteed to hold simultaneously for all $u \in \mathcal{H}_t$ and with high probability; However, it is guaranteed to hold with high probability for $u = h^*$, which will be enough.

The variance maximization problem $\max_{u \in \mathcal{H}_t} \beta_t(s, a, u)$ is a *convex maximization problem* with linear constraints. Even in very simple settings, such optimization problems are NP-hard Pardalos and Schnitger [1988] hence computing $\max_{u \in \mathcal{H}_t} \beta_t(s, a, u)$ is not reasonable in general. Thankfully, this value can be upper-bounded by a tractable quantity that is enough in the regret analysis. The mitigation $\beta_t$ used by PMEVI-DT is provided by Algorithm 5. See Lemma 12 and Appendix A.2.2 for details.

## 4 Regret guarantees

Theorem 5 thereafter shows that PMEVI-DT has minimax optimal regret under regularity assumptions on the used confidence region $\mathcal{M}_t$. Assumption 1 asserts that the confidence region holds uniformly with high probability. Assumption 2 asserts that the reward confidence region is sub-Weissman (see

---

**Algorithm 5:** `VarianceApproximation(`$\mathcal{H}'_t, \mathcal{F}_t$`)`

---

**Parameters:** Bias region $\mathcal{H}'_t$, history $\mathcal{F}_t$

1: Extract constraints $(c, \text{error}(c, -, -)) \leftarrow \mathcal{H}'_t$;
2: Set $c_0 \leftarrow T^{\frac{1}{5}}$;
3: Pick a reference point $h_0 \leftarrow$ `BiasProjection(`$\mathcal{H}_t, c(-, s_0)$`)`;
4: **for** $(s, a) \in \mathcal{X}$ **do**
5: $\quad \rho \leftarrow \log\left(\frac{SAT}{\delta}\right) / \max\{1, N_t(s, a)\}$;
6: $\quad \text{var}(s, a) \leftarrow \mathbf{V}(\hat{p}_t(s, a), h_0) + 8c_0 \sum_{s' \in \mathcal{S}} \hat{p}_t(s'|s, a)c(s', s)$;
7: $\quad \beta_t(s, a) \leftarrow \sqrt{2\text{var}(s, a)\rho} + 3c_0\rho$ or $+\infty$ if $N_t(s, a) = 0$;
8: **end for**
9: **return** $\beta_t$.

---

Lemma 35) and Assumption 3 assumes that the model confidence region makes sure that `EVI` (6) converges in the first place. Assumption 4 asserts that the prior bias region is correct.

**Assumption 1.** With probability $1 - \delta$, we have $M \in \bigcap_{k=1}^{K(T)} \mathcal{M}_{t_k}$.

**Assumption 2.** There exists a constant $C > 0$ such that for all $(s, a) \in \mathcal{S}$, for all $t \leq T$, we have:
$$\mathcal{R}_t(s, a) \subseteq \left\{ \tilde{r}(s, a) \in \mathcal{R}(s, a) : N_t(s, a) \|\hat{r}_t(s, a) - \tilde{r}(s, a)\|_1^2 \leq C \log\left(\frac{2SA(1+N_t(s,a))}{\delta}\right) \right\}.$$

**Assumption 3.** For $t \geq 0$, $\mathcal{M}_t$ is a $(s, a)$-rectangular convex region and $\mathcal{L}_t^n u$ converges a fix-point.

**Assumption 4.** The prior bias region $\mathcal{H}_*$ contains $h^*(M)$ and is generated by constraints of the form:
$$\forall s \neq s', \quad \mathfrak{h}(s) - \mathfrak{h}(s') \leq c_*(s, s')$$
with $c_*(s, s') \in [-\infty, \infty]$ (possibly infinite).

Refer to Appendix A.2 for the feasibility of Assumption 1, Appendix A.2.3 for Assumption 2, and Appendix A.3 for Assumption 3.

**Theorem 5** (Main result). *Let $c > 0$. Assume that* `PMEVI-DT` *runs with a confidence region system $t \mapsto \mathcal{M}_t$ that guarantees Assumptions 1-3. If $T \geq c^5$, then for every weakly communicating model with $\text{sp}(h^*) \leq c$ and such that Assumption 4 is satisfied ($h^* \in \mathcal{H}_*$),* `PMEVI-DT` *achieves regret:*
$$\mathrm{O}\left(\sqrt{cSAT \log\left(\frac{SAT}{\delta}\right)}\right) + \mathrm{O}\left(cS^{\frac{5}{2}}A^{\frac{3}{2}}T^{\frac{9}{20}} \log^2\left(\frac{SAT}{\delta}\right)\right)$$
*with probability $1 - 26\delta$, and in expectation if $\delta < \sqrt{1/T}$. Moreover, if* `PMEVI-DT` *runs with the same confidence regions that* `UCRL2` *Auer et al. [2009], then it enjoys a time complexity $\mathrm{O}(DS^3AT)$.*

To have a completely prior-less algorithm, pick $\mathcal{H}_* = \mathbf{R}^{\mathcal{S}}$. The proof of Theorem 5 is tedious and its details are deferred to appendix. We will focus here on the main ideas.

**Notations.** At episode $k$, the played policy is denoted $\pi_k$. As a greedy response to $\mathfrak{h}_k$, by Proposition 2 (3), there exists $\tilde{r}_k(s) \leq \sup \mathcal{R}_{t_k}(s, \pi_k(s))$ and $\tilde{P}_k(s) \in \mathcal{P}_{t_k}(s, \pi(x))$ such that $\mathfrak{h}_k + \mathfrak{g}_k = \tilde{r}_k + \tilde{P}_k\mathfrak{h}_k$. The reward-kernel pair $\tilde{M}_k = (\tilde{r}_k, \tilde{P}_k)$ is referred to as the *optimistic model* of $\pi_k$. We write $P_k := P_{\pi_k}(M)$ the true kernel and $\hat{P}_k := P_{\pi_k}(\hat{M}_{t_k})$ the empirical kernel. Likewise, we define the reward functions $r_k$ and $\hat{r}_k$. The optimistic gain and bias satisfy $\mathfrak{g}_k = g(\pi_k, \tilde{M}_k)$ and $\mathfrak{h}_k = h(\pi_k, \tilde{M}_k)$. We further denote $c_0 = T^{\frac{1}{5}}$.

The regret is first decomposed episodically with $\text{Reg}(T) = \sum_k \sum_{t=t_k}^{t_{k+1}-1}(g^* - R_t)$. The first step goes back to the analysis of `UCRL2` Auer et al. [2009], and consists in upper-bounding the regret of the episode $k$ with optimistic quantities that are exclusive to that episode.

**Lemma 6** (Reward optimism). *With probabililty $1 - 6\delta$, we have:*
$$\text{Reg}(T) \leq \sum_k \sum_{t=t_k}^{t_{k+1}-1}(\mathfrak{g}_k - R_t) \leq \sum_k \sum_{t=t_k}^{t_{k+1}-1}(\mathfrak{g}_k - \tilde{r}_k(X_t)) + \mathrm{O}\left(\sqrt{SAT \log\left(\frac{T}{\delta}\right)}\right). \quad (12)$$

We introduce the two optimistic regrets $B(T) := \sum_k \sum_{t=t_k}^{t_{k+1}-1}(\mathfrak{g}_k - R_t)$ and $\tilde{B}(T) := \sum_k \sum_{t=t_k}^{t_{k+1}-1}(\mathfrak{g}_k - \tilde{r}_k(X_t))$. Rewriting the summand $\mathfrak{g}_k - \tilde{r}_k(X_t)$ using the Poisson equation $\mathfrak{h}_k + \mathfrak{g}_k = \tilde{r}_k + \tilde{P}_k\mathfrak{h}_k$, we get:
$$\tilde{B}(T) = \sum_k \sum_{t=t_k}^{t_{k+1}-1}(\tilde{p}_k(S_t) - e_{S_t})\mathfrak{h}_k.$$

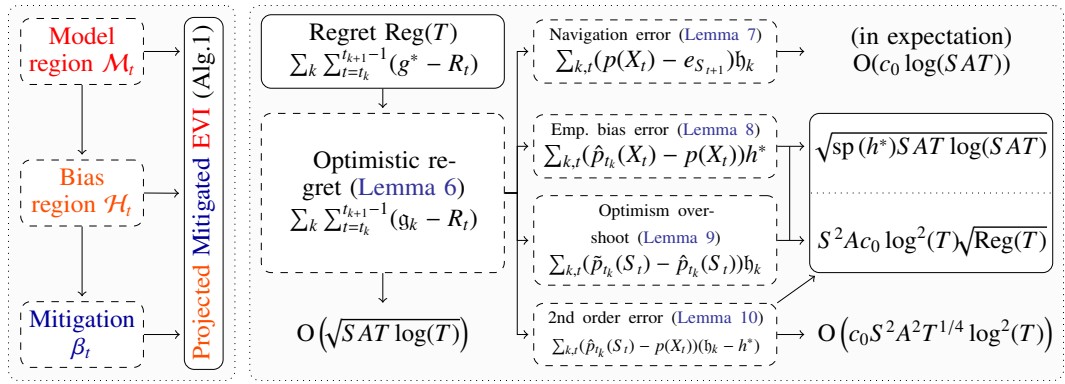

Figure 1: An overview of `PMEVI-DT` and its regret analysis. In the above, $\mathfrak{g}_k$ and $\mathfrak{h}_k$ are the optimistic gain and bias functions produced by `PMEVI` (see Algorithm 2) at episode $k$, and $\hat{p}_{t_k}$ and $\tilde{p}_{t_k}$ are respectively the empirical and optimistic kernel models at episode $k$.

The analysis proceeds by decomposing the above expression of $\tilde{B}(T)$ in the style of Zhang and Ji [2019]. We write $\sum_{t=t_k}^{t_{k+1}-1}(\tilde{p}_k(S_t) - e_{S_t})\mathfrak{h}_k$ as:

$$\sum_{t=t_k}^{t_{k+1}-1}\left(\underbrace{(p_k(S_t) - e_{S_t})\mathfrak{h}_k}_{\text{navigation error }(1k)} + \underbrace{(\hat{p}_k(S_t) - p_k(S_t))h^*}_{\text{empirical bias error }(2k)} + \underbrace{(\tilde{p}_k(S_t) - \hat{p}_k(S_t))\mathfrak{h}_k}_{\text{optimistic overshoot }(3k)} + \underbrace{(\hat{p}_k(S_t) - p_k(S_t))(\mathfrak{h}_k - h^*)}_{\text{second order error }(4k)}\right)$$

Each error term is bounded separately. Below, we denote $\mathbf{V}(q, u) := \sum_s q(s)(u(s) - q \cdot u)^2$.

**Lemma 7** (Navigation error). *With probability $1 - 7\delta$, the navigation error is bounded by:*

$$\sum_k \sum_{t=t_k}^{t_{k+1}-1}(p_k(S_t) - e_{S_t})\mathfrak{h}_k \le \sqrt{2\sum_{t=0}^{T-1}\mathbf{V}(p(X_t), h^*)\log\left(\frac{T}{\delta}\right)} + 2SA^{\frac{1}{2}}\sqrt{3B(T)}\log\left(\frac{T}{\delta}\right) + \widetilde{O}\left(T^{\frac{7}{20}}\right).$$

**Lemma 8** (Empirical bias error). *With probability $1 - \delta$, the empirical bias error is bounded by:*

$$\sum_k \sum_{t=t_k}^{t_{k+1}-1}(\hat{p}_k(S_t) - p_k(S_t))h^* \le 4\sqrt{SA\sum_{t=0}^{T-1}\mathbf{V}(p(X_t), h^*)\log\left(\frac{SAT}{\delta}\right)} + O\left(\log^2(T)\right).$$

**Lemma 9** (Optimistic overshoot). *With probability $1 - 6\delta$, the optimistic overshoot is bounded by:*

$$\sum_k \sum_{t=t_k}^{t_{k+1}-1}(\tilde{p}_k(S_t) - \hat{p}_k(S_t))\mathfrak{h}_k \le \left\{\begin{array}{l}4\sqrt{2SA\sum_{t=0}^{T-1}\mathbf{V}(p(X_t), h^*)\log\left(\frac{SAT}{\delta}\right)} \\ +8(1 + c_0)S^{\frac{3}{2}}A\log^{\frac{3}{2}}\left(\frac{SAT}{\delta}\right)\sqrt{B(T)} + \widetilde{O}\left(T^{\frac{1}{4}}\right)\end{array}\right\}.$$

**Lemma 10** (Second order error). *With probability $1 - 6\delta$, the second order error is bounded by:*

$$\sum_k \sum_{t=t_k}^{t_{k+1}-1}(\hat{p}_k(S_t) - p_k(S_t))(\mathfrak{h}_k - h^*) \le 16S^2A(1 + c_0)\log^{\frac{1}{2}}\left(\frac{S^2AT}{\delta}\right)\sqrt{2B(T)} + \widetilde{O}\left(T^{\frac{1}{4}}\right).$$

We see that the empirical bias error (Lemma 8) and the optimistic overshoot (Lemma 9) both involve the sum of variances $\sum_{t=0}^{T-1}\mathbf{V}(p(X_t), h^*)$, which is shown in Lemma 29 to be of order $\text{sp}(h^*)\text{sp}(r)T + \sum_{t=0}^{T-1}\Delta^*(X_t)$. The pseudo-regret term $\sum_{t=0}^{T-1}\Delta^*(X_t)$ is bounded with the regret using Corollary 31, then by $B(T)$. With high probability, we obtain an equation of the form:

$$B(T) \le C\sqrt{(1 + \text{sp}(h^*))SAT\log\left(\frac{T}{\delta}\right)} + CS^2A(1 + c_0)\log^2(T)\sqrt{B(T)} + \tilde{O}\left(T^{\frac{1}{4}}\right)$$

where $C$ is a constant. Setting $\alpha := CS^2A(1 + c_0)\log^2(T)$ and $\beta := C\sqrt{(1 + \text{sp}(h^*))SAT\log(T/\delta)} + \tilde{O}(T^{1/4})$, the above equation is of the form $B(T) \le \beta + \alpha\sqrt{B(T)}$. Solving in $B(T)$, we find $B(T) \le \beta + 2\sqrt{\alpha\beta} + \alpha^2$. The dominant term is $\beta$, hence we readily obtain:

$$B(T) \le C\sqrt{(1 + \text{sp}(h^*))\text{sp}(r)SAT\log\left(\frac{T}{\delta}\right)} + \widetilde{O}\left(\text{sp}(h^*)\text{sp}(r)S^{\frac{5}{2}}A^{\frac{3}{2}}(1 + c_0)T^{\frac{1}{4}}\right). \qquad (13)$$

Since $c_0 = o(T^{\frac{1}{4}})$, we conclude that $B(T) = O\left(\sqrt{\text{sp}(h^*)SAT\log(T/\delta)}\right)$, ending the proof.

# 5 Experimental illustrations

To get a grasp of how `PMEVI-DT` behaves in practice, we provide in Fig. 2 a first round of illustrative experiments. In both, the environment is a river-swim which is a model known to be hard to learn despite its size, with high diameter and bias span, see Appendix D for the model's description.

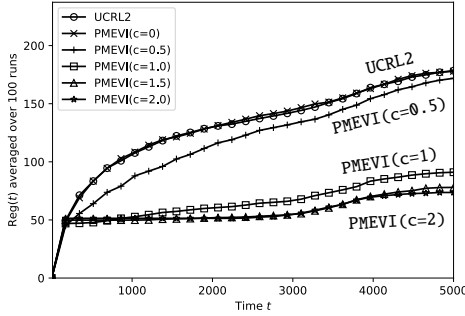 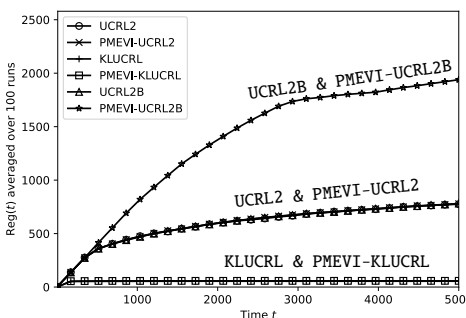

Figure 2: (**To the left**) Running `UCRL2` and `PMEVI-DT` with the same confidence region than `UCRL2` on a 3-state river-swim. `PMEVI-DT` is run with prior knowledge $h^*(s_1) \le h^*(s_2) - c \le h^*(s_3) - 2c$ for $c \in \{0, 0.5, 1, 1.5, 2\}$. (**To the right**) Running a few algorithms of the literature on 5-state river-swim and comparing their average regret against their `PMEVI` variants, obtained by changing calls to the `EVI` sub-routine to calls to `PMEVI`.

On the first experiment, we observe that `PMEVI` can exploit prior bias knowledge effectively and drastically improve the regret performance, depending on the quality of the prior region.

On the second experiment however, we observe that without prior knowledge, `PMEVI` has nearly the same regret performance that its `EVI` counterparts, meaning that the bias confidence region is too large to effectively improve the regret performance. This observation is first to be taken with caution. Indeed, the regret that is being estimated above is model specific, hence is not an estimate of the minimax regret — This being said, it undoubtedly shows that the bias confidence region is ineffective and this can be explained as follows. On experiments, we see that most of the regret is due to the early phase of the learning process, where proper bias information is nearly impossible to get. Indeed, the regret is still growing linearly, so no bias information can be inferred. But in addition, this "bad" early data pollutes the bias estimator for a long duration. In other words, while the theoretical regret guarantees of `PMEVI-DT` are better than its `EVI` analogues, there is room to improve the bias estimation mechanism and the practical performance.

# 6 Conclusion

In this work, we have shown that regret guarantees of order $\sqrt{\mathrm{sp}(h^*)SAT\log(T)}$ can be achieved for weakly communicating MDPs without prior knowledge, nor exponential computational cost. In particular, regret guarantees can scale with the bias span rather than the diameter without prior knowledge. This is in opposition to the recent results that the sample complexity cannot be bounded in term of bias span without prior knowledge for average reward MDPs Tuynman et al. [2024], Wang et al. [2024], Zurek and Chen [2024a,b]. This difference lies in the fact $(\epsilon, \delta)$-PAC algorithm must produce a policy $\pi_\tau$ after $\tau$ learning steps where $\tau$ is a stopping time, with $\mathbf{P}(g^{\pi_\tau} \le g^* - \epsilon) \le \delta$. Implicitly, these algorithms must hereby *certify* that the output policy is approximately optimal. In opposition, regret robust algorithms have no need to assess that deployed policies are indeed optimal.

In the end, the regret advantages of `PMEVI-DT` over pure `EVI`-based methods remain theoretical, and the experimental shortcomings displayed in Section 5 leave a few opportunities for future work. Can bias information be inferred more efficiently? Or, do the experiments indicate that the regret analysis of `EVI`-based methods may be drastically improved?

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

# A    Construction of `PMEVI-DT`

This section provides the technical details required to understand the design of `PMEVI-DT` in Section 3. We further discuss the assumptions 1-4 appearing in Theorem 5 and provide sufficient conditions so that they are met.

## A.1    Proof of Lemma 3, estimation of the bias error

Fix $s, s' \in \mathcal{S}$. We denote $\alpha_T := N_T(s \leftrightarrow s')(h^*(s) - h^*(s') - c_T(s, s'))$. We will start by considering the better estimator $c'_T(s, s')$ that satisfies the same equation (9) than $c_T(s, s')$ but with $\hat{g}(T)$ changed to $g^*$, readily:

$$N_t(s \leftrightarrow s')c'_T(s, s') = \sum_{t=0}^{N_T(s\leftrightarrow s')-1} (-1)^i \sum_{t=\tau_i^{s\leftrightarrow s'}}^{\tau_{i+1}^{s\leftrightarrow s'}-1} (g^* - R_t).$$

To avoid a typographical clutter, we write $\tau_i$ instead of $\tau_i^{s\leftrightarrow s'}$ in the remaining of the proof and we write $\alpha'_T := N_T(s \leftrightarrow s')(h^*(s) - h^*(s') - c'_T(s, s'))$.

(**STEP 1**) We start by relating the two estimators. Intuitively, $\hat{g}(T)$ is a good estimator for $g^*$ when the regret is small. Recall that $\hat{g}(T) := \frac{1}{T} \sum_{t=0}^{T-1} R_t$, hence:

$$\sum_{t=0}^{T-1} |\hat{g}(T) - g^*| = \left| \sum_{t=0}^{T-1} (R_t - g^*) \right| = |\text{Reg}(T)|.$$

Therefore,

$$|\alpha_T| \le |\alpha'_T| + |\alpha_T - \alpha'_T| \le |\alpha'_T| + \sum_{t=0}^{T-1} |\hat{g}(T) - g^*| \le |\alpha'_T| + |\text{Reg}(T)|.$$

We are left with upper-bounding $|\alpha'_T|$.

(**STEP 2**) If $i$ is even, then $S_{\tau_i}$ and $S_{\tau_{i+1}} = s'$; otherwise $S_{\tau_i} = s'$ and $S_{\tau_{i+1}} = s$. In both cases, we have $h^*(S_{\tau_{i+1}}) - h^*(S_{\tau_i}) = (-1)^i(h^*(s') - h^*(s))$. Therefore, using Bellman's equation, the quantity $A := \sum_{t=\tau_i}^{\tau_{i+1}-1}(g^* - R_t)$ satisfies:

$$A = \sum_{t=\tau_i}^{\tau_{i+1}-1} (p(X_t) - e_{S_t}) h^* + \sum_{t=\tau_i}^{\tau_{i+1}-1} (r(X_t) - R_t) + \sum_{t=\tau_i}^{\tau_{i+1}-1} \Delta^*(X_t)$$

$$= \sum_{t=\tau_i}^{\tau_{i+1}-1} (e_{S_{t+1}} - e_{S_t}) h^* + \sum_{t=\tau_i}^{\tau_{i+1}-1} (p(X_t) - e_{S_{t+1}}) h^* + \sum_{t=\tau_i}^{\tau_{i+1}-1} (r(X_t) - R_t) + \sum_{t=\tau_i}^{\tau_{i+1}-1} \Delta^*(X_t)$$

$$= (-1)^i(h^*(s') - h^*(s)) + \sum_{t=\tau_i}^{\tau_{i+1}-1} (p(X_t) - e_{S_{t+1}}) h^* + \sum_{t=\tau_i}^{\tau_{i+1}-1} (r(X_t) - R_t) + \sum_{t=\tau_i}^{\tau_{i+1}-1} \Delta^*(X_t).$$

Multiplying by $(-1)^i$ and rearranging, $h^*(s') - h^*(s) + (-1)^{i+1} \sum_{t=\tau_i}^{\tau_{i+1}-1}(g^* - R_t)$ appears to be equal to:

$$(-1)^{i+1} \left( \sum_{t=\tau_i}^{\tau_{i+1}-1} ((p(X_t) - e_{S_{t+1}}) h^* + r(X_t) - R_t) + \sum_{t=\tau_i}^{\tau_{i+1}-1} \Delta^*(X_t) \right).$$

Proceed by summing over $i$. By triangular inequality, we obtain:

$$|\alpha'_T| \le \left| \sum_{i=0}^{N_T(s\leftrightarrow s')-1} \sum_{t=\tau_i}^{\tau_{i+1}-1} (-1)^{i+1} ((p(X_t) - e_{S_{t+1}}) h^* + r(X_t) - R_t) \right| + \sum_{i=0}^{N_T(s\leftrightarrow s')-1} \sum_{t=\tau_i}^{\tau_{i+1}-1} \Delta^*(X_t).$$

Because all Bellman gaps $\Delta^*$ are non-negative, the second term is upper-bounded by the pseudo-regret $\sum_{t=0}^{T-1} \Delta^*(X_t)$. The first term is a martingale, and the martingale difference sequence $(-1)^{i+1}((p(X_t) - e_{S_{t+1}})h^* + r(X_t) - R_t$ has span at most $\text{sp}(h^*) + 1$ since rewards are supported in $[0, 1]$. Although the number of involved terms is random, it is upper-bounded by $T$, hence by the maximal version of Azuma-Hoeffding's inequality (Lemma 32), we have that with probability at least $1 - \delta$ and uniformly for $T' \le T$,

$$\left| \sum_{i=0}^{N_{T'}(s\leftrightarrow s')-1} \sum_{t=\tau_i}^{\tau_{i+1}-1} (-1)^{i+1} ((p(X_t) - e_{S_{t+1}}) h^* + r(X_t) - R_t) \right| \le (1 + \text{sp}(h^*)) \sqrt{\frac{1}{2} T \log\left(\frac{2}{\delta}\right)}.$$

(**STEP 3**) We conclude that with probability $1 - \delta$, for all $T' \le T$,

$$\alpha_{T'} \le (1 + \text{sp}(h^*)) \sqrt{\frac{1}{2} T \log\left(\frac{2}{\delta}\right)} + \sum_{t=0}^{T'-1} \Delta^*(X_t) + |\text{Reg}(T')|.$$

We are left with relating both $\sum_{t=0}^{T'-1} \Delta^*(X_t)$ and $\left|\text{Reg}(T')\right|$ to $\sum_{t=0}^{T'-1}(\tilde{g}-R_t)$. Using the Bellman equation again, we find that:

$$\left|\sum_{t=0}^{T'-1}(g^* - R_t - \Delta^*(X_t))\right| \leq |h^*(S_0) - h^*(S_{T'})| + \left|\sum_{t=0}^{T'-1}\left((p(X_t) - e_{S_{t+1}})h^* + (r(X_t) - R_t)\right)\right|$$

$$\leq \text{sp}(h^*) + (1 + \text{sp}(h^*))\sqrt{\tfrac{1}{2}T\log\left(\tfrac{2}{\delta}\right)}$$

where the last inequality holds with probability $1 - \delta$ uniformly over $T' \leq T$ by Azuma-Hoeffding's inequality again (Lemma 32). Remark that if $y - z \leq x \leq y + z$, then $|x| \leq |y| + |z|$, hence we conclude that with probability $1 - \delta$, for all $T' \leq T$:

$$\sum_{t=0}^{T'-1}\Delta^*(X_t) + \left|\text{Reg}(T')\right| \leq 2\sum_{t=0}^{T'-1}\Delta^*(X_t) + (1 + \text{sp}(h^*))\sqrt{\tfrac{1}{2}T\log\left(\tfrac{2}{\delta}\right)} + \text{sp}(h^*)$$

$$\leq 2\sum_{t=0}^{T'-1}(g^* - R_t) + 3(1 + \text{sp}(h^*))\sqrt{\tfrac{1}{2}T\log\left(\tfrac{2}{\delta}\right)} + 3\,\text{sp}(h^*)$$

$$\leq 2\sum_{t=0}^{T'-1}(\tilde{g} - R_t) + 3(1 + \text{sp}(h^*))\sqrt{\tfrac{1}{2}T\log\left(\tfrac{2}{\delta}\right)} + 3\,\text{sp}(h^*)$$

where the last inequality invokes $\tilde{g} \geq g^*$. We conclude that, with probability $1 - 2\delta$, for all $T' \leq T$, we have:

$$N_{T'}(s \leftrightarrow s')(h^*(s) - h^*(s') - c_{T'}(s, s')) \leq 3\,\text{sp}(h^*) + (1 + \text{sp}(h^*))\sqrt{8T\log\left(\tfrac{2}{\delta}\right)} + \sum_{t=0}^{T'-1}(\tilde{g} - R_t).$$

This concludes the proof. $\qquad\qquad\square$

## A.2  The confidence region of `PMEVI-DT`

The algorithm `PMEVI-DT` can be instantiated with a large panel of possibilities, depending on the type of confidence region one is willing to use for rewards and kernels. In this work, we allow for four types of confidence regions, described below. For conciseness, $q \in \{r, p\}$ is a symbolic letter that can be a reward or a kernel and we denote $Q_t(s, a)$ the confidence region for $q(s, a)$ at time $t$. If $q = r$, then $\dim(q) = 2$ (Bernoulli rewards) with $Q(s, a) = [0, 1]$; and if $q = p$, then $\dim(q) = S$ with $Q(s, a) = \mathcal{P}(\mathcal{S})$.

(**C1**) *Azuma-Hoeffding* or *Weissman* type confidence regions, with $Q_t(s, a)$ taken as:

$$\left\{\tilde{q}(s, a) \in Q(s, a) : N_t(s, a)\|\hat{q}_t(s, a) - \tilde{q}(s, a)\|_1^2 \leq \dim(q)\log\left(\tfrac{2SA(1+N_t(s,a))}{\delta}\right)\right\}.$$

(**C2**) *Empirical Bernstein* type confidence regions, with $Q_t(s, a)$ taken as:

$$\left\{\tilde{q}(s, a) \in Q(s, a) : \forall i, |\hat{q}_t(i|s, a) - \tilde{q}(i|s, a)| \leq \sqrt{\frac{2\mathbf{V}(\hat{q}_t(i|s,a))\log\left(\frac{2\dim(q)SAT}{\delta}\right)}{N_t(s,a)}} + \frac{3\log\left(\frac{2\dim(q)SAT}{\delta}\right)}{N_t(s,a)}\right\}.$$

with the convention that $x/0 = +\infty$ for $x > 0$.

(**C3**) *Empirical likelihood* type confidence regions, with $Q_t(s, a)$ taken as:

$$\left\{\tilde{q}(s, a) \in Q(s, a) : N_t(s, a)\,\text{KL}(\hat{q}_t(s, a)\|\tilde{q}(s, a)) \leq \log\left(\tfrac{2SA}{\delta}\right) + (\dim(q) - 1)\log\left(e\left(1 + \tfrac{N_t(s,a)}{\dim q - 1}\right)\right)\right\}.$$

(**C4**) *Trivial* confidence region with $Q_t(s, a) = Q(s, a)$.

A few remarks are in order. When rewards are not Bernoulli, only the confidence regions (**C1**) and (**C4**) are elligible among the above. Then, Weissman's inequality must be changed to Azuma's inequality for $\sigma$-sub-Gaussian random variables, see Lemma 34. Since rewards are supported in $[0, 1]$, Hoeffding's Lemma guarantees that reward distributions are $\sigma$-sub-Gaussian with $\sigma = \tfrac{1}{2}$.

### A.2.1 Correctness of the model confidence region $\mathcal{M}_t$ and Assumption 1

The confidence regions $Q_t(s, a)$ described with (**C1-4**) are tuned so that the following result holds:

**Lemma 11.** *Assume that, for all $q \in \{r, p\}$ and $(s, a) \in X$, we choose $Q_t(s, a)$ among (**C1-4**). Then Assumption 1 holds. More specifically, the region of models $\mathcal{M}_t := \prod_{s,a}(\mathcal{R}_t(s, a) \times \mathcal{P}_t(s, a))$ satisfies $\mathbf{P}(\exists t \leq T : M \notin \mathcal{M}_t) \leq \delta$.*

*Proof.* We show that, for all $q \in \{r, q\}$ and $(s, a) \in X$, if $Q_t(s, a)$ is chosen amoung (**C1-4**), then

$$\mathbf{P}\left(\exists t \leq T : q(s, a) \notin Q_t(s, a)\right) \leq \delta.$$

If $Q_t(s, a)$ is chosen with (**C1**), this is a direct application of Lemma 35; with (**C2**), this is Lemma 36; with (**C3**), this is Lemma 37; and with (**C4**) this is by definition. □

### A.2.2 Simultaneous correctness of bias confidence region $\mathcal{H}_t$, mitigation $\beta_t$ and optimism

In this section, we show that if Assumption 1 holds, then the bias confidence region constructed by `PMEVI-DT` is correct with high probability, and that the mitigation is not too strong. Recall that $(\mathfrak{g}_k, \mathfrak{h}_k)$ are the optimistic gain and bias of the policy deployed in episode $k$ (see Algorithm 1). In particular, we have $\mathfrak{g}_k = \mathfrak{L}_{t_k}\mathfrak{h}_k - \mathfrak{h}_k$ with $\mathfrak{h}_k \in \mathcal{H}_{t_k}$. We start by a result on the deviation of the variance, which is what the variance approximation Algorithm 5 is based on. Recall that the bias confidence region $\mathcal{H}_t$ is obtained as the collection of constraints:

(1) prior constraints (if any) $\mathfrak{h}(s) - \mathfrak{h}(s') \leq c_*(s, s')$;

(2) span constraints $\mathfrak{h}(s) - \mathfrak{h}(s') \leq c_0 := T^{1/5}$;

(3) dynamically inferred constraints $|\mathfrak{h}(s) - \mathfrak{h}(s') - c_t(s', s)| \leq \mathrm{error}(c_t, s', s)$ (see Algorithm 3).

We start with the technical result that is behind the variance approximation Algorithm 5.

**Lemma 12.** *Let $u, v \in \mathcal{H}_t$ and fix $p$ a probability distribution on $S$. Then for all $s \in S$,*

$$\mathbf{V}(p, u) \leq \mathbf{V}(p, v) + 8c_0 \sum_{s' \in S} p(s')\, \mathrm{error}(c_t, s', s).$$

*Proof.* We start by establishing the following result: If $p$ is a probability distribution on $S$ and $u, v \in \mathbf{R}^S$, we have:

$$\mathbf{V}(p, u) \leq \mathbf{V}(p, v) + 2\,(p \cdot |u - v|)\max(u + v) \tag{14}$$

where $\cdot$ is the dot product, $u^2$ the Hadamard product $uu$ and $|u|$ the vector whose entry $s$ is $|u(s)|$. (14) is obtained with a straight forward computation:

$$
\begin{aligned}
\mathbf{V}(p, u) - \mathbf{V}(p, v) &= p \cdot (u^2 - v^2) + (p \cdot v)^2 - (p \cdot u)^2 \\
&= p \cdot ((u - v)(u + v)) + (p \cdot (u - v))(p \cdot (u + v)) \\
&\leq p \cdot (|u - v|(u + v)) + (p \cdot |u - v|)(p \cdot |u + v|) \\
&\leq 2(p \cdot |u - v|)\max(u + v).
\end{aligned}
$$

Observe that $v$ can be changed to $v + \lambda e$, where $e$ is the vector full of ones, without changing the result. The same goes for $u$. We now move to the proof of the main statement. First, translate $u$ and $v$ such that $u(s) = v(s) = 0$. Then, we have:

$$
\begin{aligned}
p \cdot (u - v) &= \sum_{s' \in S} p(s')\left|u(s') - u(s) - c_t(s', s) + v(s) - v(s') + c_t(s', s)\right| \\
&\leq \sum_{s' \in S} p(s')\left(\left|u(s') - u(s) - c_t(s', s)\right| + \left|v(s') - v(s) - c_t(s', s)\right|\right) \\
&\leq 2 \sum_{s' \in S} p(s')\, \mathrm{error}(c_t, s', s).
\end{aligned}
$$

Conclude using that $\max(u + v) \leq \max(u) + \max(v) + 2c_0$ for $u, v \in \mathcal{H}$ such that $u(s) = v(s) = 0$. □

**Lemma 13.** *Assume that Assumption 1 holds and that $c_0 \geq \mathrm{sp}(h^*)$. Then, with probability $1 - 4\delta$, for all $k \leq K(T)$, (1) $\mathfrak{g}_k \geq g^*$ and (2) $h^* \in \mathcal{H}_{t_k}$ and (3) for all $(s, a)$, $(\hat{p}_{t_k}(s, a) - p(s, a))h^* \leq \beta_{t_k}(s, a)$.*

*Proof.* Let $E_1$ the event $(\forall k \leq K(T), M \in \mathcal{M}_{t_k})$. Let $E_2$ the event stating that, for all $T' \leq T$,

$$N_{T'}(s \leftrightarrow s') \left| h^*(s) - h^*(s') - c_{T'}(s,s') \right| \leq 3\mathrm{sp}(h^*) + (1 + \mathrm{sp}(h^*)) \sqrt{8T \log(\tfrac{2}{\delta})} + 2 \sum_{t=0}^{T'-1} (g^* - R_t),$$

and let $E_3$ the event stating that, for all $T' \leq T$ and for all $(s,a) \in \mathcal{X}$, we have:

$$(\hat{p}_{T'}(s,a) - p(s,a)) h^* \leq \sqrt{\frac{2\mathbf{V}(\hat{p}_{T'}(s,a),h^*) \log\left(\frac{SAT}{\delta}\right)}{N_{T'}(s,a)}} + \frac{3\mathrm{sp}(h^*) \log\left(\frac{SAT}{\delta}\right)}{N_{T'}(s,a)}.$$

By [Assumption 1], we have $\mathbf{P}(E_1) \geq 1 - \delta$. By [Lemma 3], we have $\mathbf{P}(E_2) \geq 1 - 2\delta$ and by [Lemma 36], we have $\mathbf{P}(E_3) \geq 1 - \delta$, so $\mathbf{P}(E_1 \cap E_2 \cap E_3) \geq 1 - 4\delta$. We prove by induction on $k \leq K(T)$ that, on $E_1 \cap E_2 \cap E_3$, (1) $\mathfrak{g}_k \geq g^*$, (2) $h^* \in \mathcal{H}_{t_k}$ (3) and for all $(s,a)$, $(\hat{p}_{t_k}(s,a) - p(s,a)) h^* \leq \beta_{t_k}(s,a)$, where $\mathfrak{g}_k$ is the optimistic gain of the policy deployed at episode $k$.

It is obvious for $k = 0$. Indeed, $N_0(s \leftrightarrow s') = 0$ for all $s, s'$ hence $c_0(s,s') = c_0 \geq \mathrm{sp}(h^*)$. Therefore,

$$\mathcal{H}_0 \supseteq \left\{ \mathfrak{h} \in \mathbf{R}^{\mathcal{S}} : \mathrm{sp}(\mathfrak{h}) \leq c_0 \right\} \supseteq \left\{ \mathfrak{h} \in \mathbf{R}^{\mathcal{S}} : \mathrm{sp}(\mathfrak{h}) \leq \mathrm{sp}(h^*) \right\}$$

so contains $h^*$, proving (2). Moreover, since $N_0(s,a) = 0$, we have $\beta_0(s,a) = +\infty$, proving (3). Finally, since $M \in \mathcal{M}_0$ on $E_1$, by the statement (2) of [Proposition 2], we have $\mathfrak{g}_k \geq g^*$, hence proving (1).

Now assume that $k \geq 1$. By induction $\mathfrak{g}_\ell \geq g^*$ for all $\ell < k$, so on $E_2$ we have:

$$N_{t_k}(s \leftrightarrow s') \left| h^*(s) - h^*(s') - c_{t_k}(s,s') \right| \leq 3\mathrm{sp}(h^*) + (1 + \mathrm{sp}(h^*)) \sqrt{8T \log(\tfrac{2}{\delta})} + 2 \sum_{\ell=1}^{k-1} \sum_{t=t_\ell}^{t_{\ell+1}-1} (\mathfrak{g}_\ell - R_t).$$

By design of $\mathcal{H}_{t_k}$ (see [Algorithm 3]), we deduce that (2) $h^* \in \mathcal{H}_{t_k}$. Denote $h_0 \in \mathcal{H}_{t_k}$ the reference point used by [Algorithm 5]. We have, for all $(s,a) \in \mathcal{X}$, on $E_1 \cap E_2 \cap E_3$, we have:

$$(\hat{p}_{t_k}(s,a) - p(s,a)) h^* \leq \sqrt{\frac{2\mathbf{V}(\hat{p}_{t_k}(s,a),h^*) \log\left(\frac{SAT}{\delta}\right)}{N_{t_k}(s,a)}} + \frac{3\mathrm{sp}(h^*) \log\left(\frac{SAT}{\delta}\right)}{N_{t_k}(s,a)}$$

$$(h^* \in \mathcal{H}_{t_k} + \text{Lemma 12}) \leq \sqrt{\frac{2\left(\mathbf{V}(\hat{p}_{t_k}(s,a),h_0) \log\left(\frac{SAT}{\delta}\right) + 8c_0 \sum_{s' \in \mathcal{S}} \hat{p}_{t_k}(s'|s,a) \, \mathrm{error}(c_{t_k},s',s)\right) \log\left(\frac{SAT}{\delta}\right)}{N_{t_k}(s,a)}} + \frac{3c_0 \log\left(\frac{SAT}{\delta}\right)}{N_{t_k}(s,a)}$$

$$=: \beta_{t_k}(s,a)$$

by construction of [Algorithm 5]. Accordingly, (3) is satisfied. Finally, $M \in \mathcal{M}_{t_k}$ on $E_1$ so by [Proposition 2], we have (1) $\mathfrak{g}_k \geq g^*$. $\qquad\square$

**Corollary 14.** *Assume that, for all $q \in \{r, p\}$ and $(s,a) \in \mathcal{X}$, we choose $Q_t(s,a)$ among (**C1-4**). Then, with probability $1 - 4\delta$, for all $k \in K(T)$, we have $\mathfrak{g}_k \geq g^*$ and (2) $h^* \in \mathcal{H}_{t_k}$ and (3) for all $(s,a)$, $(\hat{p}_{t_k}(s,a) - p(s,a)) h^* \leq \beta_{t_k}(s,a)$.*

*Proof.* By [Lemma 11], [Assumption 1] is satisfied. Apply [Lemma 13]. $\qquad\square$

### A.2.3 Sub-Weissman reward confidence region and [Assumption 2]

Although the kernel confidence region can even chosen to be trivial with (**C4**), in order to work, `PMEVI-DT` needs the reward confidence region to be sub-Weissman in the following sense:

**Assumption 2.** There exists a constant $C > 0$ such that for all $(s,a) \in \mathcal{S}$, for all $t \leq T$, we have:

$$\mathcal{R}_t(s,a) \subseteq \left\{ \tilde{r}(s,a) \in \mathcal{R}(s,a) : N_t(s,a) \| \hat{r}_t(s,a) - \tilde{r}(s,a) \|_1^2 \leq C \log\left(\frac{2SA(1+N_t(s,a))}{\delta}\right) \right\}.$$

This is indeed the case if $\mathcal{R}_t(s,a)$ is chosen among (**C1-3**).

### A.3 Convergence of `EVI` and [Assumption 3]

We start with a preliminary lemma on the speed of convergence of `EVI`. The [Lemma 15] is thought to be applied to extended MDPs. Below, when we claim that the action space is compact, we further claim that $a \in \mathcal{A}(s) \mapsto p(s,a)$ is a continuous map, so that the Bellman operator is continuous and that $g^*$ and $h^*$ are well-defined, see [Puterman][1994].

**Lemma 15.** *Let $M$ a weakly-communicating MDP with finite state space $\mathbf{R}^{\mathcal{S}}$ and compact action space, and let $L$ its Bellman operator. Assume that there exists $\gamma > 0$ such that, $\forall u \in \mathbf{R}^{\mathcal{S}}$,*

$$\forall s \in \mathcal{S}, \exists a \in \mathcal{A}(s), \quad Lu(s) = r(s,a) + p(s,a)u = r(s,a) + \gamma \max(u) + (1-\gamma)q_s^u u \qquad (*)$$

*with $q_s^u \in \mathcal{P}(\mathcal{S})$. Then, for all $u \in \mathbf{R}^{\mathcal{S}}$ and all $\epsilon > 0$, if $\mathrm{sp}\left(L^{n+1}u - L^n u\right) \geq \epsilon$, then:*

$$n \leq 2 + \frac{4\mathrm{sp}\,(w_0)}{\gamma\epsilon} + \frac{2}{\gamma}\log\left(\frac{2\mathrm{sp}\,(w_0)}{\epsilon}\right).$$

*Proof.* Since $M$ is weakly communicating, has finitely many states and compact action space, it has well-defined gain $g^*$ and bias $h^*$ functions. Denote $u_{n+1} := L^n u$.

$$w_n := \max_{\pi \in \Pi} \{r_\pi + P_\pi u_{n-1}\} - ng^* - h^*$$
$$= \max_{\pi \in \Pi} \{r_\pi - g^* + (P_\pi - I)h^* + P_\pi (u_{n-1} - h^* - (n-1)g^*)\} =: \max_{\pi \in \Pi} \{r'_\pi + P_\pi w_{n-1}\}.$$

Observe that the policy achieving the maximum is the one achieving $u_n = r_\pi + P_\pi u_{n-1}$. Remark that $r'_\pi(s) = -\Delta^*(s, \pi(s)) \leq 0$ is the Bellman gap of the pair $(s, \pi(s))$, that we more simply write $\Delta_\pi$. For all $n$, there exists $\pi_n \in \Pi$ such that $w_{n+1} = -\Delta_{\pi_n} + P_{\pi_n} w_n$. Moreover, by assumption, we have $P_{\pi_n} = \gamma \cdot e_{s_n}^\top e + (1-\gamma)Q_n$ where $Q_n$ is a stochastic matrix. Moreover,

$$(\min(-\Delta_{\pi_n}) + \gamma w_n(s_n))\,e + (1-\gamma)Q_n w_n \leq w_{n+1} \leq (\max(-\Delta_{\pi_n}) + \gamma w_n(s_n))\,e + (1-\gamma)Q_n w_n.$$

Hence, $\mathrm{sp}\,(w_{n+1}) \leq (1-\gamma)\mathrm{sp}\,(w_n) + \mathrm{sp}\,(\Delta_{\pi_n})$. In addition, $w_n = L^n u - L^n h^*$, so by non-expansiveness of $L$ in span semi-norm, $\mathrm{sp}\,(w_{n+1}) \leq \mathrm{sp}\,(w_n)$. Overall,

$$\mathrm{sp}\,(w_{n+1}) \leq \min\left((1-\gamma)\mathrm{sp}\,(w_n) + \mathrm{sp}\,(\Delta_{\pi_n}), \mathrm{sp}\,(w_n)\right). \qquad (15)$$

Fix $\epsilon > 0$, and let $n_\epsilon := \inf\{n : \mathrm{sp}\,(w_n) < \epsilon\}$.

Let $\pi^*$ an optimal policy. We have $w_{n+1} \geq P_{\pi^*} w_n$ so by induction, $w_{n+1} \geq P_{\pi^*}^{n+1} w_0 \geq \min(w_0)e$. Meanwhile, we see that $\|w_n\|_1 \geq \sum_{k=0}^{n-1} \|\Delta_{\pi_k}\|_1 + S\min(w_0)$, so $\sum_{k=0}^{n-1} \|\Delta_{\pi_k}\|_1 \leq \mathrm{sp}\,(w_0)$. Since $\Delta_{\pi_k} \leq 0$ for all $k$, we have $\mathrm{sp}\,(\Delta_{\pi_k}) \leq \|\Delta_{\pi_k}\|_1$ so $\sum_{k=0}^{n-1} \mathrm{sp}\,(\Delta_{\pi_k}) \leq \mathrm{sp}\,(w_0)$.

By (15), either $\mathrm{sp}\,(w_{n+1}) \leq (1 - \frac{1}{2}\gamma)\max(\epsilon, \mathrm{sp}\,(w_n))$ or $\mathrm{sp}\,(\Delta_{\pi_n}) \geq \frac{1}{2}\gamma\epsilon$, but because $\sum_{k=0}^{+\infty} \mathrm{sp}\,(\Delta_{\pi_k}) \leq \mathrm{sp}\,(w_0)$, the second case can happen at most $\frac{2\mathrm{sp}(w_0)}{\gamma\epsilon}$ times. We deduce that, for all $n \leq n_\epsilon$,

$$\mathrm{sp}\,(w_{n+1}) \leq \left(1 - \tfrac{1}{2}\gamma\right)^{n - \frac{2\mathrm{sp}(w_0)}{\gamma\epsilon}} \mathrm{sp}\,(w_0).$$

In particular, for $n = n_\epsilon - 1$, we get:

$$\epsilon \leq \left(1 - \tfrac{1}{2}\gamma\right)^{n_\epsilon - 2 - \frac{2\mathrm{sp}(w_0)}{\gamma\epsilon}} \mathrm{sp}\,(w_0).$$

We obtain:

$$n_\epsilon \leq 2 + \frac{2\mathrm{sp}\,(w_0)}{\gamma\epsilon} + \frac{2}{\gamma}\log\left(\frac{\mathrm{sp}\,(w_0)}{\epsilon}\right).$$

To conclude, check that $\mathrm{sp}\left(L^{n+1}u - L^n u\right) = \mathrm{sp}\,(w_{n+1} - w_n) \leq 2\mathrm{sp}\,(w_n)$. $\qquad\square$

Before moving to the application of interest, remark that this result can be greatly improved if the supremum $\sup\{\Delta^*(s,a) : \Delta^*(s,a) < 0\}$ is not zero, to change the dominant term $\frac{4\mathrm{sp}(w_0)}{\gamma\epsilon}$ for a constant independent of $\epsilon$.

**Corollary 16.** *Assume that the $\mathcal{M}_t$ has non-empty interior, and that its Bellman operator satisfies the requirement of [Lemma 15](), i.e., there exists $\gamma > 0$ such that, $\forall u \in \mathbf{R}^{\mathcal{S}}, \forall s \in \mathcal{S}, \exists a \in \mathcal{A}(s), \exists \tilde{r}_t(s,a) \in \mathcal{R}_t(s,a), \exists \tilde{p}_t(s,a) \in \mathcal{P}_t(s,a)$:*

$$\mathcal{L}_t u(s) = \tilde{r}_t(s,a) + \tilde{p}_t(s,a)u = \tilde{r}_t(s,a) + \gamma \max(u) + (1-\gamma)q_s^u u$$

*for some $q_s^u \in \mathcal{P}(\mathcal{S})$. Then [Assumption 3]() is satisfied, and span fix-points $\tilde{h}_t$ of $\mathcal{L}_t$ are such that $g^*(\mathcal{M}_t) = \mathcal{L}_t\tilde{h}_t - \tilde{h}_t$.*

*Proof.* If $\mathcal{M}_t$ is has non-empty interior, it means that for all $(s, a)$, $\mathcal{P}_t(s, a)$ has non-empty interior. Therefore, for all state-action pair, there exists $\tilde{p}_t(s, a) \in \mathcal{P}_t(s, a)$ that is fully supported. It follows that $\mathcal{M}_t$ is communicating, and it follows from standard results Puterman [1994] that its span fix-points $\tilde{h}$ do exist and that $\tilde{g}_t := \mathcal{L}\tilde{h}_t - \tilde{h}_t \in \mathbf{R}e$ does not depend on the initial state.

Moreover, if $\widetilde{M} \in \mathcal{M}_t$ and $\pi \in \Pi$ with $\tilde{g}_\pi \equiv g(\pi, \mathcal{M}_t) \in \mathbf{R}e$, letting $\tilde{r}_\pi := r_\pi(\tilde{M})$ and $\tilde{P}_\pi := P_\pi(\tilde{M})$, we have:
$$\tilde{r}_\pi + \tilde{p}_\pi \tilde{h}_t \leq \mathcal{L}_t \tilde{h}_t \leq \tilde{g}_t e + \tilde{h}_t.$$
So by induction and since $\mathcal{L}_t$ is obviously monotone and linear, we show that:
$$\sum_{k=0}^{n} \tilde{P}_\pi^k \tilde{r}_\pi \leq n\tilde{g}_t e + (I - \tilde{P}_\pi^n)\tilde{h}_\pi.$$

Dividing by $n$ and letting it go to infinity, we obtain $g(\pi, \mathcal{M}_t) \leq \tilde{g}_t$. Observe that we have equility by taking the policy achieving $(\tilde{g}_t, \tilde{h}_t)$.

To see that `EVI` converges indeed, simply observe that Lemma 15 provides a finite bound on how much time is required until the $\text{sp}\left(\mathcal{L}_t^{n+1}u - \mathcal{L}_t^n u\right) \leq \epsilon$. Hence $\text{sp}\left(\mathcal{L}_t^{n+1}u - \mathcal{L}_t^n u\right)$ vanishes to $0$. $\quad\square$

**About Assumption 3.** The assumptions made by Corollary 16 are met if the kernel confidence regions are:

- Built out of Weissman's inequality (**C1**) (see the next section, also Auer et al. [2009]);
- Built out of Bernstein's inequality (**C2**) (because the maximization algorithm to compute $\tilde{p}_t(s, a)u_i$ in `EVI` has the same greedy properties than with Weissman's inequality);
- Trivial (**C4**).

For confidence regions build with empirical likelihood estimates (**C3**), there is no guarantee of convergence (although we conjecture that one could be established), although the gain is still well-defined because $\mathcal{M}_t$ remains communicating. However, just like the original work of Filippi et al. [2010], the convergence is always met numerically.

## A.4 Proof of Theorem 5: Complexity of `PMEVI` with Weissman confidence regions

In this section, we show that when one is using Weissman confidence regions for kernels (**C1**), then the iterates of $\mathcal{L}_t$ converge to an $\epsilon$ span-fix-point quickly.

**Proposition 17.** *Assume that `PMEVI-DT` uses kernel confidence regions of Weissman type (**C1**) satisfying Assumption 1. Then with probability $1 - \delta$, the number of iterations of `PMEVI` (see Algorithm 2) is $\mathrm{O}\left(D\sqrt{S}AT\right)$, hence the algorithm has polynomial per-step amortized complexity.*

*Proof.* With Weissman type confidence regions for kernels, for all $t \leq T$ and $(s, a) \in \mathcal{X}$, we have
$$\mathcal{P}_t(s, a) \supseteq \left\{ \tilde{p}(s, a) \in \mathcal{P}(s, a) : \|\tilde{p}(s, a) - \hat{p}_t(s, a)\|_1 \leq \sqrt{\frac{S \log(2SAT)}{T}} \right\}$$

It follows that, for all $t \leq T$, the extended Bellman operator $\mathcal{L}_t$ satisfies the prerequisite $(*)$ of Lemma 15 with
$$\gamma = \frac{1}{2}\sqrt{\frac{S \log(2SAT/\delta)}{T}} = \Omega\left(\sqrt{\frac{S \log(T/\delta)}{T}}\right).$$

Under Assumption 1, we have $M \in \mathcal{M}_t$ with probability $1 - \delta$. Under this event, $\mathcal{M}_t$ is weakly communicating and $\text{sp}\left(h^*(\mathcal{M}_t)\right) \leq D(M)$, we can apply Lemma 15 and conclude that every calls to `PMEVI` (Algorithm 2) takes
$$\mathrm{O}\left(\frac{\text{sp}(w_0)\sqrt{T}}{\epsilon\sqrt{\frac{S \log(T/\delta)}{T}}}\right) = \mathrm{O}\left(\frac{DT}{\sqrt{S}\log(T)}\right)$$

where we use that $\epsilon = \sqrt{\frac{\log(SAT/\delta)}{T}}$, that $\text{sp}(w_0) = \mathrm{O}\left(\text{sp}\left(h^*(\mathcal{M}_t)\right)\right) = \mathrm{O}(D(M))$ and that $\delta \geq \frac{1}{T}$. Since the number of episodes under the doubling trick (DT) is $\mathrm{O}(SA\log(T))$, we conclude accordingly. $\quad\square$

Every call to the projection operator solves a linear program. Although in theory, this time is polynomial (relying on recent work on the complexity of LP such as Cohen et al. [2020], it is the current matrix multiplication time $O(S^{2.38})$), in practice, reducing the number of calls to the projection operator is key to run `PMEVI-DT` in reasonable time.

# B  Analysis of the projected mitigated Bellman operator

In this section, we fix the model region $\mathcal{M}$, the bias region $\mathcal{H}$ and the mitigation vector $\beta$, dropping the sub-script $t$ for conciseness. We denote $\hat{r}, \hat{p}$ the respective empirical reward and kernel. Further assume that $\mathcal{H} = \mathcal{H}_0 + \mathbf{R}e$ with $\mathcal{H}_0$ a compact convex set. The associated projection operation (see Appendix B.2) is denoted $\Gamma$. The (vanilla) extended Bellman operator $\mathcal{L}$ associated to $\mathcal{M}$ is given by $\mathcal{L}u(s) := \max_{a \in \mathcal{A}(s)} \{\sup \mathcal{R}(s, a) + \sup \mathcal{P}(s, a)u\}$. The $\beta$-*mitigated extended Bellman operator* associated to $\mathcal{M}$ is:

$$\mathcal{L}^\beta u(s) := \max_{a \in \mathcal{A}(s)} \sup_{\tilde{r}(s,a) \in \mathcal{R}(s,a)} \sup_{\tilde{p}(s,a) \in \mathcal{P}(s,a)} \left\{ \tilde{r}(s, a) + \min\{\tilde{p}(s,a)u_i, \hat{p}(s,a)u_i + \beta(s,a)\} \right\}. \tag{16}$$

The function $\text{Greedy}(\mathcal{M}, u, \beta)$ returns a stationary deterministic policy that picks its actions among the one reaching the maximum above. The projection of $\mathcal{L}^\beta$ to $\mathcal{H}$ is

$$\mathfrak{L} \equiv \mathfrak{L}^{\beta,\mathcal{H}} := \Gamma \circ \mathcal{L}^\beta. \tag{17}$$

The goal of this section is to establish Proposition 2 and

- Proposition 2 statement (1) is a consequence of Lemma 22;
- Proposition 2 statement (2) follows from Theorem 25;
- Proposition 2 statement (3) follows from Corollary 27;
- Proposition 2 statement (4) follows from Corollary 21;
- Proposition 2 prerequisites on the projection operator and Lemma 4 follows from Lemma 19.

## B.1  Finding an optimistic policy under bias constraints

The main goal is to find and optimistic policy under *bias constraints* (projection) and *bias error constraints* (mitigation). The bias constraints imply that we search for a policy $\pi$ together with a model $\widetilde{M}$ such that $h^\pi(\widetilde{M}) \in \mathcal{H}$. The bias error means that, for $\tilde{h} \equiv h^\pi(\widetilde{M})$, we want in addition $\tilde{p}(s, \pi(s))\tilde{h} \leq \hat{p}(s, \pi(s))\tilde{h} + \beta(s, \pi(s))$ where $\tilde{p}$ is the transition kernel of $\widetilde{M}$. In the end, our goal is to track the solution of the following optimization problem:

$$g^*(\mathcal{H}, \beta, \mathcal{M}) := \sup \left\{ g^\pi(\widetilde{M}) : \begin{array}{c} \pi \in \Pi, \widetilde{M} \in \mathcal{M}, \\ \forall s \in \mathcal{S}, \ \tilde{p}(s, \pi(s))\tilde{h} \leq \hat{p}(s, \pi(s))\tilde{h} + \beta(s, \pi(s)), \\ \tilde{h} \equiv h^\pi(\widetilde{M}) \in \mathcal{H}, \ \text{sp}(g^\pi(\widetilde{M})) = 0 \end{array} \right\} \tag{18}$$

where the supremum is taken with respect to the product order $\mathbf{R}^\mathcal{S}$. In particular, if $\mathcal{U} \subseteq \mathcal{R}^\mathcal{S}$, check that $u^* = \sup \mathcal{U}$ is obtained as $u^*(s) := \sup\{v(s) : v \in \mathcal{U}\}$. The constraint $\text{sp}(g^\pi(\widetilde{M})) = 0$ is suggested by the work of Fruit et al. [2018], Fruit [2019] and is key for the problem to be solvable.

The bias constraint and the constraint involving $\beta$ make the problem impossible to handle with a "pure" extended MDP solution, which is why the extended Bellman operators are mitigated (with $\beta$) then projected (with $\Gamma$). The mitigation operation guarantees that the constraint involving $\beta$ is satisfied, while the projection on $\mathcal{H}$ makes sure that the bias constraint is satisfied. It is important for both operations to be compatible, i.e., that the constraint involving $\beta$ that $\mathcal{L}^\beta$ forces is not lost when applying $\Gamma$. As a matter of fact, projecting then mitigating would not work.

We now explain why $\mathfrak{L}$ can be used to solve (18).

## B.2  Projection operation and definition of $\mathfrak{L}$

We start by discussing why $\mathfrak{L}$ is well-defined at all. The well-definition of $\mathcal{L}^\beta$ is obvious. The point is to explain why the projection onto $\mathcal{H}$ is possible while preserving mandatory structural properties such as monotony, non-expansivity, linearity and more. For general $\mathcal{H}$, such properties are impossible to meet. But the bias confidence region constructed with Algorithm 3 has a specific shape that makes the projection possible. The central property is the one below:

(**A1**) *The downward closure $\{v \leq u : v \in \mathcal{H}\}$ of every $u \in \mathbf{R}^\mathcal{S}$ has a maximum in $\mathcal{H}$.*

The only order that we will be considering is the product order on $\mathbf{R}^\mathcal{S}$. Recall that a set $\mathcal{U} \subseteq \mathbf{R}^\mathcal{S}$ has a *maximum* if there exists $u \in \mathcal{U}$ such that $v \leq u$ for all $u \in \mathcal{U}$. A *supremum* of $\mathcal{U}$ is a

minimal upper-bound of $\mathcal{U}$, i.e., $u$ such that (1) $v \leq u$ for all $v \in \mathcal{U}$ and (2) no $w$ satisfying (1) can be smaller than $u$. For the product order, the supremum of a subset $\mathcal{U}$ is unique and of the form $u(s) = \sup\{v(s) : v \in \mathcal{U}\}$.

Define the projection $\Gamma : \mathbf{R}^{\mathcal{S}} \rightarrow \mathcal{H}$ as such:

$$\Gamma u := \max\{v \leq u : v \in \mathcal{H}\}. \tag{19}$$

In general, Assumption (**A1**) is satisfied when $\mathcal{H}$ admits a join, i.e., is stable by finite supremum: $u, v \in \mathcal{H} \Rightarrow \sup(u, v) \in \mathcal{H}$.

**Lemma 18.** *If $\mathcal{H}$ is generated by constraints of the form $\mathfrak{h}(s) - \mathfrak{h}(s') - c(s, s') \leq d(s, s')$, then it has a join and (A1) is satisfied. Moreover, $\Gamma$ is then correctly computed with Algorithm 4.*

*Proof.* The first half of the result is well-known, see Zhang and Xie [2023], but we recall a proof for self-containedness. Let $v_1, v_2 \in \mathcal{H}$ and define $v_3 := \sup(v_1, v_2)$. Observe that $v_3(s) - v_3(s') \leq \max(v_1(s) - v_1(s'), v_2(s) - v_2(s')) \leq c(s, s') + d(s, s')$. So $v_3 \in \mathcal{H}$.

We continue by showing that if $\mathcal{H}$ has a join, then (19) is well-defined. For $s \in \mathcal{S}$, take a sequence $v_n^s$ such that $v_n^s(s) \rightarrow \alpha(s) := \sup\{v(s) : v \leq u, v \in \mathcal{H}\}$. Because the span of every element of $\mathcal{H}$ is upper-bounded by $c := \sup\{\mathrm{sp}(v) : v \in \mathcal{H}\}$, it follows that $v_n^s$ evolves in the compact region $\{v \leq u : v \in \mathcal{H}\} \cap \{v : \|v - \alpha se\|_\infty = 1 + c\}$. We can therefore extract a convergent sequence of $v_n^s$, converging $v_*^s$ that belongs to $\mathcal{H}$ since the latter is closed. By construction, $v_*^s(s) = \alpha(s)$. Because $\mathcal{H}$ has a join, $v_* := \sup\{v_*^s : s \in \mathcal{S}\} \in \mathcal{H}$. $\qquad\square$

**Lemma 19.** *Under assumption (A1), the operator $\Gamma u := \max\{v \leq u : v \in \mathcal{H}\}$ is well-defined, and is:*

*(1) monotone: $u \leq v \Rightarrow \Gamma u \leq \Gamma v$;*

*(2) non span-expansive: $\mathrm{sp}(\Gamma u - \Gamma v) \leq \mathrm{sp}(u - v)$;*

*(3) linear: $\Gamma(u + \lambda e) = \Gamma u + \lambda e$;*

*(4) $\Gamma u \leq u$.*

*Proof.* The well-definition of $\Gamma$ is obvious from (**A1**). For (2), if $u \leq v$ then $w \leq u \Rightarrow w \leq v$. Hence $\Gamma u := \max\{w \leq u : w \in \mathcal{H}\} \leq \max\{w \leq v : w \in \mathcal{H}\} =: \Gamma v$. For (3), check that it follows from $\mathcal{H} = \mathcal{H} + \mathbf{R}e$. For (4), we obviously have $\Gamma u := \max\{v \leq u : v \in \mathcal{H}\} \leq u$.

The more difficult point is (2) span non-expansivity. Pick $u, v \in \mathbf{R}^{\mathcal{S}}$. By linearity, it suffices to show the result for $\sum_s u(s) = \sum_s v(s)$. In that case, we have $\mathrm{sp}(v - u) = \max(v - u) + \max(u - v)$. Observe that for all $w \leq u$, we have $w + \min(v - u)e \leq v$. Since $\mathcal{H} = \mathcal{H} + \mathbf{R}e$, it follows that:

$$\max\{w \leq u : u \in \mathcal{H}\} \leq \max\{w \leq v : w \in \mathcal{H}\} + \max(u - v)e.$$

Similarly, we have $\max\{w \leq u : w \in \mathcal{H}\} \geq \max\{w \leq v : w \in \mathcal{H}\} + \min(v - u)e$. Using them both at once, we find $\mathrm{sp}(\Gamma u - \Gamma v) \leq \mathrm{sp}(v - u)$. $\qquad\square$

The properties (1), (3) and (4) are essential for $\mathfrak{L}$ to properly address the optimization problem (18). The property (2) is just as important, because it plays a central part in the convergence of value iteration. The next result shows similar properties for the $\beta$-mitigated extended Bellman operator $\mathcal{L}^\beta$. From now on, we will assume (A1), because it is almost-surely satisfied by the bias confidence region generated by Algorithm 3.

**Lemma 20.** *The $\beta$-mitigated extended Bellman operator $\mathcal{L}^\beta$ is (1) monotone, (2) non-span-expansive and (3) linear.*

*Proof.* The properties (1) and (3) directly follow from the definition. We focus on (2). Fix $u, u' \in \mathbf{R}^{\mathcal{S}}$. By Lemma 26, we can write $\mathcal{L}^\beta u = \tilde{r}_\pi + \tilde{P}_\pi u$ and $\mathcal{L}^\beta u' = \tilde{r}_{\pi'} + \tilde{P}_{\pi'} u'$. In the following, we write $\beta_\pi(s) := \beta(s, \pi(s))$. Check that:

$$\mathcal{L}^\beta u - \mathcal{L}^\beta u' = \tilde{r}_\pi + \tilde{P}_\pi u - \left(\tilde{r}_{\pi'} + \tilde{P}_{\pi'} u'\right) \leq \tilde{r}_\pi + \tilde{P}_\pi u - \left(\tilde{r}_\pi + \min\left\{\tilde{P}_\pi u', \hat{P}_\pi u' + \beta_\pi\right\}\right).$$

If the minimum is reached with $\tilde{P}_\pi u'$, then:

$$\mathcal{L}^\beta u - \mathcal{L}^\beta u' \leq \tilde{P}_\pi(u - u').$$

If the minimum is reached with $\hat{P}_\pi u' + \beta_\pi$, then upper-bound $\tilde{P}_\pi u$ by $\hat{P}_\pi u + \beta_\pi$ to obtain:

$$\mathcal{L}^\beta u - \mathcal{L}^\beta u' \leq \hat{P}_\pi(u - u').$$

Overall, we find that there exists $Q_\pi \in \mathcal{P}_\pi$ such that $\mathcal{L}^\beta u - \mathcal{L}^\beta u' \leq Q_\pi(u - u')$. Similarly, we find $Q_{\pi'} \in \mathcal{P}_{\pi'}$ such that $\mathcal{L}^\beta u - \mathcal{L}^\beta u' \geq Q_{\pi'}(u - u')$. We conclude that:

$$\mathrm{sp}\left(\mathcal{L}^\beta u - \mathcal{L}^\beta u'\right) \leq \mathrm{sp}\left((Q_\pi - Q_{\pi'})(u - u')\right) \leq \mathrm{sp}\left(u - u'\right).$$

This concludes the proof. $\qquad\square$

By composition, we obtain the following result.

**Corollary 21.** $\mathfrak{L}$ *is (1) monotone, (2) non-span-expansive and (3) linear. Moreover,* $\mathrm{sp}\left(\mathfrak{L}u - \mathfrak{L}v\right) \leq \mathrm{sp}\left(\mathcal{L}u - \mathcal{L}v\right)$ *for all* $u, v \in \mathbf{R}^{\mathcal{S}}$.

### B.3 Fix-points of $\mathfrak{L}$ and (weak) optimism

**Lemma 22.** $\mathfrak{L}$ *has a fix-point in span semi-norm, i.e.,* $\exists u \in \mathcal{H}, \mathrm{sp}\left(\mathfrak{L}u - u\right) = 0$.

*Proof.* The idea is to apply Brouwer's fix-point theorem in $\mathbf{R}^{\mathcal{S}}$ quotiented by the equivalence relation $u \sim v \Leftrightarrow \mathrm{sp}\left(u - v\right) = 0$, where $\mathrm{sp}\left(-\right)$ becomes a norm. By linearity (Corollary 21), $\mathfrak{L}$ is well-defined in this quotient space, and if $\mathfrak{L}$ is shown continuous on $\mathbf{R}^{\mathcal{S}}$, so will it be on the quotient.

We show that $\mathfrak{L}$ is sequentially continuous on $\mathcal{H}$. Consider a sequence $u_n \in \mathcal{H}^{\mathbf{N}}$ converging to $u \in \mathcal{H}$ and fix $\epsilon > 0$. Provided that $n > N_\epsilon$ for $N_\epsilon$ large enough, we have $\|u_n - u\|_\infty < \epsilon$, i.e., $u_n - \epsilon e \leq u_n \leq u + \epsilon e$. Therefore, in the one hand, for all $v \leq u_n$, we have $v - \epsilon e \leq u$ so $\max\{v \leq u_n : v \in \mathcal{H}\} \leq \max\{v \leq u : v \in \mathcal{H}\} + \epsilon e$; And on the other hand, for all $v \leq u$, $v + \epsilon e \leq u_n$ so $\max\{v \leq u : v \in \mathcal{H}\} \leq \max\{v \leq u_n : v \in \mathcal{H}\} + \epsilon e$. Hence:

$$\|\max\{v \leq u : v \in \mathcal{H}\} - \max\{v \leq u_n : v \in \mathcal{H}\}\| \leq \epsilon.$$

It shows that $\Gamma$ is continuous. The operator $\mathcal{L}^\beta$ is obviously continuous as well, so $\mathfrak{L} = \Gamma \circ \mathcal{L}^\beta$ is continuous by composition. Since $\mathcal{H} = \mathcal{H}_0 + \mathbf{R}e$ with $\mathcal{H}_0$ compact and ocnvex, the quotient $\mathcal{H}/\sim$ is compact and convex, and is preserved by $\mathfrak{L}/\sim$. By Brouwer's fix-point theorem, $\mathfrak{L}/\sim$ has a fix-point in $\mathcal{H}/\sim$. So $\mathfrak{L}$ has a span fix-point in $\mathcal{H}$. $\qquad\square$

We write $\mathrm{Fix}(\mathfrak{L})$ the span fix-points of $\mathfrak{L}$.

**Lemma 23.** $\mathfrak{L}$ *has well-defined growth. Specifically, if* $\mathfrak{L}u = u + \mathfrak{g}e$, *then:*

*(1) There exists* $c > 0$, *s.t., for all* $v \in \mathcal{H}_0$, $(n\mathfrak{g} - c)e + u \leq \mathfrak{L}^n v \leq (n\mathfrak{g} + c)e + u$;

*(2) If* $u' \in \mathrm{Fix}(\mathfrak{L})$, *then* $\mathfrak{L}u' - u' = \mathfrak{g}e$.

*Proof.* Setting $c := \max_{v \in \mathcal{H}_0} \|v - u\|_\infty < \infty$, one can check that $u - ce \leq v \leq u + ce$ for all $v \in \mathcal{H}_0$. this proves (1) for $n = 0$ and we then proceed by induction on $n \geq 0$. By induction, $\mathfrak{L}^n v \leq u + (n\mathfrak{g} + c)e$ and by Corollary 21, $\mathfrak{L}$ is monotone, so we have:

$$\mathfrak{L}^{n+1}v \leq \mathfrak{L}\mathfrak{L}^n v \leq \mathfrak{L}(u + (n\mathfrak{g} + c)e) = u + ((n+1)\mathfrak{g} + c)e$$

where the last inequality use the linearity of $\mathfrak{L}$ together with $\mathfrak{L}u = u + \mathfrak{g}e$. The lower bound of $\mathfrak{L}^n v$ is shown similarly, establishing (1).

For (2), pick $u' \in \mathrm{Fix}(\mathfrak{L})$ with $\mathfrak{L}u' = u' + \mathfrak{g}'e$. Up to translating $u'$, we can assume that $u' \in \mathcal{H}_0$ and apply (1). We get:

$$(n\mathfrak{g} - c)e + u \leq n\mathfrak{g}'e + u' \leq (n\mathfrak{g} + c)e + u.$$

Divided by $n$ and let it go to infinity. We conclude that $\mathfrak{g} = \mathfrak{g}'$. $\qquad\square$

We finally have everything in hand to claim that $\mathfrak{L}$ solves (18).

**Corollary 24.** *The* growth *of* $\mathfrak{L}$ *given by* $\mathfrak{g} = \mathfrak{L}u - u$ *for* $u \in \mathrm{Fix}(\mathfrak{L})$ *is well-defined, and:*

$$\forall u \in \mathcal{H}, \quad \mathfrak{g}e = \liminf_{n \to \infty} \frac{\mathfrak{L}^n u}{n} = \limsup_{n \to \infty} \frac{\mathfrak{L}^n u}{n}.$$

*Moreover,* $\mathfrak{g} \geq g^*(\mathcal{H}, \beta, \mathcal{M})$.

*Proof.* The growth property is a direct consequence of Lemma 23. We show $\mathfrak{g} \geq g^*(\mathcal{H}, \beta, \mathcal{M})$ which is defined in (18). Pick $\pi \in \Pi$, $\widetilde{M} \in \mathcal{M}$ its model with $\tilde{h} \equiv h(\pi, \widetilde{M})$ and $\tilde{P}_\pi \tilde{h} \leq \hat{P}_\pi \tilde{h} + \beta_\pi$ where $\beta_\pi(s) := \beta(s, \pi(s))$. Up to translation, we can assume that $\tilde{h} \in \mathcal{H}_0$.

We have $g(\pi, \widetilde{M}) = \tilde{g}e$ for $\tilde{g} \in \mathbf{R}$, so

$$\tilde{h} + \tilde{g}e = \tilde{r}_\pi + \tilde{P}_\pi \tilde{h} \leq \mathfrak{L}\tilde{h}$$

by definition. By monotony of $\mathfrak{L}$, see Corollary 21, $n\tilde{g}e + \tilde{h} \leq \mathfrak{L}^n \tilde{h}$ follows by induction on $n \geq 0$. By Lemma 23, we further have $\mathfrak{L}^n \tilde{h} \leq n(\mathfrak{g} + c)e + u$ where $u \in \text{Fix}(\mathfrak{L})$. In tandem,

$$\tilde{g}e \leq \mathfrak{g}e + \frac{ce + u - \tilde{h}}{n}.$$

Letting $n \to \infty$, we deduce that $\tilde{g} \leq \mathfrak{g}$. Conclude by taking the best $\pi$ and $\widetilde{M}$. □

The next theorem follows directly with the same proof technique, and guarantees optimism.

**Theorem 25.** *Assume that $g^* + h^* \leq \mathfrak{L}h^*$. Then $\mathfrak{g} \geq g^*$.*

The condition "$g^* + h^* \leq \mathfrak{L}h^*$" can be referred to as a *weak* form of optimism. We qualify this version of optimism as *weak* because it is much weaker than optimism property suggested by Fruit [2019] $\mathcal{L} \geq L$ where $L$ is the Bellman operator of the true MDP. Here, we only ask for $\mathfrak{L}h^* \geq Lh^*$, i.e., optimism at a span fix-point of $L$. This condition is met as soon as $M \in \mathcal{M}$, $h^* \in \mathcal{H}$ and $\beta$ large enough.

### B.4 Modelization of the projected mitigated Bellman operator $\mathfrak{L}$

The aim of this paragraph is to establish Corollary 27, stating that $\mathfrak{L}u$ can be viewed as a policy produced by $\texttt{Greedy}(\mathcal{M}, u, \beta)$.

**Lemma 26** (Modelization). *For $\pi \in \Pi$, denote $\beta_\pi(s) := \beta(s, \pi(s))$, $\mathcal{R}_\pi := \prod_s \mathcal{R}(s, \pi(s))$ and $\mathcal{P}_\pi := \prod_s \mathcal{P}(s, \pi(s))$. Fix $u \in \mathbf{R}^\mathcal{S}$ and let $\pi := \texttt{Greedy}(\mathcal{M}, u, \beta)$.*

*(1) If $\mathcal{P}$ is convex, then there exists $(\tilde{r}_\pi, \tilde{P}_\pi) \in \mathcal{R}_\pi \times \mathcal{P}_\pi$ such that $\mathcal{L}_\beta u = \tilde{r}_\pi + \tilde{P}_\pi u$.*

*(2) Assume that $\mathcal{L}_\beta u = \tilde{r}_\pi + \tilde{P}_\pi u$. There exists $r'_\pi \leq \tilde{r}_\pi$ such that $\mathfrak{L}u = r'_\pi + \tilde{P}_\pi u$.*

The convexity requirement of (1) is always true if the kernel confidence region is chosen via (**C1-4**).

*Proof.* For (1), fix a state $s \in \mathcal{S}$, let $a := \pi(s)$ and $\rho := \min(\sup \mathcal{P}(s, a)u, \hat{p}(s, a)u + \beta(s, a))$. If $\rho = \sup \mathcal{P}(s, a)u$, then there is nothing to say because $\mathcal{P}$ is compact, hence the sup is a max and $\rho$ is of the form $\tilde{p}(s, a)u$. Otherwise, let $\tilde{p}(s, a)u > \hat{p}(s, a)u + \beta(s, a)$ with $\tilde{p}(s, a) \in \mathcal{P}(s, a)$. Introduce, for $\lambda \in [0, 1]$,
$$\tilde{p}_\lambda(s, a) := \lambda \tilde{p}(s, a) + (1 - \lambda)\hat{p}(s, a).$$
By continuity, there exists $\lambda \in (0, 1)$ such that $\tilde{p}_\lambda(s, a)u = \hat{p}(s, a)u + \beta(s, a)$ and by convexity of $\mathcal{P}(s, a)$, $\tilde{p}_\lambda(s, a) \in \mathcal{P}(s, a)$. This proves (1).

For (2), recall that $\mathfrak{L}u = \Gamma \mathcal{L}^\beta u = \Gamma(\tilde{r}_\pi + \tilde{P}_\pi u)$. Since $\Gamma v \leq v$, for $v \in \mathbf{R}^\mathcal{S}$, we have:

$$\Gamma(\tilde{r}_\pi + \tilde{P}_\pi u) \leq \tilde{r}_\pi + \tilde{P}_\pi u.$$

Set $r'_\pi := \Gamma(\tilde{r}_\pi + \tilde{P}_\pi u) - \tilde{P}_\pi u$. Check that $r'_\pi$ satisfies $r'_\pi \leq \tilde{r}_\pi$ and $\mathfrak{L}u = r'_\pi + \tilde{P}_\pi u$. □

The last corollary bellow is crucial to claim that greedy policies are good choices in $\texttt{PMEVI-DT}$.

**Corollary 27** (Greedy modelization). *Let $u \in \mathbf{R}^\mathcal{S}$ and fix $\pi := \texttt{Greedy}(\mathcal{M}, u, \beta)$. If $\mathcal{P}$ is convex, then with the notations of Lemma 26, there exists $\tilde{r}_\pi \leq \sup \mathcal{R}_\pi$ and $\tilde{P}_\pi \in \mathcal{P}_\pi$ such that $\mathfrak{L}u = \tilde{r}_\pi + \tilde{P}_\pi u$.*

## C Proof of [Theorem 5](#): Regret analysis of `PMEVI-DT`

**Notations.** At episode $k$, the played policy is denoted $\pi_k$. As a greedy response to $\mathfrak{h}_k$, by [Proposition 2](#) (3), there exists $\tilde{r}_k(s) \leq \sup \mathcal{R}_{t_k}(s, \pi_k(s))$ and $\tilde{P}_k(s) \in \mathcal{P}_{t_k}(s, \pi(x))$ such that $\mathfrak{h}_k + \mathfrak{g}_k = \tilde{r}_k + \tilde{P}_k \mathfrak{h}_k$. The reward-kernel pair $\tilde{M}_k = (\tilde{r}_k, \tilde{P}_k)$ is referred to as the *optimistic model* of $\pi_k$. We write $P_k := P_{\pi_k}(M)$ the true kernel and $\hat{P}_k := P_{\pi_k}(\hat{M}_{t_k})$ the empirical kernel. Likewise, we define the reward functions $r_k$ and $\hat{r}_k$. The optimistic gain and bias satisfy $\mathfrak{g}_k = g(\pi_k, \widetilde{M}_k)$ and $\mathfrak{h}_k = h(\pi_k, \widetilde{M}_k)$. We further denote $c_0 = T^{\frac{1}{5}}$.

**Important remark.** To slightly simplify the analysis, we assume that `PMEVI` is run with perfect precision $\epsilon = 0$, i.e., that $\mathfrak{h}_k = \texttt{PMEVI}(\mathcal{M}_{t_k}, \beta_{t_k}, \Gamma_{t_k}, 0)$ hence is a span fix-point of $\mathfrak{L}_{t_k}$. This assumption is mild and can be dropped by adding an extra error term that has to be carried out in the calculations.

### C.1 Number of episodes under doubling trick (DT)

**Lemma 28** (Number of episodes, [Auer et al. [2009]](#)). *The number of episodes up to time $T \geq SA$ is upper-bounded by:*

$$K(T) \leq SA \log_2 \left( \tfrac{8T}{SA} \right).$$

### C.2 Sum of bias variances

The [Lemma 29](#) below shows that $\sum_{t=0}^{T-1} \mathbf{V}(p(X_t), h^*)$ scales as $T \operatorname{sp}(h^*) \operatorname{sp}(r) + \operatorname{sp}(h^*) \operatorname{Reg}(T)$ in probability.

**Lemma 29.** *With probability at least $1 - \delta$, we have:*

$$\sum_{t=0}^{T-1} \mathbf{V}(p(X_t), h^*) \leq 2 \operatorname{sp}(h^*) \operatorname{sp}(r) T + \operatorname{sp}(h^*)^2 \sqrt{\tfrac{1}{2} T \log \left( \tfrac{1}{\delta} \right)} + 2 \operatorname{sp}(h^*) \sum_{t=0}^{T-1} \Delta^*(X_t) + \operatorname{sp}(h^*)^2.$$

*Proof.* Using the Bellman equation $h^*(s) + g^*(s) = r(s, a) + p(s, a) h^* + \Delta^*(s, a)$, we have:

$$\mathbf{V}(p(X_t), h^*) = (p(X_t) - e_{S_t}) h^{*2} + 2 h^*(S_t)(\Delta^*(X_t) + r(X_t) - g^*(S_t)).$$

Since $\operatorname{sp}\left( h^{*2} \right) \leq \operatorname{sp}(h^*)^2$, we get:

$$\sum_{t=0}^{T-1} \mathbf{V}(p(X_t), h^*) \leq \sum_{t=0}^{T-1} (p(X_t) - e_{S_t}) h^{*2} + 2 \operatorname{sp}(h^*) \left( \operatorname{sp}(r) T + \sum_{t=0}^{T-1} \Delta^*(X_t) \right)$$

$$= \sum_{t=0}^{T-1} (p(X_t) - e_{S_{t+1}}) h^{*2} + 2 \operatorname{sp}(h^*) \left( \tfrac{1}{2} \operatorname{sp}(h^*) \operatorname{sp}(r) T + \sum_{t=0}^{T-1} \Delta^*(X_t) \right)$$

$$\text{(Lemma 32)} \leq 2 \operatorname{sp}(h^*) \operatorname{sp}(r) T + \operatorname{sp}(h^*)^2 \sqrt{\tfrac{1}{2} T \log \left( \tfrac{1}{\delta} \right)} + 2 \operatorname{sp}(h^*) \sum_{t=0}^{T-1} \Delta^*(X_t) + \operatorname{sp}(h^*)^2$$

where the last inequality holds with probability $1 - \delta$. This concludes the proof. $\square$

### C.3 Regret and pseudo-regret: A tight relation

In this paragraph, we bound the regret with respect to the pseudo-regret (and conversely) up to a factor of order $(\operatorname{sp}(h^*) \operatorname{sp}(r) \log(\tfrac{T}{\delta}))^{1/2}$. Hence, in proofs, the pseudo-regret can be changed to the regret with ease.

**Lemma 30.** *With probability $1 - 4\delta$, the regret and the pseudo-regret and linked as follows:*

$$\left| \sum_{t=0}^{T-1} (g^* - R_t) - \sum_{t=0}^{T-1} \Delta^*(X_t) \right| \leq \left\{ \begin{aligned} & 2 \sqrt{\left( 2 \operatorname{sp}(h^*) \operatorname{sp}(r) + \tfrac{1}{8} \right) T \log \left( \tfrac{T}{\delta} \right)} + \sqrt{2 \operatorname{sp}(h^*) \log \left( \tfrac{T}{\delta} \right) \sum_{t=0}^{T-1} \Delta^*(X_t)} \\ & + \operatorname{sp}(h^*) \left( \tfrac{1}{2} T \right)^{\frac{1}{4}} \log^{\frac{3}{4}} \left( \tfrac{T}{\delta} \right) + 4 \operatorname{sp}(h^*) \log \left( \tfrac{T}{\delta} \right) + 2 \operatorname{sp}(h^*) \end{aligned} \right\}.$$

*Proof.* We rely again on the Poisson equation $g^*(S_t) - r(X_t) - \Delta^*(X_t) = (p(X_t) - e_{S_t}) h^*$, so:

$$A := \left| \sum_{t=0}^{T-1} (g^* - R_t - \Delta^*(X_t)) \right| \leq \left| \sum_{t=0}^{T-1} (p(X_t) - e_{S_t}) h^* \right| + \left| \sum_{t=0}^{T-1} (R_t - r(X_t)) \right|$$

$$\leq \operatorname{sp}(h^*) + \left| \sum_{t=0}^{T-1} (p(X_t) - e_{S_{t+1}}) h^* \right| + \left| \sum_{t=0}^{T-1} (R_t - r(X_t)) \right|.$$

Up to the constant $\mathrm{sp}\,(h^*)$, the two error terms are respectively a navigation and a reward error. The second is bounded using Azuma's inequality (Lemma 32), showing that with probability $1 - 2\delta$, we have:

$$\left|\sum_{t=0}^{T-1}(R_t - r(X_t))\right| \leq \sqrt{\tfrac{1}{2}T\log\left(\tfrac{1}{\delta}\right)}.$$

We continue by using Freedman's inequality, instantiated in the form of Lemma 33. With probability $1 - \delta$, we have:

$$\left|\sum_{t=0}^{T-1}\left(p(X_t) - e_{S_{t+1}}\right)h^*\right| \leq \sqrt{2\sum_{t=0}^{T-1}\mathbf{V}(p(X_t),h^*)\log\left(\tfrac{T}{\delta}\right)} + 4\mathrm{sp}\,(h^*)\log\left(\tfrac{T}{\delta}\right).$$

The quantity $\sum_{t=0}^{T-1}\mathbf{V}(p(X_t),h^*)$ is a classical one that appears at several places throughout the analysis. Using Lemma 29, we bount it explicitly. Further simplifying the bound with $\sqrt{a+b} \leq \sqrt{a} + \sqrt{b}$, we get that with probability $1 - 4\delta$, we have:

$$A \leq \left\{\begin{array}{c}\sqrt{2\mathrm{sp}\,(h^*)\mathrm{sp}\,(r)T\log\left(\tfrac{T}{\delta}\right)} + \sqrt{\tfrac{1}{2}T\log\left(\tfrac{1}{\delta}\right)} + \sqrt{2\mathrm{sp}\,(h^*)\log\left(\tfrac{T}{\delta}\right)\sum_{t=0}^{T-1}\Delta^*(X_t)} \\ +\mathrm{sp}\,(h^*)\left(\tfrac{1}{2}T\right)^{\frac{1}{4}}\log^{\frac{3}{4}}\left(\tfrac{T}{\delta}\right) + 4\mathrm{sp}\,(h^*)\log\left(\tfrac{T}{\delta}\right) + 2\mathrm{sp}\,(h^*)\end{array}\right\}.$$

Bound $\log(\tfrac{1}{\delta})$ by $\log(\tfrac{T}{\delta})$ and use $\sqrt{a} + \sqrt{b} \leq 2\sqrt{a+b}$ to merge the terms in $\sqrt{T\log(\tfrac{T}{\delta})}$ under a single square-root. □

Overall, Lemma 30 states that the regret $\sum_{t=0}^{T-1}(g^* - R_t)$ and the pseudo-regret $\sum_{t=0}^{T-1}\Delta^*(X_t)$ differ by about $(\mathrm{sp}\,(h^*)T\log(\tfrac{T}{\delta}))^{1/2}$ in probability (up to asymptotically negligible additional terms). In general, the precise form of Lemma 30 is not convenient to use because it is of form form $x \leq y + \alpha\sqrt{y} + \beta$ that is not linear in $y$. Corollary 31 factorizes the result into one which will be more convenient in proofs.

**Corollary 31.** *Denote* $x := \sum_{t=0}^{T-1}(g^* - R_t)$ *and* $y := \sum_{t=0}^{T-1}\Delta^*(X_t)$. *Further introduce:*

$$\alpha := \sqrt{2\mathrm{sp}\,(h^*)\log\left(\tfrac{T}{\delta}\right)}$$

$$\beta := 2\sqrt{\left(2\mathrm{sp}\,(h^*)\mathrm{sp}\,(r) + \tfrac{1}{2}\right)T\log\left(\tfrac{T}{\delta}\right)} + \mathrm{sp}\,(h^*)\left(\tfrac{1}{2}T\right)^{\frac{1}{4}}\log^{\frac{3}{4}}\left(\tfrac{T}{\delta}\right) + 2\mathrm{sp}\,(h^*)\left(2\log\left(\tfrac{T}{\delta}\right) + 1\right).$$

*Then, with probability* $1 - 4\delta$, *we have* $\sqrt{x} \leq \sqrt{y} + \tfrac{1}{2}\alpha + \sqrt{\beta}$ *and* $\sqrt{y} \leq \sqrt{x} + \alpha + \sqrt{\beta}$.

*Proof.* This is straight forward algebra from the result of Lemma 30. □

### C.4 Proof of Lemma 6, reward optimism

We start by getting rid of the reward noise. We have:

$$\begin{aligned}\mathrm{Reg}(T) &:= \sum_{t=0}^{T-1}(g^* - R_t) = \sum_{t=0}^{T-1}(g^* - r(X_t)) + \sum_{t=0}^{T-1}(r(X_t) - R_t) \\ &\leq \sum_{t=0}^{T-1}(g^* - r(X_t)) + \sqrt{\tfrac{1}{2}T\log\left(\tfrac{1}{\delta}\right)}\end{aligned}$$

with probability $1 - \delta$ by Azuma's inequality (Lemma 32). We are left with $\sum_{t=0}^{T-1}(g^* - r(X_t))$. We continue by splitting the regret episodically and invoking optimism. By Lemma 13, with probability $1 - 4\delta$, we have $\sum_{t=0}^{T-1}(g^* - r(X_t)) \leq \sum_k \sum_{t=t_k}^{t_{k+1}-1}(g_k - r(X_t))$. Introduce

$$B_0(T) := \sum_k \sum_{t=t_k}^{t_{k+1}-1}(g_k - r(X_t)). \tag{20}$$

We focus on bounding $B_0(T)$. By Assumption 2, $\tilde{r}_k(s,a)$ is of the form $\hat{r}_k(s,a) + \sqrt{C\log(2SAT/\delta)/N_{t_k}(s,a)} - \eta_k(s,a)$ with $\eta_k(s,a) \in \mathbf{R}$. By the statement (3) of Proposition 2, $\eta_k(s,a) \geq 0$. Therefore,

$$B_0(T) = \sum_k \sum_{t=t_k}^{t_{k+1}-1}(g_k - \tilde{r}_k(X_t)) + \sum_k \sum_{t=t_k}^{t_{k+1}-1}(\tilde{r}_k(X_t) - r(X_t))$$

$$\leq \sum_k \sum_{t=t_k}^{t_{k+1}-1} (\mathfrak{g}_k - \tilde{r}_k(X_t)) + SA + \sum_k \sum_{t=t_k}^{t_{k+1}-1} \mathbf{1}(N_{t_k}(X_t) \geq 1) \left( \hat{r}_k(X_t) - r(X_t) + \sqrt{\frac{C \log\left(\frac{2SAT}{\delta}\right)}{N_{t_k}(X_t)}} \right)$$

$$\overset{(*)}{\leq} \sum_k \sum_{t=t_k}^{t_{k+1}-1} (\mathfrak{g}_k - \tilde{r}_k(X_t)) + SA + \sum_k \sum_{t=t_k}^{t_{k+1}-1} \mathbf{1}(N_{t_k}(X_t) \geq 1) \left( \sqrt{\frac{2 \log\left(\frac{2SAT}{\delta}\right)}{N_{t_k}(s,a)}} + \sqrt{\frac{C \log\left(\frac{2SAT}{\delta}\right)}{N_{t_k}(s,a)}} \right)$$

where $(*)$ holds with probability $1 - \delta$ following Lemma 35. By the doubling trick rule (DT), we have $N_t(X_t) \leq 2N_{t_k}(X_t)$ for $t < t_{k+1}$, so, with probability $1 - \delta$,

$$B_0(T) \leq \sum_k \sum_{t=t_k}^{t_{k+1}-1} (\mathfrak{g}_k - \tilde{r}_k(X_t)) + SA + 2 \sum_k \sum_{t=t_k}^{t_{k+1}-1} \mathbf{1}(N_{t_k}(X_t) \geq 1) \sqrt{\frac{(2+C) \log\left(\frac{2SAT}{\delta}\right)}{N_{t_k}(s,a)}}$$

$$\leq \sum_k \sum_{t=t_k}^{t_{k+1}-1} (\mathfrak{g}_k - \tilde{r}_k(X_t)) + SA + 2 \sqrt{(2+C) \log\left(\frac{2SAT}{\delta}\right)} \sum_{s,a} \sum_{n=1}^{N_T(s,a)-1} \sqrt{\frac{1}{n}}$$

$$\leq \sum_k \sum_{t=t_k}^{t_{k+1}-1} (\mathfrak{g}_k - \tilde{r}_k(X_t)) + SA + 4 \sqrt{(2+C) \log\left(\frac{2SAT}{\delta}\right)} \sum_{s,a} \sqrt{N_T(s,a)}$$

$$(\text{Jensen}) \leq \sum_k \sum_{t=t_k}^{t_{k+1}-1} (\mathfrak{g}_k - \tilde{r}_k(X_t)) + SA + 4 \sqrt{(2+C)SAT \log\left(\frac{2SAT}{\delta}\right)}.$$

We conclude that with probability $1 - 6\delta$, we have:

$$\text{Reg}(T) \leq \sum_k \sum_{t=t_k}^{t_{k+1}-1} (\mathfrak{g}_k - \tilde{r}_k(X_t)) + 4 \sqrt{(2+C)SAT \log\left(\frac{2SAT}{\delta}\right)} + \sqrt{\frac{1}{2}T \log\left(\frac{2SAT}{\delta}\right)} + SA. \qquad (21)$$

This concludes the proof. $\qquad \square$

## C.5  Proof of Lemma 7, navigation error

We have:

$$\sum_k \sum_{t=t_k}^{t_{k+1}-1} (p_k(S_t) - e_{S_t})\mathfrak{h}_k \leq \sum_k \sum_{t=t_k}^{t_{k+1}-1} (p_k(S_t) - e_{S_{t+1}})\mathfrak{h}_k + \sum_k \text{sp}(\mathfrak{h}_k)$$

$$\leq \underbrace{\sum_k \sum_{t=t_k}^{t_{k+1}-1} (p_k(S_t) - e_{S_{t+1}})(\mathfrak{h}_k - h^*)}_{A_1} + \underbrace{\sum_k \sum_{t=t_k}^{t_{k+1}-1} (p_k(S_t) - e_{S_{t+1}})h^*}_{A_2} + \sum_k \text{sp}(\mathfrak{h}_k).$$

The last term is $O(c_0 SA \log(T))$ by Lemma 28, hence is $O(T^{1/5} \log(T))$.

(**STEP 1**) We start by bounding $A_1$. By Lemma 13, with probability $1 - 4\delta$, we have $h^* \in \mathcal{H}_{t_k}$ for all $k \leq K(T)$. So $\text{sp}(\mathfrak{h}_k - h^*) \leq \text{sp}(\mathfrak{h}_k) + \text{sp}(h^*) \leq 2c_0$. By Freedman's inequality invoked in the form of Lemma 33, we have with probability $1 - 5\delta$,

$$A_1 \leq \sqrt{2 \sum_k \sum_{t=t_k}^{t_{k+1}-1} \mathbf{V}(p(X_t), \mathfrak{h}_k - h^*) \log\left(\frac{T}{\delta}\right)} + 8c_0 \log\left(\frac{T}{\delta}\right)$$

It suffices to bound the first term. Recall that $e$ is the vector full of ones. We have:

$$\sum_k \sum_{t=t_k}^{t_{k+1}-1} \mathbf{V}(p(X_t), \mathfrak{h}_k - h^*) = \sum_k \sum_{t=t_k}^{t_{k+1}-1} \mathbf{V}(p(X_t), \mathfrak{h}_k - h^* - (\mathfrak{h}_k(S_t) - h^*(S_t)) \cdot e)$$

$$\leq \sum_k \sum_{t=t_k}^{t_{k+1}-1} \sum_{s' \in \mathcal{S}} p(s'|X_t)(\mathfrak{h}_k(s') - h^*(s') - (\mathfrak{h}_k(S_t) - h^*(S_t)))^2$$

$$\overset{(*)}{\leq} 3 \sum_{k} \sum_{t=t_k}^{t_{k+1}-1} \mathbf{E}\left[\sum_{s' \in \mathcal{S}} p(s'|X_t) \left(\mathfrak{h}_k(s') - h^*(s') - (\mathfrak{h}_k(S_t) - h^*(S_t))\right)^2 \middle| \mathcal{F}_t\right] + 16c_0^2 \log\left(\tfrac{1}{\delta}\right)$$

$$= 3 \sum_{k} \sum_{t=t_k}^{t_{k+1}-1} \left(\mathfrak{h}_k(S_{t+1}) - h^*(S_{t+1}) - (\mathfrak{h}_k(S_t) - h^*(S_t))\right)^2 + 16c_0^2 \log\left(\tfrac{1}{\delta}\right).$$

Here the inequality $(*)$ holds with probability $1 - \delta$ following Lemma 40. We will bound the summand with the bias estimation error error$(c_k, s, s')$ that spawns the inner regret estimation $B_0(t_k) = \sum_{\ell=1}^{k-1} \sum_{t=t_\ell}^{t_{\ell+1}-1} (\mathfrak{g}_\ell - R_t)$. This inner estimation is linked to $B(T) := \sum_{k,t} (\mathfrak{g}_k - R_t)$ the overall optimistic regret by:

$$B_0(t_k) \leq \sum_{\ell=1}^{K(T)} \sum_{t=t_\ell}^{t_{\ell+1}-1} (\mathfrak{g}_k - R_t) - \sum_{\ell=k}^{K(T)} \sum_{t=t_\ell}^{t_{\ell+1}-1} (\mathfrak{g}_k - R_t)$$

$$\overset{(*)}{\leq} \sum_{\ell=1}^{K(T)} \sum_{t=t_\ell}^{t_{\ell+1}-1} (\mathfrak{g}_k - R_t) - \sum_{\ell=k}^{K(T)} \sum_{t=t_\ell}^{t_{\ell+1}-1} (g^* - R_t)$$

$$\leq \sum_{\ell=1}^{K(T)} \sum_{t=t_\ell}^{t_{\ell+1}-1} (\mathfrak{g}_k - R_t) - \sum_{\ell=k}^{K(T)} \sum_{t=t_k}^{T-1} \left(\Delta^*(X_t) + (p(X_t) - e_{S_t}) h^* + r(X_t) - R_t\right)$$

$$\leq \sum_{\ell=1}^{K(T)} \sum_{t=t_\ell}^{t_{\ell+1}-1} (\mathfrak{g}_k - R_t) + \mathrm{sp}\,(h^*) - \sum_{\ell=k}^{K(T)} \sum_{t=t_k}^{T-1} \left((p(X_t) - e_{S_{t+1}}) h^* + r(X_t) - R_t\right)$$

$$\overset{(\dagger)}{\leq} \sum_{\ell=1}^{K(T)} \sum_{t=t_\ell}^{t_{\ell+1}-1} (\mathfrak{g}_k - R_t) + \mathrm{sp}\,(h^*) + (1 + \mathrm{sp}\,(h^*)) \sqrt{\tfrac{1}{2} T \log\left(\tfrac{1}{\delta}\right)}$$

$$=: B(T) + \mathrm{sp}\,(h^*) + (1 + \mathrm{sp}\,(h^*)) \sqrt{\tfrac{1}{2} T \log\left(\tfrac{1}{\delta}\right)}.$$

In the above, $(*)$ holds with probability $1 - 4\delta$ uniformly on $k$ following Lemma 13 and $(\dagger)$ holds, also uniformly on $k$, with probability $1 - \delta$ by applying Azuma-Hoeffding's inequality (Lemma 32). Continuing, still on the event specified by Lemma 13, we have with probability $1 - 6\delta$:

$$\sum_{k} \sum_{t=t_k}^{t_{k+1}-1} \mathbf{V}(p(X_t), \mathfrak{h}_k - h^*) \leq 3 \sum_{k} \sum_{t=t_k}^{t_{k+1}-1} \frac{3c_0 + (1 + c_0)\sqrt{8t_k \log\left(\tfrac{2}{\delta}\right)} + 2B_0(t_k)}{N_{t_k}(S_{t+1} \leftrightarrow S_t)} + 16c_0^2 \log\left(\tfrac{1}{\delta}\right)$$

$$\leq 3 \sum_{k} \sum_{t=t_k}^{t_{k+1}-1} \frac{4c_0 + (1 + c_0)\sqrt{32T \log\left(\tfrac{2}{\delta}\right)} + 2B(T)}{N_{t_k}(S_t, A_t, S_{t+1})} + 16c_0^2 \log\left(\tfrac{1}{\delta}\right)$$

$$(\mathrm{DT}) \leq 12c_0^2 S^2 A + 3\left(4c_0 + (1 + c_0)\sqrt{32T \log\left(\tfrac{2}{\delta}\right)} + 2B(T)\right) S^2 A \log(T)$$

$$+ 16c_0^2 \log\left(\tfrac{1}{\delta}\right).$$

(**STEP 2**) For $A_2$, by Freedman's inequality invoked in the form of Lemma 33 again, we have with probability $1 - \delta$,

$$A_2 \leq \sqrt{2 \sum_{k} \sum_{t=t_k}^{t_{k+1}-1} \mathbf{V}(p_k(S_t), h^*) \log\left(\tfrac{T}{\delta}\right) + 8c_0 \log\left(\tfrac{T}{\delta}\right)}$$

$$\leq \sqrt{2 \sum_{t=0}^{T-1} \mathbf{V}(p(X_t), h^*) \log\left(\tfrac{T}{\delta}\right) + 8c_0 \log\left(\tfrac{T}{\delta}\right)}.$$

We recognize the sum of variance $\sum_{t=0}^{T-1} \mathbf{V}(p(X_t), h^*)$ that we leave as is.

(**STEP 3**) As a result, with probability $1 - 7\delta$, we have:

$$\sum_{k} \sum_{t=t_k}^{t_{k+1}-1} (p_k(S_t) - e_{S_t}) \mathfrak{h}_k \leq \sqrt{2 \sum_{t=0}^{T-1} \mathbf{V}(p(X_t), h^*) \log\left(\tfrac{T}{\delta}\right)} + 2SA^{\frac{1}{2}} \sqrt{3B(T)} \log\left(\tfrac{T}{\delta}\right) + \mathrm{O}\left(SA^{\frac{1}{2}} T^{\frac{7}{20}} \log^{\frac{3}{4}}\left(\tfrac{T}{\delta}\right)\right)$$

when $c_0 = T^{\frac{1}{5}}$. $\qquad\square$

## C.6 Proof of Lemma 8, empirical bias error

Because $h^*$ is a fixed vector, Bennett's inequality (see Lemma 39) guarantees that $(\hat{p}_k(S_t) - p_k(S_t))h^*$ is small as follows. By doing a union bound over Lemma 39 with confidence $\frac{\delta}{SAT}$ over all pairs $(s, a)$ and visits counts $N(s, a) \leq T$, we see that with probability $1 - \delta$, for all $k$, we have:

$$\sum_{t=t_k}^{t_{k+1}-1} (\hat{p}_k(S_t) - p_k(S_t)) h^* \leq \mathrm{sp}(h^*)SA + \sum_{t=t_k}^{t_{k+1}-1} \mathbf{1}\left(N_{t_k}(X_t) \geq 1\right)\left(\sqrt{\frac{2\mathbf{V}(p(X_t),h^*)\log\left(\frac{SAT}{\delta}\right)}{N_{t_k}(X_t)}} + \frac{\mathrm{sp}(h^*)\log\left(\frac{SAT}{\delta}\right)}{3N_{t_k}(X_t)}\right)$$

$$\text{(by doubling trick)} \leq \mathrm{sp}(h^*)SA + 2\sum_{t=t_k}^{t_{k+1}-1} \mathbf{1}\left(N_t(X_t) \geq 1\right)\left(\sqrt{\frac{2\mathbf{V}(p(X_t),h^*)\log\left(\frac{SAT}{\delta}\right)}{N_t(X_t)}} + \frac{\mathrm{sp}(h^*)\log\left(\frac{SAT}{\delta}\right)}{3N_t(X_t)}\right).$$

Summing this over $k$ and factorizing over state-action pairs, we get that with probability $1 - \delta$,

$$\sum_k (2k) \leq \mathrm{sp}(h^*)SA + 2\sum_{s,a}\left(\sum_{n=1}^{N_T(s,a)} \sqrt{\frac{2\mathbf{V}(p(s,a),h^*)\log\left(\frac{SAT}{\delta}\right)}{n}} + \sum_{n=1}^{N_T(s,a)} \frac{\mathrm{sp}(h^*)\log\left(\frac{SAT}{\delta}\right)}{n}\right)$$

$$\leq \mathrm{sp}(h^*)SA + 4\sum_{s,a}\sqrt{N_T(s,a)\mathbf{V}(p(s,a),h^*)\log\left(\frac{SAT}{\delta}\right)} + 2\mathrm{sp}(h^*)SA\log\left(\frac{SAT}{\delta}\right)\log(T)$$

$$\text{(Jensen)} \leq \mathrm{sp}(h^*)SA + 4\sqrt{SA\sum_{s,a}\mathbf{V}(p(s,a),h^*)\log\left(\frac{SAT}{\delta}\right)} + 2\mathrm{sp}(h^*)SA\log\left(\frac{SAT}{\delta}\right)\log(T)$$

$$= \mathrm{sp}(h^*)SA + 4\sqrt{\sum_{t=0}^{T-1}\mathbf{V}(p(X_t),h^*)\log\left(\frac{SAT}{\delta}\right)} + 2\mathrm{sp}(h^*)SA\log\left(\frac{SAT}{\delta}\right)\log(T)$$

We recognize the sum of variances $\sum_{t=0}^{T-1}\mathbf{V}(p(X_t),h^*)$, that is left to be upper-bounded later on. □

## C.7 Proof of Lemma 9, optimistic overshoot

Because of the $\beta$-mitigation generated by Algorithm 5, the quantity $(\tilde{p}_k(S_t) - \hat{p}_k(S_t))\mathfrak{h}_k$ is shown to be directly related to $\mathbf{V}(p(X_t), h^*)$ up to a provably negligible error. Denote $h'_k$ the reference point $\texttt{BiasProjection}(\mathcal{H}_{t_k}, c_{t_k}(-, s_0))$ used in Algorithm 5 (denoted $h_0$ in the algorithm). By Lemma 13, with probability $1 - 4\delta$, we have $h^* \in \mathcal{H}_{t_k}$ for all $k$. To lighten up notations, we write $d_{t_k}(s', s)$ instead of $\mathrm{error}(c_{t_k}, s', s)$.

(**STEP 1**) Denote $\mathrm{A} := (\tilde{p}_k(S_t) - \hat{p}_k(S_t))\mathfrak{h}_k$. By construction of $\tilde{p}_k$, we have $\mathrm{A} \leq \beta_{t_k}(X_t)$, so:

$$\mathrm{A} \leq \beta_{t_k}(X_t)$$

$$=: \sqrt{\frac{2\left(\mathbf{V}(\hat{p}_k(S_t), h'_k) + 8c_0\sum_{s'\in\mathcal{S}}\hat{p}_k(s'|S_t)d_{t_k}(s', S_t)\log\left(\frac{SAT}{\delta}\right)\right)}{N_{t_k}(X_t)}} + \frac{3c_0\log\left(\frac{SAT}{\delta}\right)}{N_{t_k}(X_t)}$$

$$\leq \underbrace{\sqrt{\frac{2\mathbf{V}(\hat{p}_k(S_t), h'_k)}{N_{t_k}(X_t)}}}_{\mathrm{A}_1} + \underbrace{\sqrt{\frac{16c_0\sum_{s'\in\mathcal{S}}\hat{p}_k(s'|S_t)d_{t_k}(s', S_t)\log\left(\frac{SAT}{\delta}\right)}{N_{t_k}(X_t)}}}_{\mathrm{A}_2} + \frac{3c_0\log\left(\frac{SAT}{\delta}\right)}{N_{t_k}(X_t)}.$$

The rightmost term of $\mathrm{A}$ is of order $O(\log^2(T))$ hence is negligible. We focus on the other two. The analysis of $\mathrm{A}_1$ will spawn a term similar to $\mathrm{A}_2$, hence we start by the second. Recall that $d_{t_k}$ is the bias error provided by Algorithm 3 and that the inner regret estimation is $B_0(t_k) = \sum_{\ell=1}^{k-1}\sum_{t=t_\ell}^{t_{\ell+1}-1}(\mathfrak{g}_\ell - R_t)$. Now, remark that:

$$B_0(t_k) \leq \sum_{\ell=1}^{K(T)}\sum_{t=t_\ell}^{t_{\ell+1}-1}(\mathfrak{g}_k - R_t) - \sum_{\ell=k}^{K(T)}\sum_{t=t_\ell}^{t_{\ell+1}-1}(\mathfrak{g}_k - R_t)$$

$$\overset{(*)}{\leq} \sum_{\ell=1}^{K(T)}\sum_{t=t_\ell}^{t_{\ell+1}-1}(\mathfrak{g}_k - R_t) - \sum_{\ell=k}^{K(T)}\sum_{t=t_\ell}^{t_{\ell+1}-1}(g^* - R_t)$$

$$\leq \sum_{\ell=1}^{K(T)}\sum_{t=t_\ell}^{t_{\ell+1}-1}(\mathfrak{g}_k - R_t) - \sum_{\ell=k}^{K(T)}\sum_{t=t_k}^{T-1}(\Delta^*(X_t) + (p(X_t) - e_{S_t})h^* + r(X_t) - R_t)$$

$$\leq \sum_{\ell=1}^{K(T)}\sum_{t=t_\ell}^{t_{\ell+1}-1}(\mathfrak{g}_k - R_t) + \mathrm{sp}(h^*) - \sum_{\ell=k}^{K(T)}\sum_{t=t_k}^{T-1}((p(X_t) - e_{S_{t+1}})h^* + r(X_t) - R_t)$$

$$\overset{(\dagger)}{\leq} \sum_{\ell=1}^{K(T)} \sum_{t=t_\ell}^{t_{\ell+1}-1} (g_k - R_t) + \mathrm{sp}\,(h^*) + (1 + \mathrm{sp}\,(h^*)) \sqrt{\tfrac{1}{2}T \log\left(\tfrac{1}{\delta}\right)}$$

$$=: B(T) + \mathrm{sp}\,(h^*) + (1 + \mathrm{sp}\,(h^*)) \sqrt{\tfrac{1}{2}T \log\left(\tfrac{1}{\delta}\right)}.$$

In the above, $(*)$ holds with probability $1 - 4\delta$ uniformly on $k$ following Lemma 13 and $(\dagger)$ holds, also uniformly on $k$, with probability $1 - \delta$ by applying Azuma-Hoeffding's inequality (Lemma 32). Therefore, with probability $1 - 5\delta$, for all $k$ and $t \in \{t_k, \ldots, t_{k+1} - 1\}$, we have:

$$\sqrt{\frac{16c_0 \sum_{s' \in \mathcal{S}} \hat{p}_k(s'|S_t) d_{t_k}(s', S_t) \log\left(\frac{SAT}{\delta}\right)}{N_{t_k}(X_t)}} \leq \frac{\sqrt{16c_0 \log\left(\frac{SAT}{\delta}\right) \sum_{s' \in \mathcal{S}} N_{t_k}(S_t, A_t, s') d_{t_k}(s', S_t)}}{N_{t_k}(X_t)}$$

$$\leq \frac{\sqrt{16c_0 \log\left(\frac{SAT}{\delta}\right) \sum_{s' \in \mathcal{S}} N_{t_k}(S_t \leftrightarrow s') d_{t_k}(s', S_t)}}{N_{t_k}(X_t)}$$

$$\leq \frac{\sqrt{16c_0 \log\left(\frac{SAT}{\delta}\right) S\left(3c_0 + (1+c_0)\left(1 + \sqrt{8T \log\left(\frac{2}{\delta}\right)}\right) + 2B_0(t_k)\right)}}{N_{t_k}(X_t)}$$

$$\leq \frac{\sqrt{16c_0 \log\left(\frac{SAT}{\delta}\right) S\left((1+c_0)\left(3 + 2\sqrt{8T \log\left(\frac{2}{\delta}\right)}\right) + 2B(T)\right)}}{N_{t_k}(X_t)}$$

$$\leq \frac{\sqrt{16c_0 \log\left(\frac{SAT}{\delta}\right) S\left((1+c_0)\left(3 + 2\sqrt{8T \log\left(\frac{2}{\delta}\right)} + 2B(T)\right)\right)}}{N_{t_k}(X_t)}.$$

This bound will be enough. We move on to $A_1$. We have:

$$\sqrt{\mathbf{V}(\hat{p}_k(S_t), h'_k)} \leq \sqrt{\left|\mathbf{V}(\hat{p}_k(S_t), h'_k) - \mathbf{V}(p(X_t), h^*)\right|} + \sqrt{\mathbf{V}(p(X_t), h^*)}$$

$$\leq \sqrt{\left|\mathbf{V}(\hat{p}_k(S_t), h'_k) - \mathbf{V}(\hat{p}_k(X_t), h^*)\right|} \sqrt{\left|\mathbf{V}(\hat{p}_k(S_t), h^*) - \mathbf{V}(p(X_t), h^*)\right|} + \sqrt{\mathbf{V}(p(X_t), h^*)}$$

$$\overset{(*)}{\leq} \sqrt{8c_0 \sum_{s' \in \mathcal{S}} \hat{p}_k(s'|S_t) d_k(s', S_t)} + \mathrm{sp}\,(h^*)\sqrt{\|\hat{p}_k(S_t) - p_k(S_t)\|_1} + \sqrt{\mathbf{V}(p(X_t), h^*)}$$

$$\overset{(\dagger)}{\leq} \sqrt{8c_0 \sum_{s' \in \mathcal{S}} \hat{p}_k(s'|S_t) d_k(s', S_t)} + \mathrm{sp}\,(h^*)\left(\frac{S \log\left(\frac{SAT}{\delta}\right)}{N_{t_k}(X_t)}\right)^{\frac{1}{4}} + \sqrt{\mathbf{V}(p(X_t), h^*)}$$

$$\leq \frac{A_2}{\sqrt{2N_{t_k}(X_t)}} + \mathrm{sp}\,(h^*)\left(\frac{S \log\left(\frac{SAT}{\delta}\right)}{N_{t_k}(X_t)}\right)^{\frac{1}{4}} + \sqrt{\mathbf{V}(p(X_t), h^*)}$$

where $(*)$ is obtained by applying Lemma 12 and $(\dagger)$ holds with probability $1 - \delta$ by applying Weissman's inequality, see Lemma 35. All together, with probability $1 - 6\delta$, A is upper-bounded by:

$$A \leq \sqrt{\frac{2\mathbf{V}(p(X_t), h^*) \log\left(\frac{SAT}{\delta}\right)}{N_{t_k}(X_t)}} + 2A_2 + \mathrm{sp}\,(h^*)\underbrace{\sqrt{\frac{2\log\left(\frac{SAT}{\delta}\right)\sqrt{S \log \frac{SAT}{\delta}}}{N_{t_k}(X_t)\sqrt{N_{t_k}(X_t)}} + \frac{3c_0 \log\left(\frac{SAT}{\delta}\right)}{N_{t_k}(X_t)}}}_{A_3(k,t)}.$$

(**STEP 2**) The number of visits $N_k(X_t)$ is lower-bounded by $\frac{1}{2}N_t(X_t)$ when $N_k(X_t) \geq 1$ by doubling trick (DT). By summing over $t$ and $k$, we find that with probability $1 - 6\delta$,

$$\sum_k (3k) \leq SAc_0 + \sum_k \sum_{t=t_k}^{t_{k+1}-1} \mathbf{1}_{N_{t_k}(X_t) \geq 1} \sqrt{\frac{2\mathbf{V}(p(X_t), h^*) \log\left(\frac{SAT}{\delta}\right)}{N_{t_k}(X_t)}} + \sum_k \sum_{t=t_k}^{t_{k+1}-1} \mathbf{1}_{N_{t_k}(X_t) \geq 1}(2A_2(k,t) + A_3(k,t))$$

$$(DT) \leq SAc_0 + 2\sum_k \sum_{t=t_k}^{t_{k+1}-1} \mathbf{1}_{N_{t_k}(X_t) \geq 1} \sqrt{\frac{2\mathbf{V}(p(X_t), h^*) \log\left(\frac{SAT}{\delta}\right)}{N_t(X_t)}} + \sum_k \sum_{t=t_k}^{t_{k+1}-1} \mathbf{1}_{N_{t_k}(X_t) \geq 1}(2A_2(k,t) + A_3(k,t))$$

$$\leq SAc_0 + 4\sqrt{2SA \sum_{t=0}^{T-1} \mathbf{V}(p(X_t), h^*) \log\left(\frac{SAT}{\delta}\right)} + \sum_k \sum_{t=t_k}^{t_{k+1}-1} \mathbf{1}_{N_{t_k}(X_t) \geq 1}(2A_2(k,t) + A_3(k,t))$$

where the last inequality is obtained with computations that are similar to those detailed in the proof of Lemma 8. We recognize the variance that we will leave as is. We finish the proof by bounding the lower order terms $A_2$ and $A_3$.

(**STEP 3**) We start with $A_2$. We have:

$$\sum_k \sum_{t=t_k}^{t_{k+1}-1} \mathbf{1}_{N_{t_k}(X_t)\geq 1} A_2(k,t) := \sum_k \sum_{t=t_k}^{t_{k+1}-1} \mathbf{1}_{N_{t_k}(X_t)\geq 1} \frac{\sqrt{16c_0 \log\left(\frac{SAT}{\delta}\right) S\left((1+c_0)\left(3+2\sqrt{8T\log\left(\frac{2}{\delta}\right)}+2B(T)\right)\right)}}{N_{t_k}(X_t)}$$

$$(DT) \leq 2\sqrt{16c_0 S \log\left(\frac{SAT}{\delta}\right)\left((1+c_0)\left(3+2\sqrt{8T\log\left(\frac{2}{\delta}\right)}+2B(T)\right)\right) SA \log(T)}$$

$$\leq 8(1+c_0)S^{\frac{3}{2}}A \log^{\frac{3}{2}}\left(\frac{SAT}{\delta}\right)\left(2+4T^{\frac{1}{4}}\log^{\frac{1}{4}}\left(\frac{SAT}{\delta}\right)+\sqrt{2B(T)}\right).$$

(**STEP 4**) We are left with $A_3$. We have:

$$\sum_k \sum_{t=t_k}^{t_{k+1}-1} \mathbf{1}_{N_{t_k}(X_t)\geq 1} A_3(k,t) := \sum_k \sum_{t=t_k}^{t_{k+1}-1} \mathbf{1}_{N_{t_k}(X_t)\geq 1} \left( \mathrm{sp}\,(h^*)\sqrt{\frac{2\log\left(\frac{SAT}{\delta}\right)\sqrt{S\log\frac{SAT}{\delta}}}{N_{t_k}(X_t)\sqrt{N_{t_k}(X_t)}}} + \frac{3c_0\log\left(\frac{SAT}{\delta}\right)}{N_{t_k}(X_t)} \right)$$

$$(DT) \leq \sum_k \sum_{t=t_k}^{t_{k+1}-1} \mathbf{1}_{N_{t_k}(X_t)\geq 1} \left( \mathrm{sp}\,(h^*)\sqrt{\frac{2\log\left(\frac{SAT}{\delta}\right)\sqrt{S\log\frac{SAT}{\delta}}}{N_{t_k}(X_t)\sqrt{N_{t_k}(X_t)}}} + \frac{3c_0\log\left(\frac{SAT}{\delta}\right)}{N_{t_k}(X_t)} \right)$$

$$\leq C\,\mathrm{sp}\,(h^*)S^{\frac{5}{4}}AT^{\frac{1}{4}}\log^{\frac{3}{4}}\left(\frac{SAT}{\delta}\right) + 6c_0SA\log\left(\frac{SAT}{\delta}\right)$$

$$= O\left(\mathrm{sp}\,(h^*)S^{\frac{5}{4}}AT^{\frac{1}{4}}\log\left(\frac{SAT}{\delta}\right)\right).$$

This concludes the proof. $\qquad\square$

### C.8  Proof of Lemma 10, second order error

Recall that by Lemma 13, with probability $1-4\delta$, $h^* \in \mathcal{H}_{t_k}$ for all $k$, hence $\mathrm{sp}\,(\mathfrak{h}_k - h^*) \leq 2c_0$ for all $k$ on the same event. Therefore, with probability $1-4\delta$,

$$\sum_k (4k) := 2c_0 SA + \sum_k \sum_{t=t_k}^{t_{k+1}-1} \mathbf{1}_{N_{t_k}(X_t)\geq 1} (\hat{p}_k(S_t) - p_k(S_t))(\mathfrak{h}_k - h^*)$$

$$= 2c_0 SA + \sum_k \sum_{t=t_k}^{t_{k+1}-1} \sum_{s'\in\mathcal{S}} \mathbf{1}_{N_{t_k}(X_t)\geq 1}(\hat{p}_k(s'|S_t) - p_k(s'|S_t))(\mathfrak{h}_k - h^*(s'))$$

$$\overset{(*)}{\leq} 2c_0 SA + 2\sum_k \sum_{t=t_k}^{t_{k+1}-1} \sum_{s'\in\mathcal{S}} \mathbf{1}_{N_{t_k}(X_t)\geq 1}(\hat{p}_k(s'|S_t) - p_k(s'|S_t))d_{t_k}(s',S_t)$$

$$\overset{(\dagger)}{\leq} 2c_0 SA + 2\sum_k \sum_{t=t_k}^{t_{k+1}-1} \sum_{s'\in\mathcal{S}} \mathbf{1}_{N_{t_k}(X_t)\geq 1} \left( d_k(s',S_t)\sqrt{\frac{2\hat{p}_k(s'|S_t)\log\left(\frac{S^2AT}{\delta}\right)}{N_{t_k}(X_t)}} + 3d_k(s'|S_t)\frac{\log\left(\frac{S^2AT}{\delta}\right)}{N_{t_k}(X_t)} \right)$$

$$\leq 2c_0 SA + 2\sum_k \sum_{t=t_k}^{t_{k+1}-1} \sum_{s'\in\mathcal{S}} \mathbf{1}_{N_{t_k}(X_t)\geq 1} \left( \sqrt{c_0}\sqrt{\frac{2\hat{p}_k(s'|S_t)d_k(s',S_t)\log\left(\frac{S^2AT}{\delta}\right)}{N_{t_k}(X_t)}} + \frac{3c_0\log\left(\frac{S^2AT}{\delta}\right)}{N_{t_k}(X_t)} \right)$$

$$\leq 2c_0 SA + 4\sum_k \sum_{t=t_k}^{t_{k+1}-1} \sum_{s'\in\mathcal{S}} \mathbf{1}_{N_{t_k}(X_t)\geq 1} \left( \sqrt{c_0}\sqrt{\frac{2\hat{p}_k(s'|S_t)d_k(s',S_t)\log\left(\frac{S^2AT}{\delta}\right)}{N_t(X_t)}} + \frac{3c_0\log\left(\frac{S^2AT}{\delta}\right)}{N_t(X_t)} \right)$$

where $(*)$ uses that $h^* \in \mathcal{H}_{t_k}$, and $(\dagger)$ is obtained by applying the empirical Bernstein's inequality, see Lemma 36, to $\hat{p}_k(s'|S_t) - p_k(s'|S_t)$, and holds with probability $1-\delta$. The rightmost term's sum is

upper-bounded by:

$$4 \sum_k \sum_{t=t_k}^{t_{k+1}-1} \sum_{s' \in \mathcal{S}} \frac{3c_0 \log\left(\frac{S^2 A T}{\delta}\right)}{N_t(X_t)} \leq 12 S^2 A \log(T) \log\left(\frac{S^2 A T}{\delta}\right).$$

For the other term, follow the line of the proof of Lemma 9 (term $A_2$). We have with probability $1 - 5\delta$ ($4\delta$ of which is by invoking Lemma 13):

$$\hat{p}_k(s'|S_t) d_k(s', S_t) = \frac{N_{t_k}(S_t, A_t, s')\left((1 + c_0)\left(1 + \sqrt{8t_k \log\left(\frac{2}{\delta}\right)}\right) + 2B_0(t_k)\right)}{N_{t_k}(S_t \leftrightarrow s') N_{t_k}(X_t)}$$

$$\leq \frac{\left((1 + c_0)\left(3 + 2\sqrt{8T \log\left(\frac{2}{\delta}\right)} + 2B(T)\right)\right)}{N_{t_k}(X_t)}.$$

Therefore,

$$\sqrt{c_0} \sqrt{\frac{2\hat{p}_k(s'|S_t) d_{t_k}(s', S_t) \log\left(\frac{S^2 A T}{\delta}\right)}{N_t(X_t)}} \leq \frac{4(1 + c_0)\sqrt{\left(3 + 2\sqrt{8T \log\left(\frac{2}{\delta}\right)} + 2B(T)\right)\log\left(\frac{S^2 A T}{\delta}\right)}}{N_t(X_t)}.$$

Summing over $k$, $t$, $s'$, with probability $1 - 6\delta$, we have:

$$\sum_k (4k) \leq \left\{ \begin{array}{l} 16 S^2 A (1 + c_0) \log^{\frac{1}{2}}\left(\frac{S^2 A T}{\delta}\right)\left(\sqrt{2B(T)} + 2\left(8T \log\left(\frac{2}{\delta}\right)\right)^{\frac{1}{4}}\right) \\ +32 S^2 A \left(\log(T) \log\left(\frac{S^2 A T}{\delta}\right) + (1 + c_0) \log^{\frac{1}{2}}\left(\frac{S^2 A T}{\delta}\right)\right) \end{array} \right\}$$

This concludes the proof. □

## D  More details on experiments

### D.1  River swim as a hard communicating environment

Experiments of Fig. 2 are run on $n$-states river-swim. Such MDPs are, despite their size, known to be hard to learn. They consists in $n$ states aligned in a straight line with two playable actions RIGHT and LEFT whose dynamics are given in the figure below. Rewards are Bernoulli and null everywhere excepted for $r(s_n, \text{RIGHT}) = 0.95$ and $r(s_0, \text{LEFT}) = 0.05$.

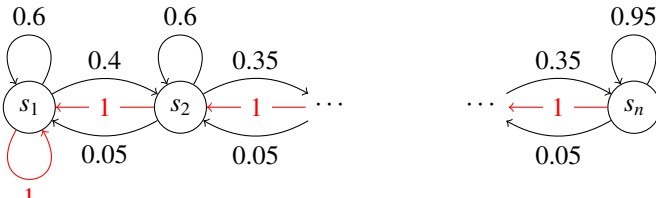

Figure 3:  The kernel of a $n$-state river-swim.

**3-state river-swim.**  The gain is $g^* \approx 0.82$ and $h^* \approx (-4.28, -2.24, 0.4)$.

**5-state river-swim.**  The gain is $g^* \approx 0.82$ and $h^* \approx (-9.62, -7.58, -4.96, -2.27, 0.45)$.

### D.2  Experiments in weakly-communicating environments

Beyond communicating models, `PMEVI-DT` is superior to `EVI`-based methods. On Fig. 4, we see that `PMEVI-DT` can learn in environments of infinite diameter while `UCRL2` cannot. The gain is due to the truncation operation, that makes sure that the optimistic bias vector has span less than $T^{1/5}$.

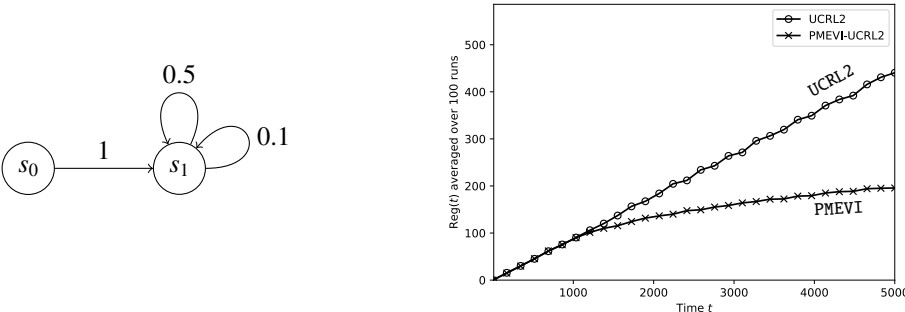

Figure 4:  Bernoulli bandit with dandling state (weakly-communicating model). Each arrow is a choice of action whose label is the mean reward of the associated state-action pair. The learner starts in state $s_0$ and transitions to the absorbing state $s_1$ as soon as an action is played.

Without the truncation operation, `EVI`-based methods can never reject the plausibility that the reward at $s_0$ is maximal (equal to one) and that there is a positive probability $\epsilon$ to switch from $s_1$ to $s_0$ when taking the sub-optimal action from $s_1$. More precisely, denote the actions $a, b$ where $r(s_0, a) = 1$, $r(s_1, a) = 0.5$ and $r(s_1, b) = 0.1$, so that $a$ is the optimal action and $b$ is sub-optimal. There are two policies $\pi_a$ and $\pi_b$ choosing action $a$ and $b$ from $s_1$ respectively. Because $s_0$ is only visited once, the best reward that can be achieved from $s_0$ is $\tilde{r}_t(s_0, a) = 1$ at all times. If one runs `UCRL2`, the confidence region for transition kernels is roughly of the form $\mathcal{P}_t(x) = \{\tilde{p}(x) : N_t(x) \|\tilde{p}(x) - \hat{p}_t(x)\|_1^2 \leq C_p \log(t)\}$ and the plausible transition kernel $\tilde{p}(x) \in \mathcal{P}_t(x)$ that goes the quickest from $s_1$ to $s_0$ is of the form:

$$\tilde{p}_t(s_0|s_1, a) = \sqrt{\frac{C_p \log(t)}{N_t(s_1, a)}} \quad \text{and} \quad \tilde{p}_t(s_0|s_1, b) = \sqrt{\frac{C_p \log(t)}{N_t(s_1, b)}}.$$

After running `EVI` for $i$ steps, the current vector $v_i$ (see (6)) is $V_i^*(\mathcal{M}_t)$, the maximal amount of reward that one can collect in $i$ steps on $\mathcal{M}_t$ seen as an extended Markov decision process, see Auer et al. [2009]. If the data is well concentrated, the optimal reward from $s_1$ are respectively $\tilde{r}_t(s_1, a) \approx 0.5 + \sqrt{C_r \log(t)/N_t(s_1, a)}$ and $\tilde{r}_t(s_1, b) = 0.1 + \sqrt{C_r \log(t)/N_t(s_1, b)}$. From all this, one can argue that when $i$ is large enough, the optimistic scores of $\pi_a$ and $\pi_b$ over $i$ steps are roughly equal to:

$$V_i^{\pi_a}(s_1; \mathcal{M}_t) \approx i + \left(-0.5 + \sqrt{\frac{C_r \log(t)}{N_t(s_1, a)}}\right)\sqrt{\frac{N_t(s_1, a)}{C_p \log(t)}} = i - 0.5\sqrt{\frac{N_t(s_1, a)}{C_p \log(t)}} + \sqrt{\frac{C_r}{C_p}}$$

$$V_i^{\pi_b}(s_1; \mathcal{M}_t) \approx i + \left(-0.9 + \sqrt{\frac{C_r \log(t)}{N_t(s_1, b)}}\right)\sqrt{\frac{N_t(s_1, b)}{C_p \log(t)}} = i - 0.9\sqrt{\frac{N_t(s_1, b)}{C_p \log(t)}} + \sqrt{\frac{C_r}{C_p}}.$$

So $V_i^{\pi_a}(s_1; \mathcal{M}_t) \le V_i^{\pi_b}(s_1; \mathcal{M}_t)$ if $N_t(s, b) \ll \frac{25}{81} N_t(s, a)$.

This means that `EVI` will output $\pi_b$ if $N_t(s, b) \ll \frac{25}{81} N_t(s, a)$, leading to $N_t(s, a) \asymp N_t(s, b)$ so both growing linearly with $t$. This informal argument can be generalized to `EVI`-based algorithms with other types of confidence regions: `EVI`-based methods such as `UCRL` Auer et al. [2002], `UCRL2B` Fruit et al. [2020] and `KLUCRL` Filippi et al. [2010] will suffer from $N_t(s, b) = \Theta(t)$ and their regret will grow linearly.

In opposition, `PMEVI`-based methods use truncation, making sure that $|v_i(s_1) - v_i(s_0)| \le T^{1/5}$ at all times. Intuitively, it makes `PMEVI`-based method "think" that $\tilde{p}_t(s_0|s_1, b)$ cannot be as small as $\sqrt{C_p \log(t)/N_t(s_1, b)}$, because the optimistic bias of $\pi_b$ would be too large otherwise; Or, equivalently, that $\tilde{p}_t(s_0|s_1, b) = \sqrt{C_p \log(t)/N_t(s_1, b)}$ but with $\tilde{r}_t(s_0, a) \ll 1$, hence killing the optimistic reward at $(s_0, a)$ to meet the bias constraints.

Overall, these features of `PMEVI`-based methods are shared with algorithms such as `REGAL` Bartlett and Tewari [2009] and `SCAL` Fruit et al. [2018]. The difference is that these methods require precise prior information on $\mathrm{sp}(h^*)$ that `PMEVI-DT` does not need.

# E  Standard concentration inequalities

**Lemma 32** (Azuma's inequality, Azuma [1967])**.** *Let $(U_t)_{t\geq 0}$ a martingale difference sequence such that* $\mathrm{sp}\,(U_t) \leq c$ *a.s., i.e., there exists $a_t \in \mathbf{R}$ such that $a_t \leq U_t \leq a_t + c$ a.s. Then, for all $\delta > 0$,*

$$\mathbf{P}\left(\sum_{t=0}^{T-1} U_t \geq c\sqrt{\tfrac{1}{2}T\log\left(\tfrac{1}{\delta}\right)}\right) \leq \delta.$$

**Lemma 33** (Freedman's inequality, Zhang et al. [2020])**.** *Let $(U_t)_{t\geq 0}$ a martingale difference sequence such that $|U_t| \leq c$ a.s., and denote its conditional variance $V_t := \mathbf{E}[U_t^2|\mathcal{F}_{t-1}]$. Then, for all $\delta > 0$,*

$$\mathbf{P}\left(\exists T' \leq T : \sum_{t=0}^{T'-1} U_t \geq \sqrt{2\sum_{t=0}^{T'-1}V_t\log\left(\tfrac{T}{\delta}\right)} + 4c\log\left(\tfrac{T}{\delta}\right)\right) \leq \delta.$$

**Lemma 34** (Time-uniform Azuma, Bourel et al. [2020])**.** *Let $(U_t)$ a martingale difference sequence such that, for all $\lambda \in \mathbf{R}$, $\mathbf{E}[\exp(\lambda U_t)|U_1,\ldots,U_{t-1}] \leq \exp(\frac{\lambda^2\sigma^2}{2})$. Then:*

$$\forall \delta > 0, \quad \mathbf{P}\left(\exists n \geq 1, \quad \left(\sum_{k=1}^n U_k\right)^2 \geq n\sigma^2\left(1+\tfrac{1}{n}\right)\log\left(\tfrac{\sqrt{1+n}}{\delta}\right)\right) \leq \delta.$$

**Lemma 35** (Time-uniform Weissman)**.** *Let $q$ a distribution over $\{1,\ldots,d\}$. Let $(U_t)$ a sequence of i.i.d. random variables of distribution $q$. Then:*

$$\forall \delta > 0, \quad \mathbf{P}\left(\exists n \geq 1, \left\|\sum_{i=1}^n (e_{U_i}-q)\right\|_1^2 \geq nd\log\left(\tfrac{2\sqrt{1+n}}{\delta}\right)\right) \leq \delta.$$

*Proof.* Remark that $\left\|\sum_{k=1}^n (e_{U_k}-q)\right\|_1 = \max_{v\in\{-1,1\}^d}\sum_{k=1}^n \langle e_{U_k}-q, v\rangle$. Let $W_k^v := \langle e_{U_k}-q, v\rangle$. Remark that for each $v \in \{-1,1\}^d$, $(W_k^v)$ is a family of i.i.d. random variables with $-\langle q,v\rangle \leq W_k^v \leq 1 - \langle q,v\rangle$, so $\mathbf{E}[\exp(\lambda W_k^v)] \leq \exp(\frac{\lambda^2}{8})$ by Hoeffding's Lemma. By Lemma 34, we have:

$$\mathbf{P}\left(\exists n \geq 1, \left\|\sum_{k=1}^n (e_{U_k}-q)\right\|_1 \geq \sqrt{nd\log\left(\tfrac{2\sqrt{1+n}}{\delta}\right)}\right) = \mathbf{P}\left(\exists v \in \{-1,1\}^d, \exists n, \sum_{k=1}^n W_k^v \geq \sqrt{nd\log\left(\tfrac{2\sqrt{1+n}}{\delta}\right)}\right)$$

$$\leq \sum_{v\in\{-1,1\}^d} \mathbf{P}\left(\exists n, \sum_{k=1}^n W_k^v \geq \sqrt{nd\log\left(\tfrac{2\sqrt{1+n}}{\delta}\right)}\right)$$

$$\leq \sum_{v\in\{-1,1\}^d} \mathbf{P}\left(\exists n, \sum_{k=1}^n W_k^v \geq \sqrt{\tfrac{1}{2}n\left(1+\tfrac{1}{n}\right)\log\left(\tfrac{\sqrt{1+n}}{2^{-d}\delta}\right)}\right)$$

$$\leq 2^d \cdot 2^d\delta = \delta.$$

This concludes the proof. $\qquad\square$

**Lemma 36** (Time-uniform Empirical Bernstein)**.** *Let $(U_k)_{k\geq 1}$ a martingale difference sequence such that $\mathrm{sp}\,(U_n) \leq c$ a.s., let $\hat{U}_n := \frac{1}{n}\sum_{k=1}^n U_k$ the empirical mean and $\hat{V}_n := \frac{1}{n}\sum_{k=1}^n (U_k - \hat{U}_n)^2$ the population variance. Then,*

$$\forall \delta > 0, \forall T > 0, \quad \mathbf{P}\left(\exists t \leq T, \sum_{i=1}^t U_i \geq \sqrt{2t\hat{V}_t\log\left(\tfrac{3T}{\delta}\right)} + 3c\log\left(\tfrac{3T}{\delta}\right)\right) \leq \delta.$$

*Proof.* This is obtained with a union bound on the values of $n \leq T$, then applying Lemma 38. $\qquad\square$

**Lemma 37** (Time-uniform Empirical Likelihoods, Jonsson et al. [2020])**.** *Let $q$ a distribution on $\{1,\ldots,d\}$. Let $(U_t)$ a sequence of i.i.d. random variables of distribution $q$. Then:*

$$\forall \delta > 0, \quad \mathbf{P}\left(\exists n \geq 1, n\,\mathrm{KL}(\hat{q}_n\|q) > \log\left(\tfrac{1}{\delta}\right) + (d-1)\log\left(e\left(1+\tfrac{n}{d-1}\right)\right)\right) \leq \delta.$$

**Lemma 38** (Empirical Bernstein inequality, Audibert et al. [2009])**.** *Let $(U_k)_{k\geq 1}$ a martingale difference sequence such that $\mathrm{sp}\,(U_n) \leq c$ a.s., let $\hat{U}_n := \frac{1}{n}\sum_{k=1}^n U_k$ the empirical mean and $\hat{V}_n := \frac{1}{n}\sum_{k=1}^n (U_k - \hat{U}_n)^2$ the population variance. Then,*

$$\forall \delta > 0, \forall n \geq 1, \quad \mathbf{P}\left(\sum_{k=1}^n U_k \geq \sqrt{2n\hat{V}_n\log\left(\tfrac{3}{\delta}\right)} + 3c\log\left(\tfrac{3}{\delta}\right)\right) \leq \delta.$$

**Lemma 39** (Bennett's inequality, Audibert et al. [2009])**.** *Let $(U_t)_{t \geq 0}$ a martingale difference sequence such that $|U_t| \leq c$ a.s., and denote its conditional variance $V_t := \mathbf{E}[U_t^2 | \mathcal{F}_{t-1}]$. Then,*

$$\forall \delta > 0, \forall n \geq 1, \quad \mathbf{P}\left(\exists k \leq n, \sum_{i=1}^{k} U_i \geq \sqrt{2 \sum_{i=1}^{n} V_i \log\left(\tfrac{1}{\delta}\right)} + \tfrac{1}{3} c \log\left(\tfrac{1}{\delta}\right)\right) \leq \delta.$$

**Lemma 40** (Lemma 3 of Zhang and Xie [2023])**.** *Let $(U_t)$ be a sequence of random variables such that $0 \leq U_t \leq c$ a.s., and let $\mathcal{F}_t := \sigma(U_0, U_1, \ldots, U_{t-1})$. Then:*

$$\forall \delta > 0, \quad \mathbf{P}\left(\exists T \geq 0, \sum_{t=0}^{T-1} U_t \geq 3 \sum_{t=0}^{T-1} \mathbf{E}[U_t | \mathcal{F}_{t-1}] + c \log\left(\tfrac{1}{\delta}\right)\right) \leq \delta;$$

$$\forall \delta > 0, \quad \mathbf{P}\left(\exists T \geq 0, \sum_{t=0}^{T-1} \mathbf{E}[U_t | \mathcal{F}_{t-1}] \geq 3 \sum_{t=0}^{T-1} U_t + c \log\left(\tfrac{1}{\delta}\right)\right) \leq \delta.$$

