# OpenReview forum: "Achieving Tractable Minimax Optimal Regret in Average Reward MDPs"
_NeurIPS.cc/2024/Conference — NeurIPS 2024 poster_

### Official Review · Reviewer_9hWK · 2024-06-29

**Soundness:** 2
**Presentation:** 3
**Contribution:** 2
**Rating:** 4
**Confidence:** 3

**Summary:**

The paper proposes the first tractable algorithm that achieves minimax optimal regret for average reward tabular MDPs. The algorithm does not require prior information on the span of the optimal bias function.

**Strengths:**

The paper proposes the first tractable algorithm that achieves minimax optimal regret for average reward tabular MDPs.

**Weaknesses:**

See questions.

**Questions:**

1. The minimax lower bound in [4] is sqrt{DSAT}. Because H=span(h^*) <= D, I wonder why sqrt{HSAT} is even achievable by the algorithm developed in the paper.

2. The authors cited [14, 25] for the lower bound sqrt{HSAT}, but I can not find this result/theorem in these two papers with a rigorous proof.  In fact, [14] mentioned it is an open problem whether the “actual” lower bound depends on diameter D or the bias span H. Please provide a proper reference if the actual lower bound depends on H.

3. As shown in Figure 2, PMEVI behaves almost the same with its EVI counterparts when no prior information on bias is given. This does not illustrate the advantage of PMEVI compared with UCRL2 or KLUCRL, even though the regret of the algorithm in this paper is claimed to be better.

4. In Figure 2 (to the right), PMEVI is run with bias information c, but it is compared with UCLR2 which does not use c. This is not fair comparison. If PMEVI is given bias information, then it should be compared with other algorithms such as SCAL or UCB-AVG which require bias information for implementation.

5. Computing max_u beta_t(s,a,u) may not be tractable in general. Algorithm 5 provides a way to bound/approximate this quantity. I wonder what is the intuition? I also can not find the proof that it is indeed a bound that does not impact the regret efficiency.

6. The regret proof in Section 4 does not highlight how the key components projection and mitigation in the algorithm impact the regret. In particular, it is not clear from the proof sketch where these components are used, and how the beta-mitigated extended bellman operator is applied to achieve the minimax optimal regret.

**Limitations:**

I do not find the discussions of limitations in the main body of the paper.

---

> ### Author Rebuttal · Authors · 2024-08-06
>
> We address your concerns below.
>
> > 1. The minimax lower bound in [4] is $\sqrt{DSAT}$. Because $H=\mathrm{span}(h^*) \le D$, I wonder why $\sqrt{HSAT}$ is even achievable by the algorithm developed in the paper.
>
> The lower bound of [4] is indeed $\sqrt{DSAT}$, but like pointed out by [14], the "hard instance" provided by [4] is one such that $\mathrm{span}(h^*) = D$. To quote [14], "*The proof of the lower bound relies on the construction of an MDP whose diameter actually coincides with the bias span (up to a multiplicative numerical constant), thus leaving the open question whether the “actual” lower bound depends on $D$ or the bias span.*"
>
> What [4] actually shows is that, whatever the algorithm, there exists a MDP with $c = \mathrm{span}(h^*) \asymp D$ such that the regret is $\sqrt{c S A T}$. It means that the lowerbound of [4] can be read $\sqrt{\mathrm{span}(h^*) S A T}$ just as $\sqrt{DSAT}$.
>
> > 2. The authors cited [14, 25] for the lower bound $\sqrt{HSAT}$, but I can not find this result/theorem in these two papers with a rigorous proof. In fact, [14] mentioned it is an open problem whether the “actual” lower bound depends on diameter $D$ or the bias span $H$. Please provide a proper reference if the actual lower bound depends on $H$.
>
> Indeed, [14, 25] do not provide any explicit lower bound. Like said in 1., [14] precises that the lower bound of [4] can be read as $\sqrt{\mathrm{span}(h^*) S A T}$, and [25] shows that it can be achieved by an intractable method with prior knowledge on the bias span, implicitely showing the the lower bound is tight. It is even tighter than the diameter lower bound, since $\mathrm{span}(h^*) \le D$ and $\mathrm{span}(h^*) \ll D$ in general.
>
> > 3. As shown in Figure 2, PMEVI behaves almost the same with its EVI counterparts when no prior information on bias is given. This does not illustrate the advantage of PMEVI compared with UCRL2 or KLUCRL, even though the regret of the algorithm in this paper is claimed to be better.
>
> The left part of Figure 2 was very disappointing at first. The poor advantage of `PMEVI` compared to `UCRL2` or `KLUCRL` can be explained as follows: The advantage of `PMEVI` depends on the quality of the bias estimator, that itself depends on the current quality of play so far. The thing is that the *early* quality of play is bad, because the algorithm still has to figure out which are the optimal actions. This early phase pollutes the bias estimator for quite some time, making it somehow irrelevant for a long time. This is what the right part of Figure 2 is there for: It shows that if the bias estimation is better (e.g., under prior information), `PMEVI` is very efficient at using it to reduce the burn-in time.
>
> Now, the left part of Figure 2 shows that there are still room for improvement, for example by improving the inner bias estimation of the method. We think that this is a different work. This paper is mostly theoretical and gives a direction, showing how to use an external trajectorial bias estimation to improve the minimax regret guarantees. Optimizing this bias estimation subroutine is an interesting direction for future work.
>
> > 4. In Figure 2 (to the right), PMEVI is run with bias information c, but it is compared with UCLR2 which does not use c. This is not fair comparison. If PMEVI is given bias information, then it should be compared with other algorithms such as SCAL or UCB-AVG which require bias information for implementation.
>
> In the right part of Figure 2, we indeed compare `PMEVI` to `UCRL2` instead of `SCAL`. We do not claim that `PMEVI` is *better* than `UCRL2` there, but rather that it can efficiently make use of a good quality prior information. This is to nuance the left part of Figure 2, that seems to show that the projection-mitigation procedure has no effect on the regret. Also, the kind of prior information that we feed to `PMEVI` is not one that `SCAL` can use that can only take a bias span upper bound (like "$\mathrm{span}(h^*) \le c$") into account. Even with tight bias information, `SCAL` will behave like just like `UCRL2` on this instance because plain bias span information is not really helpful here (the bias span is proportional to the diameter on river swim). We can discuss this further in the section **Details on Experiments** in the Appendix.
>
> > 5. Computing $\max_u \beta_t(s,a,u)$ may not be tractable in general. Algorithm 5 provides a way to bound/approximate this quantity. I wonder what is the intuition? I also can not find the proof that it is indeed a bound that does not impact the regret efficiency.
>
> The correction of Algorithm 5 is established in section A.2.2 with Lemma 13. The estimation used by Algorithm 5 is based on Lemma 12 that provides a general bound of $\mathbf{V}(p, u)$ with respect to $\mathbf{V}(p, v)$ for two $u, v$ in the bias confidence region. It will be linked better in the main text to access the proof more easily.
>
> Note that Algorithm 5 runs in polynomial time by providing , and that its output is only an upper-bound of $\max_u \beta_t(s,a;u)$. However, it is enough to achieve minimax optimal regret guarantees.
>
> > 6. The regret proof in Section 4 does not highlight how the key components projection and mitigation in the algorithm impact the regret. In particular, it is not clear from the proof sketch where these components are used, and how the beta-mitigated extended bellman operator is applied to achieve the minimax optimal regret.
>
> The impact of the projection-mitigation operation can be tracked by looking at where Lemma 13 is invoked. It is invoked pretty much everywhere but is mostly critical to bound the optimism overshoot (Lemma 9) and the second order error (Lemma 10). It is also used to bound the navigation error in probability (Lemma 7) but isn't necessary for this term if one is just looking for a bound in expectation.

---

> > ### Comment · Reviewer_EC5m · 2024-08-13
> >
> > The complaints of reviewer 9hWK29 fall into three categories:
> > 1) Role of D vs. H. This was a confusion on part of the reviewer, which stems from the fact that early literature focused on D, and it was only later that the role of H was discovered. I think the authors gave an excellent answer, clarifying all the confusion.
> > 2) The experiments are not really demonstrating that the theory holds up. I have the same problem as reviewer 9hWK29 in this regard and while I understand the response, I still think that it would have been better either completely omitting the experiments, or going after checking whether the claimed theoretical improvement also holds up in the experiments in some limited, but well chosen setting (i.e., design MDPs with variable H and confirm that the algorithm adapts to H as claimed).
> > 3) Complaints on the presentation/explanations in the paper. I share some of these concerns.
> > Overall, yet, I feel the rating of 4 is harsh and unjustified for a paper that is addressing a major open problem in the field and which, as far as we know, resolves this open problem. I, for one, would like to see papers at NeurIPS that do this, even if they are somewhat imperfect. In particular, I won't care even if there were absolutely no experiments. Of course, I also care about how well the paper is written and here I see room for improvement. However, I feel that in this regard the paper is passing the bar and I would hope the authors will improve the writing for the final paper. Eventually, what matters is whether the result is correct and whether it adds interesting new knowledge to the field and here I believe the answer is yes.

---

> > > ### Comment · Area_Chair_aVeh · 2024-08-14
> > > **Quality**
> > >
> > > Thank you for taking the time to read through. I agree with your assessment. If the results are indeed correct

---

### Official Review · Reviewer_f6aP · 2024-07-05

**Soundness:** 4
**Presentation:** 3
**Contribution:** 4
**Rating:** 7
**Confidence:** 3

**Summary:**

The authors propose a novel algorithm for weakly communicating Markov Decision Processes (MDPs) with an average-reward objective. This algorithm, at the same time, is a) tractable since it does not rely on the exact solution to a high-dimensional non-convex optimization problem as prior work, and b) achieves minimax optimal regret up to logarithmic factors.

**Strengths:**

- First tractable algorithm that achieves minimax optimal regret bound;
- The algorithm does not require any prior knowledge of the span of optimal bias function;
- The algorithm does not require any reduction to a discounted setting and works directly in the average-reward setting.

**Weaknesses:**

- The algorithm requires solving the linear programs for each doubling epoch, which is tractable but non-generalizable beyond the tabular setting;
- The large second-order term is dominated by the main asymptotic optimal term if $T > S^{20}$. This effect of a large second-order term is also observable during the experimental section.

**Questions:**

- Discussion in Section 3.3 is a little bit misleading since it is formulated in terms of arbitrary vector u and may think that, afterward, the union bound over possible u should be applied to write down a bound of type $\max_{u} \beta_{t}(s,a,u)$ (and thus inducing additional $\sqrt{S}$ factor due to equivalent with purely $\ell_1$-confidence regions),  whereas you need to do it only for $u = h^\star$. I would suggest rewriting this discussion a little bit to avoid the confusion I experienced during the first reading of the text.
- Appendix C6. It's not clear what (2k) means in the equation after line 796. I have the same question about (3k) and (4k) after line 816 and 825.
- Additionally, I would appreciate the additional discussion on the difference between regret minimization and best policy identification settings for average-reward MDPs, especially in a glance at existing concurrent work Tuynman et al. 2024.
Tuynman, A., Degenne, R., & Kaufmann, E. (2024). Finding good policies in average-reward Markov Decision Processes without prior knowledge. arXiv preprint arXiv:2405.17108.

**Limitations:**

The paper is of a theoretical nature and thus does not have any direct impact.

---

> ### Author Rebuttal · Authors · 2024-08-06
>
> ## Concerning the weaknesses
>
> > The algorithm requires solving the linear programs for each doubling epoch, which is tractable but non-generalizable beyond the tabular setting;
>
> Yes, the algorithm needs to solve many linear programs at every epoch for the vanilla version of `PMEVI`. However, solving *too many* linear programs can easily be avoided, especially in the early steps. Indeed, in the early steps of the process, the bias confidence is very bad and pretty much uninformative. After all, in order to estimate the bias correctly, the regret of the algorithm has to be sublinear and this is simply not the case in the early stages of learning. In practice, it means the the projection operation of `PMEVI` does nothing in the beginning. This can be used to avoid a call to a linear program solver (by trying beforehand if a projection on the bias confidence region is necessary at all) hence having the nearly the same running time than `EVI`. In the later stages of learning, changes of episodes are more and more rare, so solving these linear programs is somehow amortized. This is how we obtain the complexity result of Theorem 1.
>
> ## Concerning your questions
>
> > Discussion in Section 3.3 is a little bit misleading since it is formulated in terms of arbitrary vector $u$ and may think that, afterward, the union bound over possible $u$ should be applied to write down a bound of type $\max_u \beta(s,a,u)$ (and thus inducing additional factor $\sqrt{S}$ due to equivalent with purely $\ell_1$-confidence regions), whereas you need to do it only for $u = h^*$. I would suggest rewriting this discussion a little bit to avoid the confusion I experienced during the first reading of the text.
>
> We suggest the following modification at line 214.
>
> "(...) are fixed. [Many existing works [4,7,8,11,12,13,14] look for a bound on $(\hat{p}_t(s,a) - p(s,a))u$ that holds uniformly for all vector $u$, because the values of $u$ encountered along the iterations of `EVI` are not known in advance, and depend on the random data gathered by the algorithm. This is morally achieved by doing a union bound for all $u$, which is responsible for the extra $\sqrt{S}$ in the regret guarantees of these methods. In Appendix B, we show that we only need (11) to hold for $u = h^*$ that is a fixed constant, so] one is tempted to use (11) to mitigate (...)."
>
> > Appendix C6. It's not clear what (2k) means in the equation after line 796. I have the same question about (3k) and (4k) after line 816 and 825.
>
> $(2k)$ is the empirical bias error (introduced when starting the regret decomposition at line 256). We will recall the definition of this shorthand. Similarly, the definitions of $(3k)$ and $(4k)$ can be found at line 256. We will recall their definitions at the beginning of every proof section.
>
> > Additionally, I would appreciate the additional discussion on the difference between regret minimization and best policy identification settings for average-reward MDPs, especially in a glance at existing concurrent work Tuynman et al. 2024. Tuynman, A., Degenne, R., & Kaufmann, E. (2024). Finding good policies in average-reward Markov Decision Processes without prior knowledge. arXiv preprint arXiv:2405.17108.
>
> Indeed, it has been shown in the mentioned work that in the best policy identification setting (PAC learning), achieving performance that depend on the bias function rather than the diameter *requires* prior information on the bias function; While we achieve regret guarantees that depend on the bias function even though we have no prior information on it. This is because in PAC learning, the algorithms has (implicitely) to provide a certificate of optimality. It has to assess that the output policy is nearly optimal with high probability. This certificate of optimality is not necessary in regret minimization and in practice, `PMEVI` plays a policy that it thinks is optimal but it does not have a way to certify that it is optimal. This is very likely to mean that if the algorithm where to produce policy certificates in addition to minimizing the regret (see *Policy Certificates: Towards Accountable Reinforcement Learning*, Dann  et. al., 2019), then the regret $\sqrt{\mathrm{span}(h^*) S A T}$ would not be achievable. This discussion can be added to the conclusion.
>
> Note that there is sensible difference (in addition to the learning objective) between our work and the above mentioned work: **In our paper, the learner does not have access to a generative model**. With the generative model assumption, the learner can obtain a sample of any state-action pair in the MDP without constraints, making exploration much less difficult.

---

> > ### Comment · Reviewer_f6aP · 2024-08-11
> >
> > I would like to thank the authors for their response and I am happy to keep my score.

---

### Official Review · Reviewer_zKbr · 2024-07-12

**Soundness:** 3
**Presentation:** 2
**Contribution:** 4
**Rating:** 7
**Confidence:** 3

**Summary:**

This paper studies learning in average reward MDPs, and presents the first minimax optimal algorithm (in terms of $sp(h^*)$) which is computationally tractable, and also simultaneously the first which does not require prior knowledge of $sp(h^*)$.

**Strengths:**

The main theorem resolves a longstanding open problem which has been the subject of extensive research effort. Removal of knowledge of $sp(h^*)$ is a challenging problem in many settings beyond the one considered in this paper, and thus I hope that this work might lead to progress on this issue in other areas.

**Weaknesses:**

The algorithm is rather complicated and has many components. The presentation also does not make it easy to determine the key ingredients of this method. While it is interesting that this method may incorporate different confidence regions, I wonder if it might be more clear to present things in less generality.

It is nice that experiments are included, but if anything they seem to contradict the claim that the method achieves minimax regret without prior knowledge of $sp(h^*)$ (due to the left plot in Figure 2). Since the bias estimation subroutine seems to be a key ingredient, it would be much better if an experiment was included where this subroutine was making a nontrivial contribution (by my understanding, in the left plot the bias estimation routine is not doing anything since it hasn't been given enough time, and in the right plot, it is not doing anything since much better prior information has been given to the algorithm).

**Questions:**

Why is the main theorem stated with the parameter $c$? It seems like the best choice is always $c=sp(h^*)$? (And if $c \gg sp(h^*)$, then the theorem is not giving the minimax optimal regret.) Does the theorem hold for all $c$ simultaneously or is this parameter used somewhere that I didn't notice?

The main theorem mentions a confidence region system for communicating MDPs with a diameter-dependent computational complexity, but since the paper is for weakly communicating MDPs which generally have infinite diameter, I think a different system would be a better choice? Can the author(s) comment on the complexity when the diameter is infinite?

Many times notation like $h^*(\mathcal{M})$ is used, but is this well-defined? There can be multiple solutions to the Bellman optimality equation.

The paper https://arxiv.org/pdf/1905.12425 claims to achieve the optimal diameter-based $\sqrt{DSAT}$ regret, maybe it should be added to the related work.

In the display below line 785, the number of commutes between $S_t$ and $S_{t+1}$ is lower bounded by the number of transitions between these states. A similar bound is used under line 830. Does this suggest that the bias difference estimator could actually be formulated using only direct transitions between pairs of states?


Line 397: $h^*$ should be $g^*$?

Line 404: Some character I believe meant to be $S$ was used

**Limitations:**

No major limitations

---

> ### Author Rebuttal · Authors · 2024-08-06
>
> We have numbered your questions to gain a few characters and address all of them.
>
> ## Concerning the weaknesses
>
> > The algorithm is rather complicated and has many components (...) While it is interesting that this method may incorporate different confidence regions, I wonder if it might be more clear to present things in less generality.
>
> From a high level, there are two confidence regions. One is for the model parameters (rewards and kernels) and the other is for the bias function. This second confidence region can be considered as "external" and its purpose is to regularize the "vanilla" optimism machinery of `EVI` introduced by [4]. Actually, only the bias confidence region and the regularization mechanism (projection + mitigation) is necessary to achieve minimax optimal regret, although the model confidence region seems very important to reduce the burn-in time of the algorithm. We believe that the generality of the approach is a strong point, because it shows that many existing algorithms can be patched all at once with a single analysis.
>
> > It is nice that experiments are included, but if anything they seem to contradict the claim that the method achieves minimax regret without prior knowledge of (due to the left plot in Figure 2). (...)
>
> Yes, sadly, experiments are a bit disappointing. `PMEVI` provides solid formal grounds to use bias information in order to improve performance (this is displayed by the right part of Figure 2) and Figure 2 also shows that getting pertinent bias information is difficult (left part). This leaves opportunities for future work: Improving the bias estimation subroutine would directly improve the practical performance of `PMEVI`.
>
> For instance, the performance of the algorithm is very bad in the early learning stages. This cannot be avoided because the algorithm has still to figure out which actions seem bad and which seem good. This early data pollutes the bias estimator, making the bias confidence region irrelevant for quite some time. To address it, one idea could be to trash data when it is considered too old. We do not do it in the paper because it would complicate the analysis even more. Also the current version of `PMEVI` was left "not tuned" on purpose, to ease the construction of algorithms based on `PMEVI` later on.
>
> ## Concerning your questions
>
> 1. We use $c > 0$ instead of $\mathrm{span}(h^*)$ because the results holds for all MDPs with bias span less than $c$. This is just out of formality because the bound doesn't hide any other term than $\mathrm{sp}(h^*), S, A, T$ and $\delta$ (like $p_\mathrm{min}$, or the diameter, or the mixing time, or else).
>
> 2. The main theorem mentions a confidence region system for communicating MDPs with a diameter-dependent computational complexity, but since the paper is for weakly communicating MDPs which generally have infinite diameter, I think a different system would be a better choice? Can the author(s) comment on the complexity when the diameter is infinite?
>
> The confidence region for $M$ is not specific to the fact that $M$ is communicating (in fact, the confidence region of `UCRL2` [4] and its variants do not rely on the communicating property). However, the complexity result only holds for communicating MDPs as it is currently written indeed. We see two ways of addressing this.
>   + Either we simply restrict the complexity result to communicating MDPs (no need for extra material required, just make the precision in Theorem 1);
>   + Or we generalize the complexity result to weakly-communicating MDPs. This can be done like so. The complexity result is established with Proposition 17 and at line 559, we show that the `PMEVI` needs $O(\mathrm{span}(w_0)T/(S^{1/2} \log(T)))$ iterations to converge, where $w_0$ is the error between the initial value of `PMEVI` and the bias function  $h^*(\mathcal{M}_t)$ of the MDP confidence region $\mathcal{M}_t$ seen as an extended MDP [4]. If $M$ is communicating, [4, 7] show that this bias span is bounded by $D$, the diameter of $M$. If $M$ is weakly-communicating, the argument of [4, 7] can be generalized to show that it is bounded by "diameter of the communicating part of $M$" + "worst hitting time to communicating part of $M$", that we refer to as the **weak diameter** (we say "weak" because it is finite if and only if the MDP is weakly communicating).
>
> The first suggestion is better connected to the current literature because the diameter is a well-known object. The second is more complete, but requires a proof that is not currently written in the paper, and this notion of weak diameter is not standard.
>
> 3. The notation $h^*(M)$ refers to the *optimal bias function* which is a special solution to the Bellman equations. It is the maximal bias vector over policies with optimal gain, see [18]. The notation $h^*(\mathcal{M})$ is only used in the proof of Proposition 17 and is the optimal bias function of $\mathcal{M}$ seen as an extended MDP (see [4]).
>
> 4. Thank you for the pointer on https://arxiv.org/pdf/1905.12425, we will discuss discuss its contribution in the next revision opportunity.
>
> 5. You are completely right: the proof only keeps tracks of the commutes between $S_t$ and $S_{t+1}$. The bias difference estimator can not work with only direct transitions. The term $A$ in page 13 is no longer a martingale if we only count direct transitions.
>
> > Line 397: should $h^*$ be $g^*$?
>
> Yes, you are correct.
>
> > Line 404: Some character I believe meant to be $S$ was used
>
> Yes, it should be $S_{\tau_{i+1}}$ rather than $\S_{\tau_{i+1}}$.

---

> > ### Comment · Reviewer_zKbr · 2024-08-12
> >
> > Thank you for your response. I find the global rebuttal still leaves me with doubts about the experimental results. Overall, I will keep my score.

---

### Official Review · Reviewer_EC5m · 2024-07-15

**Soundness:** 3
**Presentation:** 2
**Contribution:** 4
**Rating:** 7
**Confidence:** 3

**Summary:**

The paper shows that by replacing the extended value iteration in optimistic methods, like UCRL2, it is possible to obtain regret that scales optimally with the number of states, actions, the time horizon and the span of the optimal value function, instead of the diameter, despite not knowing the diameter, assuming weakly communicating MDPs. In addition, a novel analysis is presented that also gives a polynomial bound on the compute cost of the algorithm. The new extended value iteration method is designed based on new ideas to refine the set of plausible MDPs considered in a given step: For this, an inequality is provided that relates the deviation of the values assigned to two states to observable quantities; creating a new set of constraints on the MDPs considered.

**Strengths:**

A major breakthrough if the proof holds up. Interesting insight.

**Weaknesses:**

The presentation is not great; the paper feels rushed, the paper is full of (minor) grammatical and typographical errors (starting with the abstract: "encounter suffer"). While I did not go through the whole proof, there are many minor problems that are apparent: The paper is not very careful in tracking error events. For example, \tilde{beta} is sometimes appearing in a condition, but it is also constructed by the algorithm, perhaps things work out, but details are definitely missing.
The experimental results are unconvincing: Experiments that show scaling the optimal span would have been more convincing. The presented experiments do not help at all.

**Questions:**

Are you sure you tracked all the error events correctly? (E.g. in Lemma 13, Lemma 3 is used, which needs $\tilde{g}\ge g^*$, but the proof of Lemma 13 does not mention why this would hold and exactly what value of $\tilde{g}$ is used here.)
One of the confusing aspects of the presentation of the insight was that on page 5, where the mitigation is explained, beta_t(s,a,u) is used and then a maximum over u is taken; while it is not explained whether beta_t(s,a,u) bounds the deviation that is a function of u for all u (needs a covering?). I guess this covering is not needed, but then why can we take the max as suggested in this paragraph (lines 155-161).

**Limitations:**

n.a.

---

> ### Author Rebuttal · Authors · 2024-08-06
>
> ## About weaknesses
>
> > The presentation is not great; the paper feels rushed, the paper is full of (minor) grammatical and typographical errors (starting with the abstract: "encounter suffer")
>
> We will do our best to correct as many typos as possible, that can make understanding the paper difficult.
>
> > While I did not go through the whole proof, there are many minor problems that are apparent: The paper is not very careful in tracking error events. For example, \tilde{beta} is sometimes appearing in a condition, but it is also constructed by the algorithm, perhaps things work out, but details are definitely missing.
>
> Concerning the tracking of events, we discuss in further details in the next section (see **About questions**).
>
> > The experimental results are unconvincing: Experiments that show scaling the optimal span would have been more convincing. The presented experiments do not help at all.
>
> Regarding experiments, we have carefully entitled the section **Numerical illustrations** rather than **Experiments**, that we do not claim to account for a throughout experimental campaign. This paper is theoretical and our algorithm `PMEVI` was intentionally "purified" of any form of tuning and modification to improve its empirical performance. The point of `PMEVI` and Figure 2 is to show that the method can make very good use of pertinent bias information, provided that the quality of this information is high. By leaving `PMEVI` not tuned, we make it easier to build extensions of the algorithm for other settings. Obviously, one interesting follow-up of this work would be to provide a finely tuned version of `PMEVI` with a better bias estimation subroutine, a bit like what `UCRL3` [8] is to `UCRL2` [4]. However, such a heavily tuned version would become impossible to adapt to settings others than undiscounted infinite horizon reinforcement learning problems.
>
> ## About your questions
>
> > Are you sure you tracked all the error events correctly? (E.g. in Lemma 13, Lemma 3 is used, which needs $\tilde{g} \ge g^*$, but the proof of Lemma 13 does not mention why this would hold and exactly what value of is used here.)
>
> **Concerning events.** If we understand your concern correctly, you legitimately fear the presence of a circular argument, because the optimism guarantees of an episode seem to depend on the preceding episodes. This is indeed the case, but there is no circularity and all risks of circularity are encapsulated in Lemma 13, that establishes anytime optimism guarantees (and more). Roughly speaking, there are two types of events in the analysis. There is the "main event" specified by Lemma 13, that provides time-uniform (1) optimism guarantees, (2) correctness of the bias confidence region (i.e., of projection) and (3) correctness of the mitigation.
>   + Indeed, Lemma 13 is established using Lemma 3. The blocked equation between lines 470 and 471 is universally quantified for $\tilde{g} \ge g^*$, i.e., we should read: "Let $E_2$ the event stating that, for all $T' \le T$ [and $\tilde{g} \ge g^*$], (...)". We will make the correction.
>   + Then, we agree that $E_1, E_2, E_3$ do not depend on how $\mathfrak{g}_k, \mathfrak{h}_k$ are constructed, and hold **independently of how the driver chooses actions.**
>   + So, on $E_1, E_2, E_3$, we show that the properties (1-3) (optimism, correctness of the projection and of the mitigation) *will propagate* from episode to episode.
>   + This high level view will be added in the paper to help the understanding of the proof.
>
> And then, there are "error terms specific" events, that are invoked on the fly. These are only necessary in the regret analysis. Note that the quality of the bias estimator depends on the regret of the algorithm, hence we end up with a self-bound on the regret (see line 265). However, the self-bound is of the form $x \le \alpha + \beta \sqrt{x}$ hence ends up being useful at all.
>
> > One of the confusing aspects of the presentation of the insight was that on page 5, where the mitigation is explained, $\beta_t(s,a,u)$ is used and then a maximum over u is taken; while it is not explained whether $\beta_t(s,a,u)$ bounds the deviation that is a function of $u$ for all $u$ (needs a covering?). I guess this covering is not needed, but then why can we take the max as suggested in this paragraph (lines 155-161).
>
> This indeed needs to be clarified, because this is one of the key ideas. Any bound on $(\hat{p}_t(s,a) - p(s,a))u$ that holds for *every $u$ simultaneously* will inevitably scale with $\sqrt{S}$ and compromise the minimax optimality of regret guarantees. Therefore, with `PMEVI`, we do not use the event $\{\forall u, (\hat{p}_t(s,a) - p(s,a)u \le \beta_t(s,a;u)\}$. In fact, we only need $\{(\hat{p}_t(s,a) - p(s,a))u \le \beta_t(s,a;u)\}$ to hold for $u = h^*$, and that does not require any covering.
>
> The maximum is taken for "algebraic" reasons.
> The mitigation operation makes sure that optimistic transitions $\tilde{p}(s,a)$ are chosen so that $\tilde{p}(s,a)u \le \hat{p}(s,a) u + \beta_t(s,a;u)$. It happens that using a mitigation coefficient $\beta_t(s,a;u)$ that depends on $u$ provides an ill-behaved operator (monotony is lost), and instead the mitigation coefficient $\beta_t(s,a) := \max_{u \in \mathcal{H}_t} \beta_t(s,a;u)$ is used, leading to the current equation (7).

---

> > ### Comment · Reviewer_EC5m · 2024-08-08
> > **acknowledgement of rebuttal**
> >
> > The rebuttal is fine; I wish I could read a polished version of the paper before it gets published. However, I am optimistically assuming for now that the presentation issues will be smoothened out and the result will still hold.

---

### Author Rebuttal · Authors · 2024-08-06

We appreciate the reviewers for their constructive suggestions. Please find our responses to the reviews below.

**From your collective feedback, it emerges a shared concern about the numerical experiments.** Indeed, the left part of Figure 2 shows that the projection-mitigation operations of `PMEVI` have very little impact on the behavior of the algorithm.

By monitoring the execution of the algorithms more closely, this comes from the fact that the confidence region of the bias vector is very large in the early iterations, hence the projection operation does nothing, and the mitigation operation has a very minor effect. These two operations progressively trigger over the run, but in practice it takes time, for two reasons.

- The quality of the bias estimator depends on the regret. In the early stages of learning, the regret grows linearly hence the bias estimator is way off, and the bias confidence region is accordingly very wide.
- This early phase pollutes the bias estimator/confidence region for quite some time. One idea of optimization would be to discard the early history. This would directly improve the quality of the bias estimator.

This is what the right part of Figure 2 is for: If `PMEVI` has good bias prior information, or if the bias estimation is much better, the performance are greatly improved. The current bias estimator is sufficient to obtain minimax optimal bounds and was left not tuned **on purpose**: to make sure that the theoretical analysis is free of hard-to-understand terms that only come from heavy hand-tuning. For now, the general analysis of `PMEVI` shows that all the optimistic algorithms based on `EVI` can be patched to achieve minimax optimal regret, with a bias estimation subroutine and the projection-mitigation regularization of `EVI`.

This is why the experiment section is named "**Numerical illustrations**".
This section provides a clear direction to improve `PMEVI`: improving the bias estimation subroutine. We believe that this is an interesting direction for future work.

---

### Decision · Program_Chairs · 2024-09-25

**Decision:**

Accept (poster)

**Comment:**

This paper closes a gap in the theoretical literature. There might be some issues in the proof, but they appear fixable.